# Emission of trace gases and aerosols from biomass burning – An updated assessment

Meinrat O. Andreae[1,2]

[1]Max Planck Institute for Chemistry, Mainz, Germany
5 [2]Scripps Institution of Oceanography, University of California San Diego, La Jolla, California, USA

*Correspondence to*: Meinrat O. Andreae (m.andreae@mpic.de)

**Abstract.** Since the publication of the compilation of biomass burning emission factors by Andreae and Merlet (2001), a large number of studies has greatly expanded the amount of available data on emissions from various types of biomass burning. Using essentially the same methodology as Andreae and Merlet (2001), this paper presents an updated compilation of emission 10 factors. The data from over 370 published studies were critically evaluated and integrated into a consistent format. Several new categories of biomass burning were added, and the number of species for which emission data are presented was increased from 93 to 121. Where field data are still insufficient, estimates based on appropriate extrapolation techniques are proposed. For key species, the updated emission factors are compared with previously published values. Based on these emission factors and published global activity estimates, I have derived estimates of pyrogenic emissions for important species released by the 15 various types of biomass burning.

## 1 Introduction

Biomass burning, in the form of open vegetation fires and indoor biofuel use, is one of the largest sources of many trace gases and aerosols to the global atmosphere. For some important atmospheric pollutants, like black carbon (BC) and primary organic aerosol (POA), biomass burning is the dominant global source; based on the estimates of Bond et al. (2013), 20 it accounts for 59% of BC emissions and 85% of POA emissions worldwide. Open vegetation fires alone represent about one-third to one-half of global carbon monoxide (CO) and 20% of nitrogen oxide ($NO_x$) emissions (Olivier et al., 2005; Wiedinmyer et al., 2011). Fires are also a major source of greenhouse gases, including carbon dioxide ($CO_2$), methane ($CH_4$), and nitrous oxide ($N_2O$) (Ciais et al., 2013; Tian et al., 2016; Le Quéré et al., 2018). While a significant fraction of the emitted $CO_2$ is taken up again by vegetation regrowth, much of it remains in the atmosphere for years and potentially even up to centuries, 25 e.g., in the case of tropical deforestation fires or peat soil burning (van der Werf et al., 2017). Model simulations suggest that in the absence of fires, atmospheric $CO_2$ concentrations would be about 40 ppm lower, indicating the importance of fires for the atmospheric carbon budget and climate (Ward et al., 2012). Biomass burning is the second largest global source of non-methane organic gases (NMOGs, also referred to as volatile organic compounds, VOCs) (Yokelson et al., 2008; Akagi et al., 2011). Numerous other studies have reached similar conclusions about the importance of biomass burning for atmospheric

composition (e.g., Crutzen and Andreae, 1990; Andreae and Rosenfeld, 2008; Andreae et al., 2009; Kaiser et al., 2012; van der Werf et al., 2017).

The resulting perturbations of the atmospheric burdens of trace gases and aerosols have important consequences for climate, biogeochemical cycles, and human health. Aerosols from biomass burning affect the regional and global radiation balance and impact cloud properties and precipitation (Andreae et al., 2004; Andreae and Rosenfeld, 2008; Rosenfeld et al., 2008; Ward et al., 2012; Tosca et al., 2013; Rosenfeld et al., 2014; Jiang et al., 2016; Braga et al., 2017; Cecchini et al., 2017; Hamilton et al., 2018; Thornhill et al., 2018). By shifting the proportions of direct and indirect solar radiation, they also influence primary productivity and thereby forest growth and agricultural production (Artaxo et al., 2009; Rap et al., 2015; Malavelle et al., 2019; McKendry et al., 2019). Fires mobilize nutrients, such as nitrogen, phosphorus, and potassium, which can deplete local ecosystem nutrient reservoirs on one hand and provide nutrients to other ecosystems by atmospheric transport on the other (Andreae, 1991; Andreae et al., 1998; Mahowald et al., 2008; Chen et al., 2010b). The VOCs and $NO_x$ in biomass smoke undergo smog photochemistry in the atmosphere, leading to the production of ozone, secondary organic aerosols, and other pollutants, which impact plant productivity (Crutzen and Andreae, 1990; Andreae, 1991; Robinson et al., 2007; Jaffe and Wigder, 2012; May et al., 2013; Pacifico et al., 2015; Hatch et al., 2017; Yue and Unger, 2018). These gaseous pollutants, and even more so the particulate matter from biomass burning, pose grave risks to human health (Naeher et al., 2007; Akagi et al., 2014; Dennekamp et al., 2015; Knorr et al., 2017; Apte et al., 2018). Recent estimates of global excess mortality from outdoor air pollution range from 4.2 to 8.9 million annually (Cohen et al., 2017; Lelieveld and Pöschl, 2017; Shiraiwa et al., 2017; Burnett et al., 2018; Lelieveld et al., 2019), with smoke from open vegetation burning accounting for up to 600,000 premature deaths per year globally (75th percentile of model estimates; Johnston et al., 2012). In addition to outdoors exposure, pollution from indoor solid fuel use, much of it biofuel burning, has been estimated to cause 2.8 million premature deaths annually (Kodros et al., 2018).

In view of the immense impact of biomass burning emissions on climate, ecosystem function, and human well-being, it is disconcerting that large uncertainties persist regarding the amounts emitted and their spatial and temporal distribution. For bottom-up emissions estimates, two basic types of information are required: the amount of the various types of biomass burned as a function of time and space, and the emission factors for the various emitted species, i.e., the amount of a given species emitted per unit mass of biomass burned. Considerable effort has gone into quantifying the magnitude of open biomass burning by remote sensing approaches (Mouillot et al., 2006; Reid et al., 2009; Mieville et al., 2010; Wiedinmyer et al., 2011; Kaiser et al., 2012; Ichoku and Ellison, 2014; Darmenov and da Silva, 2015; Chuvieco et al., 2016; van der Werf et al., 2017), but the estimates in these studies of the annual amounts of carbon released still range over a factor of three from 1.5 to 4.7 Pg a$^{-1}$. A model intercomparison based on state-of-the-art dynamic global vegetation models (DVGMs) yielded an even wider range of 1.0 to 4.9 Pg a$^{-1}$ (Li et al., 2019a).

Efforts to narrow the uncertainties in the emission factors for the large number of species emitted from the diverse types of burning are ongoing in the form of numerous field campaigns and laboratory studies. The results of these studies are, however, widely dispersed among hundreds of papers in a large number of journals, each dealing with a particular campaign

or experiment. Over the last two decades, there have been two efforts to synthesize these data on a global scale, one by Andreae and Merlet (2001; below referred to as A&M2001) and the other by Akagi et al. (2011). The latter included more recent data and additional species and burning types, and is available in updated form at http://bai.acom.ucar.edu/Data/fire/. As part of the The Fire INventory from NCAR (FINN) model, Wiedinmyer et al. (2011) selected data from these two sources into a "best

estimate" set of emission factors. In the present study, I am presenting an updated set of emission factors, which includes the results of studies published since the writing of the two previous compilations. It provides emission estimates for 28 more chemical species, for which a sufficient amount of field data has become available since A&M2001, as well as an extended set of burning types. The extratropical forest category is differentiated into boreal and temperate forest burning, domestic biofuel use is separated into non-dung and dung burning, and peat fires and domestic waste burning are added as new catego-

ries. Based on these emission data and recent activity estimates, I present a compilation of global emission amounts and make some recommendations regarding priorities for future investigations.

## 2 Methods

### 2.1 Data selection

This paper applies the same methodological approach as A&M2001, and therefore the methods section will only

15 provide a brief overview of the definitions and calculation methods used, and highlight those points where the present approach differs from the previous one. For all other details, the reader is referred to A&M2001. The original data, which form the basis of the emission factor averages presented in Table 1, can be found in an Excel spreadsheet in the Supplement.

With few exceptions, and consistent with the approach used in A&M2001, I only used results from field measurements in young fire plumes for the compilation of the emission factor data in Table 1. Ideally, these measurements had been

made within minutes after the smoke was released from the fires to avoid significant chemical changes during atmospheric aging, especially in the case of reactive trace gases. This is only possible, however, when sampling at the ground or from aircraft very close to the fire. In many other cases, aircraft were sampling at some distance from the fires, often without actually knowing the exact location of the fire. In such cases, I have rejected the data for the more reactive trace gases. A special case is presented by emission data calculated from remote sensing by either satellite measurements or ground-based solar Fourier

Transformation Infrared (FTIR) spectrometry. Here, the authors have often included a correction for atmospheric transformations, using model calculations involving transport times and reaction rates of the species concerned. Because of their large spatial and temporal coverage, such measurements are quite valuable, and I have therefore included some of them in this assessment, as long as they were either dealing with long-lived species or used appropriate correction methods (i.e., chemistry-transport model calculations to correct for atmospheric transformations) (Rinsland et al., 2007; Mebust et al., 2011; Tereszchuk

et al., 2011; Tereszchuk et al., 2013; Schreier et al., 2015; Viatte et al., 2015; Lutsch et al., 2016; Adams et al., 2019). They can be compared with in-situ measurement results by referring to the original data in the Supplement spreadsheet.

Another special case are the emission factors for gaseous elemental mercury ($Hg^0$). Here, only relatively few actual field emission measurements are available for most of the combustion types listed in Table 1. Therefore, I have followed the approach of Friedli et al. (2009) and included $Hg^0$ emission factors from studies that are based on the Hg content of the fuels and the assumption of total volatilization of Hg from the fuel during combustion, which appears well justified for this volatile element.

Generally, the results from laboratory combustion studies have not been included in the emission factor averages for the different fire types in Table 1, but they are given for comparison in a separate column in Table 1. The reason for this decision is that such experiments often do not reproduce realistic burning conditions in the field. For example, it has been shown that the emissions of many trace gases are strongly dependent on fuel moisture, temperature, wind, and other fire environment parameters (e.g., Chen et al., 2010a; Robertson et al., 2014; Liu et al., 2017; Thompson et al., 2019). The fuels in lab experiments, however, may be well aged and dried, and thus have a much lower moisture content than fuels in the field, and the wind conditions in the field are impossible to reproduce in the lab. This can be seen in the values of the modified combustion efficiency [MCE; the ratio of $\Delta CO_2/(\Delta CO_2+\Delta CO)$] in many lab studies, which are much higher than those typical of field burns, an extreme example being the study by Sirithian et al. (2018), who reported a mean MCE of 0.9996 in a lab study on biofuel burning. Therefore, lab results are only used in some special cases, where little or no field data are available and where the lab data appear representative based on their MCE (e.g., Christian et al., 2003), or had been adjusted to reflect field conditions using "overlap species", ERs, or MCE as discussed in Yokelson et al. (2013b). Some lab values are also used as estimates in Table 1; they are shown in italics and indicated as "LV" in the last column.

The studies on emissions from biofuel burning for cooking or heating represent a borderline case, as they are often conducted in a laboratory environment, but with an effort to simulate the actual fuel use conditions and stove setups used in households. Here, I have favored studies performed in actual households, but also included results from lab studies that appeared to realistically emulate field conditions. Results from modern residential biofuel combustion units, such as automated pellet burners or modern low-emission stoves, etc., have not been included. A more detailed analysis of emissions from different types of domestic biofuel use can be found in Akagi et al. (2011), albeit without the benefit of the numerous papers that have been published on these emissions in the last decade. A special review on this issue would be desirable, but is beyond the scope of this paper.

In contrast to gaseous compounds, which are chemically well defined, aerosols are complex and variable mixtures of organic and inorganic species and comprise particles across a wide range of sizes. This affects in particular the measurements of organic aerosol, black/elemental carbon, and size fractionated aerosol mass. Organic aerosol is usually measured either by a variety of thermochemical or mass spectrometric methods, both of which may have positive and negative artefacts, for which different authors have applied different corrections. Since some techniques report the result as organic aerosol mass and others as organic carbon mass concentrations, a conversion had to be applied. To convert between organic carbon and organic matter (OM), a default OM/OC mass ratio of 1.6 is used in the absence of specific information. This value is based on the data from fresh biomass smoke aerosol in the literature (Turpin and Lim, 2001; Aiken et al., 2008; Yokelson et al., 2009; Takahama et

al., 2011; Kostenidou et al., 2013; Brito et al., 2014; Collier et al., 2016; Fang et al., 2017; Tkacik et al., 2017; Ahern et al., 2019; Lim et al., 2019). Where only O/C ratios were given, they were converted to OM/OC ratios using the relationship given in Aiken et al. (2008).

Black carbon (BC) and elemental carbon (EC) are an even more problematic category. Various definitions for these
species have been used (Andreae and Gelencsér, 2006), but most commonly BC refers to carbon with specific optical properties (light absorption) and is measured by optical techniques, whereas EC is defined by its chemical properties and determined by a variety of thermochemical methods. Not all authors, however, adhere to these definitions, and the terms soot, EC, and BC are often used interchangeably. Unfortunately, while some techniques have been shown to have less bias than others (Li et al., 2019b), there is no general answer as to which technique is best, and which property, optical or chemical, is more representa-
tive. In view of the lack of a better alternative, both BC and EC data have been merged in the "BC" category here.

The size distribution of biomass smoke aerosols is a continuum ranging from tens of nanometers to millimeters (Reid et al., 2005), with most of the mass present in a mode at a few hundred nanometers. Mass concentration measurements are typically reported as $PM_1$, $PM_{2.5}$, $PM_{10}$, or TPM, referring to the size ranges below 1, 2.5, and 10 µm, and total mass, respectively. For convenience, data reported as $PM_1$ and $PM_{2.5}$ have been grouped together in the $PM_{2.5}$ category, which in view of
the typical BB aerosol size distribution is not expected to result in significant bias. The same applies to the $PM_{10}$ and TPM data, which were grouped together in the TPM category.

Emission data for ionic species and trace metals are not included in this data set. They are tabulated in Akagi et al. (2011), and additional information can be found in a number of papers (e.g., Goetz et al., 2018; Jayarathne et al., 2018a; Jayarathne et al., 2018b).
Another problematic "species" is the total concentration of non-methane organic gases (NMOG), also referred to as volatile organic compounds (VOCs). The diverse methods used for these compounds measure different sets of NMOG, which in some instances may quite incomplete. In general, the more recent studies from the last 5-7 years are much more comprehensive and show that the early studies were severely underestimating the amounts of NMOG emitted. Regrettably, these techniques have been so far used mostly in lab studies, and could therefore not be considered for the combustion category
emission estimates. To highlight this issue, I have added the NMOG emission factors from the online updates to Akagi et al. (2011) in Tables 1 and 3.

## 2.2 Definitions

In the literature, emission information is generally found as either emission ratios (ER) or emission factors (EF). Strictly speaking, most data presented as "emission ratios" are actually enhancement ratios (EnR), often also referred to as
normalized excess mixing ratios (NEMR; Akagi et al., 2011). They are defined as the ratio of the excess mixing ratio of the species of interest in the plume, ($\Delta X$), to the excess mixing ratio of a reference species, e.g., carbon monoxide ($\Delta CO$),

$$EnR_{X/CO} = \frac{\Delta X}{\Delta CO} = \frac{(X)_{plume} - (X)_{backgr}}{(CO)_{plume} - (CO)_{backgr}}$$

where Δ stands for the difference between the mixing ratio in the plume and in the background atmosphere (in molar units). Because of its abundance in fire emissions and its relatively low ambient background concentration, CO is most commonly used as reference species, but other gases, such as carbon dioxide ($CO_2$), methane ($CH_4$), or acetonitrile have also been used. The use of $CO_2$ can introduce large errors because it also has strong surface sources and sinks, which can lead to erroneous estimates of the background concentration, as discussed in detail in Yokelson et al. (2013a). A statistical method using multiple fire tracers (Mixed Effects Regression Emission Technique, MERET), which can resolve the problems associated with variable $CO_2$ background concentrations, has recently been developed (Chatfield et al., 2019).

An enhancement ratio can be interpreted as an emission ratio when it is assured that the concentrations of both species X and the reference species have not been affected by chemical production or loss since the emission, and that both concentrations have changed proportionally during dilution of the plume with background air. In the case of very long-lived substances, e.g., acetonitrile, EnRs can be very close to ERs even after days, while for reactive compounds, e.g., nitric oxide (NO), significant changes can occur in minutes. For very rapidly reacting species, it becomes difficult to define an appropriate time after emission at which an EnR can be treated as an effective ER. A good example is the emission of primary organic aerosol mass, where the apparent EnR decreases substantially (by about a factor of two) over the first few minutes to hours as a result of the evaporation of semivolatile compounds during plume dilution (May et al., 2013; Konovalov et al., 2019). Whether the ER at the moment of emission or the EnR after cooling and dilution to typical ambient conditions is the more meaningful value will depend on the intended application. In general, field measurements are likely to represent somewhat more aged conditions (tens of minutes to a few hours), whereas lab measurements often represent very fresh emissions. For further discussion of the advantages and disadvantages of the different reference gases, the effects of flaming vs. smoldering combustion, and ground-based vs. airborne sampling, see A&M2001, Burling et al. (2011), and Akagi et al. (2011).

While the measurement of ERs is relatively straightforward in the field, because it requires only the measurement of the atmospheric concentrations of target and reference species, it is generally desirable to obtain the amount of a species emitted per unit mass of fuel burnt, i.e., the emission factor, EF. For biomass burning, this is usually expressed as the mass of target species X released per mass of dry fuel burnt, in units of g $kg^{-1}$. This, however, requires knowledge of the mass of fuel burned, which can be measured in the lab, but difficult to obtain in the field. As an alternative, the mass balance method can be used, where the mass of fuel burned is approximated by the sum of carbon contained in the emitted carbon species ($CO_2$, CO, $CH_4$, volatile organic compounds [VOC], organic aerosol carbon [OC], and elemental carbon [EC] or black carbon [BC]), divided by the carbon fraction in the fuel. Often, the carbon mass is approximated by the sum of $CO_2$ and CO, and a default fuel carbon content of 45% is assumed.

To provide a uniform representation of the various types of data found in the literature in the form most useful to modelers, all emission data was converted to emission factors, in units of g $kg^{-1}$ dry fuel burnt. Where emission factors relative to other fuel mass indicators were given, e.g., the mass of carbon burned or released, I applied appropriate conversion factors, such as the known or assumed carbon content of the fuel. Very frequently in the literature, only EnRs or ERs in units of mol/mol are provided. These can in principle be easily converted to EFs by the following equation:

$$EF_X = ER_{(X/Y)} \frac{MW_X}{MW_Y} EF_Y$$

where $EF_X$ is the emission factor for species X, $ER_{(X/Y)}$ is the emission ratio of species X relative to the reference species Y, $MW_X$ and $MW_Y$ are the molecular weights of the species X and the reference species Y, and $EF_Y$ is the known or assumed emission factor of the reference species (often CO or $CH_4$). When the value of $EF_Y$ was not known for a specific study, the mean $EF_Y$ for the appropriate type of fire (forest, savanna, etc.) was applied to derive an estimate of $EF_X$.

### 2.3 Estimates where no data are available

For some combinations of fire type and emitted species, no suitable field data is available to provide a basis for estimating EFs. Where possible, I have used appropriate methods to derive estimates (shown in italic font in Table 1) based on other information. For each species, the estimation method is given in column EM. For species predominantly emitted during smoldering combustion, e.g., most VOCs, I have based the estimate on the assumption that their emission factors for the various fire categories are proportional to those of CO for the same categories. The estimate was then obtained by calculating the mean of the ratios $EF_X/EF_{CO}$ for the fire categories with available data and multiplying this mean ratio by the $EF_{CO}$ of the fire category for which an estimate was needed (labeled CO in column EM). Where no suitable ratios $ER_X/ER_{CO}$ were available from field studies, the lab ratio was used instead (labeled LV). For some species containing heteroelements ($N_2O$, $SO_2$, DMS, and HCl), the mean of the ERs from fire categories with available data, weighted by the amounts of biomass globally burned in those categories, was used (labeled AV). Subjective "best estimates" are labeled BE. Specifically, for missing values of total particulate carbon emissions, the sum of OC and EC emissions was used, and for aerosol potassium emissions in boreal forest fires I used the temperate forest value.

### 3 Results and discussion

#### 3.1 Combustion process and pyrogenic emissions

Our fundamental understanding of the biomass combustion process remains unchanged since the 1990s, as reviewed in A&M2001 and other papers (Lobert and Warnatz, 1993; Yokelson et al., 1996; Yokelson et al., 1997; Akagi et al., 2011), and will thus be summarized here only briefly. As the flaming or glowing front of a fire moves towards the uncombusted fuel, the fuel is heated by radiative and sensible heat transfer, leading first to evaporation of water and other volatiles, then to pyrolytic decomposition and the release of volatile and semivolatile (tar) decomposition products (Collard and Blin, 2014). When this released mixture ignites, flame chemistry sets in, which breaks down the more complex pyrolysis compounds to small molecules and radicals, but also produces new larger molecules by radical chemistry, such as alkynes, polycyclic aromatic hydrocarbons (PAH), soot, and organohalides. In addition to volatile matter being consumed by flaming combustion, char undergoes gas-solid reactions between oxygen and other gases and solid carbon at the fuel surface, called gasification or "glowing" combustion, in which a large fraction of the fuel carbon is released as CO, part of which is further oxidized to $CO_2$.

In a typical vegetation fire, all these processes occur simultaneously as the fire propagates through the fuel, so that the fire plumes at any place and time contain mixtures of flaming and smoldering (vernacular for a changing mix of distillation, pyrolysis, and glowing) combustion products in variable proportions.

Depending on the vegetation type and burning conditions, the relative amounts of fuel consumed by flaming and smoldering combustion can vary considerably. Dry grassland fires, for example, are dominated by flaming combustion and a rapid passage of the fire front, with little residual smoldering. Forest fires, on the other hand, especially those in fuels with relatively high fuel moisture and large diameters, have a long phase of residual smoldering combustion (RSC), during which larger-diameter fuels are consumed over time spans of up to several days (Ward and Hardy, 1991; Ward et al., 1992; Yokelson et al., 1997; Bertschi et al., 2003; Hao and Babbitt, 2007; Burling et al., 2011; Akagi et al., 2013; Geron and Hays, 2013; Urbanski, 2014; Reisen et al., 2018). The smoldering mode of combustion can become dominant in peat fires, which often proceed without a flaming phase and below ground (Bertschi et al., 2003; Stockwell et al., 2016b).

Since the rate of heat release during RSC is relatively low and much of it occurs during nighttime, the resulting emissions tend to accumulate close to ground in the boundary layer. At nighttime, emissions are confined in a nocturnal boundary layer, often less than 100 m thick, where the fire-emitted $CO_2$ becomes mixed with $CO_2$ from biological respiration. This presents serious problems for measuring accurate and representative fire-integrated emission factors for fires where RSC emissions are important (Bertschi et al., 2003). Ground-based studies during the RSC phase can obtain EFs of trace species, but these are difficult to relate to the corresponding amount of fuel burned. Aircraft studies have trouble measuring the RSC component of these emissions, as they are not lofted in the form of discrete plumes to aircraft altitudes, but only mixed upward during daytime convection (or fire blow-ups) where they get distorted by mixing in the ambient atmosphere (Guyon et al., 2005). The mixing of biogenic and pyrogenic $CO_2$ in fire plumes that entrain such boundary layer air into a deeper mixed layer present serious problems for deriving fire-integrated ERs and EFs from aircraft measurements (Yokelson et al., 2013a), which can potentially be addressed by the multi-tracer MERET approach (Chatfield et al., 2019)..

Because the flaming phase is characterized by $CO_2$ being the dominant combustion product by far, while the smoldering phase yields relatively large amounts of CO (up to about 30% of carbon burned), the MCE has been established over the last two decades as the key metric representing the relative role of flaming vs. smoldering combustion in vegetation fires, spanning a range of 0.77 in peat fires to 0.98 in some grassland fires (see Supplement). Mean MCE values for the different combustion categories are presented in Table 1.

Since the MCE was introduced by Ward and Radke (1993), numerous papers have used this metric and have shown significant negative correlations for many trace gases between emission factors and MCE, especially for the various VOCs that are emitted predominantly during smoldering combustion (e.g., Korontzi et al., 2003; Yokelson et al., 2003; Yokelson et al., 2008; Soares Neto et al., 2009; Urbanski et al., 2009; Burling et al., 2011; Urbanski, 2013; Yokelson et al., 2013b; Liu et al., 2014; Urbanski, 2014; Collier et al., 2016; Coffey et al., 2017; Fortner et al., 2018; Hodgson et al., 2018; Reisen et al., 2018; Jen et al., 2019). However, the correlation slopes between EFs and MCE vary considerably between studies in different fuels and burning environments, so that a global parameterization of all EFs based on observed or modeled MCE remains

problematic. As an illustration, I show in Fig. 1a and 1b plots of the EFs of ethene ($C_2H_4$) and ethane ($C_2H_6$) vs MCE, based on the average values from the individual studies in the supplemental spreadsheet. In both cases, the results scatter widely, and especially the data from the lab studies, biofuel burning, peat fires, and RSC-dominated fires introduce a large amount of scatter. The limitations in correlation between EFs and MCE have been noted previously (Yokelson et al., 1997; Bertschi et al., 2003; Burling et al., 2011; Urbanski, 2014). In the case of ethene, the correlation using all data points is not significant ($R^2$ = 0.07). However, when only the data from open vegetation fires are included (and after removing three outliers), the correlation improves to an $R^2$ of 0.27. For ethane, the correlation coefficient is $R^2$ = 0.38 for all data, but does not improve substantially by removing the peat fire data. These results suggest that the level of aggregation at which MCE is useful as a meaningful, but rough predictor of EFs for at least some species is yet to be determined. This approach is not pursued further here, but the data in the original studies listed in the supplement can be used by investigators to derive such relationships for specific compounds and combustion types of interest. An interesting and novel approach to generalizing VOC emissions is provided by Sekimoto et al. (2018), who showed that most of the variability in VOC emissions measured in a lab study using a wide variety of fuels was explained by just two factors, related to low and high temperature pyrolysis, respectively.

Using MCE as a predictor variable may be an alternative to providing separate EFs for smoldering and flaming combustion, which has been frequently requested by the modeling community, but for which there is still not enough data to provide robust estimates, as we had already remarked previously in A&M2001. However, once vegetation fire models are able to provide estimates of the contribution of flaming and smoldering combustion from a given fire, the resulting MCE could be predicted. This could then form the basis of a more fire-specific prediction of trace gas and aerosol emissions based on MCE correlations. An alternative approach was proposed by Hoffa et al. (1999) and further developed by Korontzi et al. (2003), who showed a correlation between vegetation greenness and MCE, which allowed the prediction of seasonally-dependent emissions from African savanna fires (Ito and Penner, 2004; Korontzi et al., 2004; Korontzi, 2005). In view of the limitations seen with regard to more general parametrizations, it appears that for now one can keep using the category-average EFs, but be aware they can vary considerably from region to region and from fire to fire.

**3.2 Emission factors for chemical species from the various combustion categories**

In Table 1, I present the updated estimates of emission factors for the combustion categories, savanna/grassland, tropical forest, temperate forest, boreal forest, peat fires, open agricultural waste burning (in the fields), biofuels (excluding dung), dung cakes, charcoal making, charcoal burning, and garbage burning. As more data have become available, it was now possible to split the extratropical forest category into temperate and boreal forest burning. The transition between these two types is not always clear, but in general, I have followed the authors' choice of category; where this was not possible I have taken a latitude of 60 ºN as boundary.

The large number of studies on residential biomass burning, which have been published in the last two decades, has made it possible to separate dung cakes from the other biofuels, such as fuel wood and agricultural residues. As mentioned

above, I only included studies that used fireplaces and traditional or simple "improved" stoves, as are used in developing countries, and not modern appliances, such as automated pellet stoves.

The publication of a few papers that provide emissions data for open garbage burning, still quite prevalent in many countries and a serious source of pollution especially in urban areas (Wiedinmyer et al., 2014), has made it possible to provide
EFs for this category.

Obviously, the categories used here are still quite highly aggregated, but they correspond closely to the fire types used in many global modeling studies, such as those involved in the Fire Modeling Intercomparison Project (FireMIP) (Li et al., 2019a) and in model- or satellite-based emission inventories (Wiedinmyer et al., 2011; Kaiser et al., 2012; Ichoku and Ellison, 2014; Darmenov and da Silva, 2015; van der Werf et al., 2017). Should a reader require less highly aggregated data, they can
use the Supplement to split the data into subcategories or even use the supplemental references to get back to the original literature. Valuable detail about the various burning types and further breakdown of some categories, e.g., biofuel use, into relevant subcategories can be found in Akagi et al. (2011).

For information purposes, I also include a column summarizing the results of laboratory studies. The averages in this column can only be seen as general indication of the magnitude of emission factors found in the lab studies, since all types of
fuels and burning methods are included in the statistics presented here. However, the original data and references are provided in the Supplement for readers interested in the details.

As in A&M2001 and in Akagi et al. (2011), the amount of information for any given combination of species and fire category varies greatly - for some combinations we have no measurements at all and for others there are as many as 50 values. Accordingly, the uncertainty of the estimates is also highly variable. In Table 1, I am using the same convention as in
A&M2001 to represent the uncertainty: When three or more values (based on independent references) are available for a given table cell, the results are given as means and standard deviations (x $\pm$ s). In the case of two available measurements, they are given as a range, and where only a single measurement is available, it is given without an uncertainty estimate. For single measurements, it can usually be assumed that the uncertainty is no less than a factor of three.

In spite of the fact that this paper is based on data from over 370 publications, rather than the 130 papers that formed
the basis for A&M2001, Table 1 shows that there are still many species for which there are little or no field data available. For example, there are still no field measurements of the emission factors for the alkyl amines, which have recently become implicated in aerosol nucleation and new particle formation (Smith et al., 2010; Almeida et al., 2013; Kürten et al., 2014). In view of the importance of the number concentrations of aerosol particles (CN), especially cloud condensation nuclei (CCN), for climate change, it is unfortunate that there have only been a few additional measurements of their emission factors in the
last two decades. The rapid coagulation of particles very near the source makes it difficult to choose the most appropriate plume age for such a measurement (Hobbs et al., 2003; Sakamoto et al., 2016; Hodshire et al., 2019). However, a survey of available measurements suggests that the ratio of excess particle number concentration ($\Delta$CN or $\Delta$CCN) to $\Delta$CO stabilizes at the scale of typical aircraft measurements in plumes, as a consequence of the sharp decrease of the coagulation rate with

increasing dilution (Janhäll et al., 2010). More field studies on the evolution of aerosol number concentrations and size distributions as a function of plume age under different conditions (fire size, wind speed, flux density, etc.) are warranted.

Another climate-relevant component, for which we have no field emission data at this time, is brown carbon (BrC) (Andreae and Gelencsér, 2006), which has been shown to account for about half of the aerosol light absorption by biomass smoke at 401 nm (Selimovic et al., 2019) and 25-45 % at 550 nm (Tian et al., 2019). Providing EFs for this species is problematic because of the very complex and variable mixture of compounds that make up BrC, as well as its potential for rapid change in abundance and optical properties during plume evolution (Forrister et al., 2015; Fleming et al., 2019). To some extent, data on the optical properties of BB aerosols can substitute for direct measurements of BrC (Stockwell et al., 2016a; Stockwell et al., 2016b; Goetz et al., 2018; Selimovic et al., 2018).

Regarding the role of vegetation fires in the global carbon cycle, the most problematic uncertainty pertains to the emission factors of $CO_2$ and CO from forest fires, which is surprising in view of the many available estimates. This uncertainty stems from the inadequate knowledge of the contribution from RSC, which has already been referred to above, and which may significantly contribute to large mismatches between bottom-up predictions of CO emissions and remote-sensing measurements from satellite (Pechony et al., 2013; Deeter et al., 2016). A representative measurement of fire-average $\Delta CO/\Delta CO_2$ emission ratios from large forest fires is very difficult if not impossible, as ground-based measurements in such violent fires are not possible and aircraft measurements are prone to undersampling the smoldering emissions, especially the contributions from RSC. The uncertainty regarding the $\Delta CO/\Delta CO_2$ emission ratio also seriously hampers our ability to separate the influence of the emissions from deforestation burning from those of biological carbon fluxes in regional carbon budgets (Andreae et al., 2012). For example, the uncertainty of the $\Delta CO/\Delta CO_2$ ratios of tropical forest burning is large enough that it can even change the inferred sign of the net carbon flux between the Amazon forest and the atmosphere (Gatti et al., 2014). A novel multi-tracer statistical technique (MERET; Chatfield et al., 2019) may be able to provide improved estimates of the CO ERs and EFs from such fires.

Figure 2 presents a comparison between selected EFs from this study with those published in Akagi et al. (2011) in the form of ratios between the EFs from these studies. For this comparison, I have selected species that are of major climatic or chemical significance or are important BB tracers, and for which there are enough data to allow a meaningful comparison. Data are presented for the combustion types with the largest total global emissions, i.e., savanna/grassland, tropical, temperate, and boreal fires, and biofuel use. In the case of biofuel use, the comparison is made with Akagi et al.'s "open cooking" category, because its MCE shows good agreement with that for the "biofuel use" category in this paper. Figure 2 shows close agreement for the main carbon species $CO_2$ and CO as well as for MCE, suggesting that both species capture comparable combustion conditions. For most other species, the EF ratios fall within a factor of two, with no obvious systematic shift for either the individual species or for the combustion types. A slight exception are the EFs for savanna/grassland, which tend to be somewhat higher in the present study. In one case (isoprene) this is the result of higher values from an individual study, i.e., the lab-adjusted-to-field EFs from Stockwell et al. (2015), but generally the differences appear to be the result of including a larger set of studies from this category in the present study. The lower EFs for glycolaldehyde in this study are the result of corrections

made by the Yokelson group to their data based on improved spectral data (see https://www.atmos-chem-phys-discuss.net/12/C11864/2013/acpd-12-C11864-2013.pdf), which have been incorporated here and in the online updates to Akagi et al. (2011), but for consistency the values from the original paper were used for Fig. 2. The largest and most systematic difference is seen for the NMOG category, where the values from Akagi et al. (2011) are as much as a factor of 10 higher than the averages from the published field studies in Table 1. This is largely due to differences in the analytical techniques used in the original studies. Most of the older studies, especially in field campaigns, were measuring only a very limited subset of NMOG (e.g., non-methane hydrocarbons), whereas Akagi et al. in the original paper and in the subsequent updates used techniques that measured practically all NMOG, including unidentified species. To address this issue, I am including both the field study averages (labeled VOCs) and the corresponding values from the online updates to Akagi et al. (2011) (labeled NMOGs) in Table 1. The latter values may be more appropriate as input for modeling studies that require an estimate of total NMOGs.

### 3.3 Emissions from global biomass burning

In 2001, we estimated the total amount of biomass burned by all combustion types to be 8.6 Pg dry matter annually, with an uncertainty of ±50% (A&M2001). This estimate was based on bottom-up inventories and had not yet benefitted from remote-sensing detection and quantification of fires. At present, there are several operational fire detection and emission estimation products based on remote sensing. Three of them (for example) use an approach based on burnt area and hotspot detection: Fire INventory from NCAR (FINN; Wiedinmyer et al., 2011), Fire Locating and Modeling of Burning Emissions (FLAMBE; Reid et al., 2009), and Global Fire Emissions Database (GFED; van der Werf et al., 2017). The other three products are based on fire radiative power (FRP): Quick Fire Emission Dataset (QFED; Darmenov and da Silva, 2015), Global Fire Assimilation System (GFAS; Kaiser et al., 2012), and Fire Energetics and Emissions Research (FEER; Ichoku and Ellison, 2014). The amounts of biomass burned annually in open fires estimated by these systems still spans a wide range, from 4.3 Pg (GFAS) to 11.6 Pg (FLAMBE) (for the FRP-based products, which do not use biomass burnt in their calculations, the biomass estimate was based on the stated emission of carbon compounds and an assumed carbon fraction of 45 % in the biomass).

For domestic biofuel use, there are three recent global estimates: 2.1 Pg a$^{-1}$ (Fernandes et al., 2007), 2.5 Pg a$^{-1}$ [S. J. Smith, personal communication, 2019, based on the Community Emissions Data System (CEDS) model (Hoesly et al., 2018)] and 2.3 Pg a$^{-1}$ [Z. Klimont, personal communication, 2019, based on the methodology in Klimont et al. (2017)]. These recent estimates are all somewhat lower than those of A&M2001 (2.9 Tg a$^{-1}$) and Yevich and Logan (2003) (3.1 Tg a$^{-1}$). For charcoal burning, I am also using the estimate of 53 Tg a$^{-1}$ given for 2014 by FAO (2015), and for charcoal making I am assuming a 25% yield of charcoal relative to dry wood (Yevich and Logan, 2003).

Combining these estimates of open and domestic burning yields a mean estimate of 8.8 Pg (with a range of 6.4 to 14.1 Pg) dry biomass burned annually. Interestingly, this is almost identical to the values given in A&M2001: 8.6 Pg a$^{-1}$, with an estimated range of 4.3 to 12.9 Pg a$^{-1}$. Table 2 summarizes these emission estimates. For the various categories of open burning, the satellite-derived emission estimates vary greatly, in some cases by an order of magnitude. Differences in the

definitions of the burning categories between the different retrieval algorithms, differing ability to detect small fires, and the fundamental difference between the burnt-area and FRP-based techniques may all play a role here.

In Table 3, I use the average of the available estimates from the different inventories shown in Table 2 as activity estimates for the combustion categories to derive emission values for major species emitted from biomass burning. For comparison, the last column in Table 3 shows the global total emissions estimated in A&M2001. The totals of the major emitted carbon species and many minor species remain fairly close to those in our previous assessment. Given the large number of measurements for the emission factors for the major species, $CO_2$, $CO$, and $CH_4$, the standard error of the mean is much smaller than the standard deviation, and thus the relative uncertainties of the mean for these emission factors are quite small, 1-3% for $CO_2$, 4-9% for $CO$, and 6-18% for $CH_4$ from the major burning categories savanna, forests, and biofuel. Consequently, the global emission uncertainties for these species are completely dominated by the large uncertainties in the activity estimates.

The best independent "reality check" for these emissions may still come from the inverse modeling of the CO budget. This species is the most appropriate for such a comparison, because its emission factors are well constrained, biomass burning is a large fraction of all global sources, and there is a large body of measurements both from ground stations and remote sensing. Estimates of CO emissions from the various inversion models range from 190 to 560 Tg $a^{-1}$ from biofuel burning and 360 to 610 Tg $a^{-1}$ from open burning for the years around 2000 (Park et al., 2015, and references therein). The model of Park et al. (2015), which uses a joint inversion of CO concentrations and oxygen isotopic composition and therefore is likely to be the most reliable in separating the different source types, predicts CO emissions of 380 to 610 Tg $a^{-1}$ from open burning, 400 to 520 Tg $a^{-1}$ from biofuel use, and 780 to 1130 Tg $a^{-1}$ for all biomass burning. Using the EFs from Table 1 and the activity estimates from Table 2, we obtain a range of 390 to 1210 Tg $a^{-1}$ for the CO emissions from open burning, in reasonable agreement with the inverse results. The range of biofuel CO emissions estimated from Tables 1 and 2 is only 181-196 Tg $a^{-1}$, accounting for less than one-half of the inverse estimate. This suggests either that the amount of biofuel use is significantly underestimated in present bottom-up budgets, or that the inversions attribute some of the open burning inaccurately to biofuel use. This could likely be the case for agricultural burning, which uses similar fuels and takes place in similar regions as biofuel use. The inverse analyses may also be useful to indicate unlikely estimates based on remote-sensing techniques. For example, the burning of 8750 Tg dm in tropical forests estimated by FLAMBE, combined with the corresponding $EF_{CO}$ (105 g $kg^{-1}$) would produce CO emissions of 900 Tg $a^{-1}$ from this biome alone, well above the range of inverse CO emission estimates for all open burning (see also the comments by Reviewer 1, https://doi.org/10.5194/acp-2019-303-RC1).

Major differences between the present emission estimates and A&M2001 are seen for the oxygenated volatile organic compounds and for HCN (as already noted in Akagi et al., 2011), which all are significantly greater in the present assessment than in A&M2001. This is due to the large number of new and more accurate emission factor measurements for these compounds, which have been made possible by improvements in analytical techniques since the 1990s.

## 4 Conclusions

We are left with the somewhat frustrating conclusion that, in spite of the great progress in emission factor measurements and detection and quantification of fires, the overall uncertainty of global biomass burning emissions has not decreased significantly for most substances since our analysis of almost twenty years ago. Evidently, there is a great need for improved accuracy in the activity estimates, both for open burning and especially for biofuel use. For open burning, coordinated regional CO studies in regions and at times of high biomass burning activity, including both FRP and burnt-area based remote sensing approaches as well as inversions, may be a way to resolve discrepancies and improve accuracy. This would be of great benefit for testing and improving fire emission models, which also give quite divergent results and have difficulties in capturing interannual variations and temporal trends. For example, the modelled estimates of carbon emitted from open burning in the nine models participating in the FireMIP project spans from 1.0 to 4.9 Pg $a^{-1}$ (Li et al., 2019a).

With regard to emission factors, Table 1 can serve as a guide to prioritizing future research activities. Photochemically active species and toxic compounds for which there are only a few measurements from important fire types deserve more intense study. An example is the emission of PAHs, where we have only one study from boreal fires and none at all from tropical forest fires. Given the toxicity of these compounds and the increasing exposure of populations in these regions to biomass smoke as a result of climate change and population growth, this seems an important knowledge gap. Another example are the emissions of semivolatile and intermediate-volatile compounds (I/SVOCs), which are important in the context of organic aerosol from biomass burning, but for which at this time only laboratory measurements are available (Hatch et al., 2018). I have already referred to the lack of field measurements of alkyl amine emissions, which may be of importance for new particle formation. In view of the grave health risk associated with aerosol particles (see, e.g., Lelieveld et al., 2019, and references therein) and the growing exposure to wildfire smoke in areas like the western U.S.A., the accuracy and fire condition dependence of PM emissions need to be improved. Emphasis should be on field measurements under a variety of representative conditions, to represent the influence of parameters like fuel moisture and fire weather. While the approach in this paper is focused on global averages, future work should also emphasize regional and seasonal differences in order to better support more highly geographically resolved modeling.

A spreadsheet containing Table 1, the data on which the averages in Table 1 are based, and the corresponding references is available at http://dx.doi.org/10.17617/3.26 . This spreadsheet will be updated periodically.

## 5 Acknowledgments

I thank Johannes Kaiser, Imke Hüser, Zbigniew Klimont, Anton Darmenov, Edward Hyer, and Steven Smith for providing estimates of biomass burned from their respective databases M. Desserrvetaz for unpublished emission data, and R. Yokelson, C. Ichoku, N. Surawski, J. Roberts, and an anonymous reviewer for comments on the manuscript. This work was supported by the German Max Planck Society.

**Figure Captions**

Figure 1: Scatter plots of the emission factors of ethene (a) and ethane (b) against MCE, based on studies in the different combustion categories.

5   Figure 2: Comparison between the emission factors for selected species between this study and the values in Akagi et el., (2011).

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

Table 1: Emission factors for pyrogenic species emitted from various types of biomass burn[a]

| Species | Savanna and grassland | | | Tropical forest | | | Temperate forest | | | Boreal forest | | | Peat Fires | | | Agricultural residues (open) | | |
|---|---|---|---|---|---|---|---|---|---|---|---|---|---|---|---|---|---|---|
| | average | std.dev. | N | average | std.dev. | N | average | std.dev. | N | average | std.dev. | N | average | std.dev. | N | average | std.dev. | N |
| MCE | 0.94 | 0.02 | 49 | 0.91 | 0.03 | 16 | 0.90 | 0.05 | 45 | 0.89 | 0.04 | 21 | 0.80 | 0.02 | 6 | 0.92 | 0.06 | 36 |
| $CO_2$ | 1660 | 90 | 31 | 1620 | 70 | 9 | 1570 | 130 | 39 | 1530 | 140 | 14 | 1590 | 150 | 6 | 1430 | 230 | 29 |
| CO | 69 | 20 | 50 | 104 | 39 | 16 | 113 | 50 | 47 | 121 | 47 | 22 | 260 | 23 | 6 | 76 | 55 | 39 |
| $CH_4$ | 2.7 | 2.2 | 49 | 6.5 | 1.6 | 13 | 5.2 | 2.8 | 37 | 5.5 | 2.5 | 20 | 9.1 | 1.5 | 6 | 5.7 | 6.0 | 20 |
| VOC[c] | 5.1 | 5.9 | 14 | 5.6 | 1.5 | 4 | 13.4 | 11.8 | 13 | 6.0 | 2.9 | 8 | *21* | - | *0* | 7.6 | 8.0 | 12 |
| Total NMOG, including unidentifie[d] | 30 | - | - | 52 | - | - | 39 | 18 | - | 59 | - | - | 136 | - | 0 | 51 | - | - |
| $C_2H_2$ | 0.31 | 0.29 | 29 | 0.35 | 0.39 | 6 | 0.31 | 0.09 | 21 | 0.28 | 0.13 | 12 | 0.11 | 0.05 | 3 | 0.27 | 0.24 | 11 |
| $C_2H_4$ | 0.83 | 0.38 | 26 | 1.11 | 0.24 | 5 | 1.11 | 0.29 | 21 | 1.54 | 0.66 | 7 | 1.47 | 0.72 | 3 | 1.00 | 0.49 | 14 |
| $C_2H_6$ | 0.42 | 0.32 | 29 | 0.88 | 0.23 | 7 | 0.69 | 0.56 | 21 | 0.97 | 0.37 | 14 | 1.85 | 1.5-2.2 | 2 | 0.79 | 0.62 | 11 |
| $C_3H_4$ | 0.071 | 0.111 | 8 | 0.013 | - | 1 | 0.05 | 0.02 | 7 | 0.062 | 0.031 | 3 | 0.006 | - | 1 | 0.18 | 0.01-0.34 | 2 |
| $C_3H_6$ | 0.46 | 0.45 | 26 | 0.86 | 0.41 | 5 | 0.60 | 0.40 | 20 | 0.67 | 0.45 | 7 | 1.14 | 1.07-1.21 | 2 | 0.47 | 0.36 | 16 |
| $C_3H_8$ | 0.13 | 0.18 | 20 | 0.53 | 0.15-0.91 | 2 | 0.28 | 0.18 | 15 | 0.29 | 0.10 | 8 | 0.99 | - | 1 | 0.17 | 0.07 | 9 |
| 1-Butene | 0.082 | 0.049 | 13 | 0.073 | 0.020-0.125 | 2 | 0.12 | 0.061 | 9 | 0.16 | 0.143 | 4 | 0.46 | 0.18-0.74 | 2 | 0.083 | 0.043 | 8 |
| i-Butene | 0.041 | 0.019 | 6 | 0.109 | - | 1 | 0.086 | 0.074 | 9 | 0.052 | 0.032 | 3 | 0.31 | - | 1 | 0.079 | 0.040 | 3 |
| trans-2-Butene | 0.020 | 0.012 | 11 | 0.033 | 0.016-0.050 | 2 | 0.037 | 0.031 | 9 | 0.030 | 0.018 | 3 | 0.078 | - | 1 | 0.036 | 0.014 | 6 |
| cis-2-Butene | 0.017 | 0.010 | 11 | 0.031 | 0.020-0.042 | 2 | 0.038 | 0.039 | 9 | 0.023 | 0.016 | 3 | 0.062 | - | 1 | 0.027 | 0.010 | 6 |
| Butadiene | 0.095 | 0.057 | 13 | *0.15* | - | *0* | 0.125 | 0.068 | 12 | 0.089 | 0.030 | 4 | 0.22 | 0.19-0.26 | 2 | 0.16 | 0.24 | 10 |
| n-Butane | 0.021 | 0.011 | 14 | 0.041 | - | 1 | 0.080 | 0.057 | 12 | 0.111 | 0.059 | 7 | 0.32 | - | 1 | 0.043 | 0.029 | 7 |
| i-Butane | 0.007 | 0.005 | 13 | 0.015 | - | 1 | 0.031 | 0.026 | 11 | 0.052 | 0.051 | 6 | 0.090 | - | 1 | 0.016 | 0.017 | 7 |
| 1-Pentene | 0.022 | 0.009 | 6 | 0.058 | - | 1 | 0.048 | 0.024 | 7 | 0.046 | 0.025 | 3 | 0.110 | - | 1 | 0.015 | 0.011 | 5 |
| 2-Pentenes | 0.014 | 0.020 | 4 | *0.026* | - | *0* | 0.043 | 0.023 | 5 | 0.011 | 0.006-0.016 | 2 | 0.062 | - | 1 | 0.023 | 0.005 | 4 |
| n-Pentane | 0.007 | 0.008 | 11 | 0.014 | - | 1 | 0.034 | 0.026 | 10 | 0.050 | 0.015 | 6 | 0.24 | - | 1 | 0.042 | 0.057 | 7 |
| Methyl-butenes | 0.025 | 0.037 | 7 | 0.075 | - | 1 | 0.056 | 0.045 | 6 | *0.051* | - | *0* | 0.125 | - | 1 | 0.026 | 0.012 | 5 |
| 2-Methyl-butane | 0.008 | 0.009 | 10 | 0.008 | - | 1 | 0.017 | 0.011 | 8 | 0.032 | 0.016 | 6 | 0.123 | - | 1 | 0.019 | 0.014 | 5 |
| n-Pentadienes | 0.048 | - | 1 | *0.042* | - | *0* | 0.035 | 0.016 | 4 | *0.049* | - | *0* | *0.10* | - | *0* | *0.031* | - | *0* |
| Isoprene | 0.101 | 0.158 | 10 | 0.22 | 0.016-0.42 | 2 | 0.10 | 0.05 | 9 | 0.074 | - | 1 | 0.52 | 0.05-0.98 | 2 | 0.17 | 0.26 | 7 |
| Cyclopentene | 0.019 | 0.016 | 4 | *0.022* | - | *0* | 0.041 | 0.019 | 5 | *0.03* | - | *0* | 0.025 | - | 1 | 0.007 | 0.002 | 3 |
| Cyclopentadiene | 0.026 | - | 1 | *0.035* | - | *0* | 0.027 | 0.025-0.029 | 2 | *0.041* | - | *0* | 0.010 | - | 1 | 0.001 | - | 1 |
| 4-Methyl-1-pentene | 0.049 | - | 1 | 0.049 | - | 1 | *0.040* | - | *0* | *0.043* | - | *0* | 0.09 | - | *0* | 0.004 | 0.005 | 4 |
| 2--Methyl-1-pentene | 0.018 | 0.032 | 4 | *0.037* | - | *0* | 0.058 | 0.027 | 3 | *0.043* | - | *0* | 0.11 | - | 1 | *0.027* | - | *0* |
| 1-Hexene | 0.043 | 0.018 | 6 | 0.065 | - | 1 | 0.084 | 0.022 | 3 | 0.109 | - | 1 | *0.14* | - | *0* | 0.011 | 0.005 | 3 |
| Hexadienes | 0.006 | - | 1 | *0.007* | - | *0* | 0.006 | 0.006-0.006 | 2 | *0.009* | - | *0* | *0.018* | - | *0* | *0.005* | - | *0* |
| n-Hexane | 0.018 | 0.028 | 10 | *0.032* | - | *0* | 0.032 | 0.040 | 10 | 0.054 | 0.035 | 3 | 0.14 | - | 1 | 0.032 | 0.059 | 4 |
| Isohexanes | 0.019 | 0.028 | 3 | *0.048* | - | *0* | 0.026 | 0.038 | 8 | 0.013 | 0.008-0.018 | 2 | 0.054 | - | 1 | 0.067 | 0.115 | 4 |
| Heptanes | 0.016 | 0.019 | 6 | *0.024* | - | *0* | 0.029 | 0.026 | 8 | 0.021 | 0.018-0.024 | 2 | 0.112 | - | 1 | 0.031 | 0.033 | 4 |
| Octenes | 0.021 | 0.027 | 3 | 0.012 | - | 1 | 0.036 | 0.023 | 5 | *0.021* | - | *0* | 0.065 | - | 1 | 0.002 | - | 1 |
| Terpenes | 0.104 | 0.096 | 5 | *0.15* | - | *0* | 1.17 | 1.95 | 9 | 1.53 | - | 1 | 0.08 | 0.005-0.16 | 2 | 0.027 | 0.026 | 4 |
| Benzene | 0.33 | 0.22 | 19 | 0.38 | 0.05 | 4 | 0.42 | 0.17 | 17 | 0.57 | 0.21 | 7 | 0.87 | 0.78-0.95 | 2 | 0.27 | 0.19 | 17 |
| Toluene | 0.19 | 0.14 | 17 | 0.23 | 0.04 | 4 | 0.27 | 0.15 | 16 | 0.35 | 0.11 | 6 | 0.45 | 0.37-0.52 | 2 | 0.17 | 0.10 | 17 |
| Xylenes | 0.086 | 0.077 | 8 | 0.086 | 0.049 | 3 | 0.16 | 0.090 | 9 | 0.11 | 0.016 | 3 | 0.23 | - | 1 | 0.10 | 0.12 | 10 |
| Ethylbenzene | 0.022 | 0.010 | 8 | 0.043 | 0.034 | 3 | 0.041 | 0.018 | 10 | 0.038 | 0.011 | 3 | 0.042 | - | 1 | 0.044 | 0.049 | 7 |
| Styrene | 0.056 | 0.029 | 6 | *0.028* | - | *0* | 0.066 | 0.028 | 8 | *0.13* | - | *0* | 0.055 | 0.027-0.082 | 2 | 0.043 | 0.027 | 7 |
| PAHs | 0.012 | 0.016 | 4 | *0.15* | - | *0* | 0.017 | 0.019 | 6 | 0.72 | - | 1 | *0.42* | - | *0* | 0.056 | 0.071 | 6 |
| Methanol | 1.35 | 0.47 | 14 | 2.8 | 0.5 | 4 | 2.2 | 0.9 | 20 | 2.33 | 1.45 | 13 | 2.5 | 0.4 | 3 | 3.3 | 2.7 | 11 |
| Ethanol | 0.036 | 0.017-0.055 | 2 | *0.067* | - | *0* | 0.076 | 0.089 | 7 | 0.058 | 0.063 | 3 | *0.17* | - | *0* | *0.05* | - | *0* |
| 1-Propanol | 0.025 | - | 1 | *0.038* | - | *0* | *0.041* | - | *0* | *0.044* | - | *0* | 0.094 | - | *0* | 0.028 | - | *0* |
| 2-Propanol | *0.08* | - | *0* | 0.12 | - | *0* | *0.13* | - | *0* | *0.14* | - | *0* | *0.30* | - | *0* | *0.09* | - | *0* |
| Butanols | 0.11 | 0.008-0.21 | 2 | 0.009 | - | 1 | 0.064 | 0.029-0.098 | 2 | *0.071* | - | *0* | 0.15 | - | *0* | 0.008 | - | 1 |
| Cyclopentanol | 0.033 | - | 1 | 0.032 | - | 1 | *0.035* | - | *0* | *0.038* | - | *0* | *0.081* | - | *0* | 0.012 | - | 1 |
| Phenol | 0.43 | 0.19 | 7 | 0.23 | 0.006-0.45 | 2 | 0.25 | 0.09 | 3 | 0.75 | - | *0* | 0.47 | 0.42-0.51 | 2 | 0.89 | 0.96 | 5 |
| Formaldehyde | 1.23 | 0.65 | 16 | 2.40 | 0.63 | 3 | 2.04 | 0.70 | 16 | 1.75 | 0.40 | 4 | 1.07 | 0.44 | 3 | 1.8 | 0.6 | 8 |
| Acetaldehyde | 0.84 | 0.65 | 9 | 2.26 | 1.55-2.97 | 2 | 1.21 | 0.56 | 14 | 0.81 | 0.23 | 4 | 1.16 | 0.70-1.63 | 2 | 1.8 | 1.0 | 5 |
| Hydroxyacetaldehyde (glycolaldeh.) | 0.13 | 0.08 | 5 | *0.33* | - | *0* | 0.39 | - | 1 | *0.38* | - | *0* | 0.11 | - | 1 | 3.2 | 2.3-4.1 | 2 |
| Glyoxal | *0.33* | - | *0* | 0.50 | - | *0* | *0.54* | - | *0* | 0.59 | - | *0* | *1.3* | - | *0* | 0.24 | - | 1 |
| Methylglyoxal | 0.40 | 0.15-0.64 | 2 | *0.49* | - | *0* | 0.27 | - | 1 | *0.57* | - | *0* | 0.23 | - | 1 | 0.55 | - | 1 |
| Acolein (Propenal) | 0.48 | 0.25 | 6 | 0.65 | - | 1 | 0.34 | 0.13 | 7 | 0.33 | - | 1 | 0.27 | - | 1 | 0.62 | 0.39 | 5 |

| Species | Mean | SD/Range | n | Mean | SD/Range | n | Mean | SD/Range | n | Mean | SD/Range | n | Mean | SD/Range | n | Mean | SD/Range | n |
|---|---|---|---|---|---|---|---|---|---|---|---|---|---|---|---|---|---|---|
| Propanal | 0.053 | 0.009-0.097 | 2 | 0.10 | - | 1 | 0.087 | 0.040 | 4 | 0.24 | - | 1 | *0.34* | - | *0* | 0.18 | - | 1 |
| Butanals | 0.11 | 0.054-0.220 | 2 | 0.13 | 0.073-0.18 | 2 | 0.11 | 0.07 | 5 | *0.15* | - | *0* | 0.02 | - | 1 | 0.17 | 0.01-0.32 | 2 |
| Methacrolein | *0.10* | - | *0* | 0.15 | - | 1 | 0.14 | 0.18 | 5 | 0.11 | 0.12 | 3 | *0.39* | - | *0* | 0.28 | - | 1 |
| Crotonaldehyde (2-butenal) | *0.24* | - | *0* | 0.24 | - | 1 | *0.39* | - | *0* | 0.42 | - | *0* | 0.90 | - | *0* | 0.42 | - | 1 |
| Hexanals | 0.048 | 0.068 | 3 | 0.021 | 0.010-0.031 | 2 | 0.038 | 0.033 | 4 | *0.038* | - | *0* | 0.08 | - | *0* | 0.019 | 0.008-0.03 | 2 |
| Heptanals | 0.003 | 0.001-0.005 | 2 | 0.004 | - | 1 | *0.005* | - | *0* | *0.005* | - | *0* | *0.011* | - | *0* | 0.001 | - | 1 |
| Acetone | 0.47 | 0.18 | 7 | 0.63 | - | 1 | 0.76 | 0.50 | 12 | 1.59 | 1.61 | 7 | 0.91 | 0.69-1.12 | 2 | 0.71 | 0.47 | 5 |
| 2-Butanone | 0.13 | 0.1 | 6 | 0.50 | - | 1 | 0.23 | 0.21 | 9 | 0.16 | 0.04 | 5 | 0.34 | 0.14-0.54 | 2 | 0.58 | 0.31 | 3 |
| 2,3-Butanedione | 0.35 | 0.2 | 4 | 0.73 | - | 1 | 0.89 | 0.86 | 5 | 0.34 | - | 1 | 0.32 | - | 1 | 1.17 | 0.14 | 3 |
| 1-Butene-3-one (Methylvinyl ketone) | 0.23 | - | 1 | 0.39 | - | 1 | 0.165 | 0.109 | 5 | 0.099 | 0.097-0.10 | 2 | 0.057 | - | 1 | 0.48 | 0.25-0.70 | 2 |
| Pentanones | 0.014 | 0.006 | 3 | 0.059 | 0.028-0.090 | 2 | 0.066 | 0.033 | 5 | *0.074* | - | *0* | 0.075 | - | 1 | 0.10 | 0.005-0.20 | 2 |
| Hexanones | 0.048 | - | 1 | *0.057* | - | *0* | 0.045 | 0.043-0.046 | 2 | *0.066* | - | *0* | *0.14* | - | *0* | *0.042* | - | *0* |
| Heptanones | 0.006 | - | 1 | 0.002 | - | 1 | *0.005* | - | *0* | *0.005* | - | *0* | *0.011* | - | *0* | 0.002 | - | 1 |
| Octanones | 0.015 | - | 1 | 0.019 | - | 1 | *0.023* | - | *0* | *0.025* | - | *0* | *0.053* | - | *0* | *0.015* | - | *0* |
| Benzaldehyde | 0.102 | 0.097 | 4 | 0.027 | - | 1 | 0.132 | 0.077 | 3 | *0.096* | - | *0* | 0.056 | - | 1 | 0.038 | 0.006-0.07 | 2 |
| Acetol (hydroxyacetone) | 0.56 | 0.3 | 3 | *1.81* | - | *0* | 1.13 | - | 1 | *2.1* | - | *0* | 0.64 | 0.42-0.86 | 2 | 3.12 | 3.24 | 5 |
| Furan | 0.29 | 0.14 | 8 | 0.33 | 0.25-0.41 | 2 | 0.41 | 0.26 | 8 | 0.36 | 0.28-0.44 | 2 | 1.07 | 0.74-1.4 | 2 | 0.90 | 0.88 | 4 |
| 2-Methyl-furan | 0.20 | 0.14 | 6 | 0.28 | 0.28 | 3 | 0.34 | 0.21 | 5 | *0.42* | - | *0* | 0.31 | 0.12-0.50 | 2 | 0.53 | 0.521 | 3 |
| 3-Methyl-furan | 0.010 | 0.004 | 3 | 0.055 | 0.030-0.080 | 2 | 0.034 | 0.016 | 3 | *0.052* | - | *0* | *0.11* | - | *0* | 0.076 | 0.002-0.15 | 2 |
| 2-Ethylfuran | 0.005 | 0.001-0.009 | 2 | 0.003 | - | 1 | 0.016 | 0.012 | 5 | *0.008* | - | *0* | *0.017* | - | *0* | 0.0003 | - | 1 |
| 2,4-Dimethyl-furan | 0.008 | - | 1 | 0.024 | - | 1 | *0.011* | - | *0* | *0.012* | - | *0* | *0.026* | - | *0* | 0.002 | - | *0* |
| 2,5-Dimethyl-furan | 0.063 | 0.067 | 4 | *0.085* | - | *0* | 0.070 | 0.070 | 5 | *0.10* | - | *0* | 0.14 | - | 1 | 0.098 | - | 1 |
| Tetrahydrofuran | 0.009 | 0.002-0.016 | 2 | 0.017 | - | 1 | 0.001 | 0.0005-0.0017 | 2 | *0.011* | - | *0* | *0.023* | - | *0* | 0.004 | - | 1 |
| 2,3-Dihydrofuran | 0.014 | 0.013-0.015 | 2 | 0.014 | - | 1 | 0.003 | 0.001-0.004 | 2 | *0.012* | - | *0* | *0.026* | - | *0* | 0.004 | - | 1 |
| Benzofuran | 0.045 | 0.040 | 4 | 0.016 | - | 1 | 0.094 | 0.071 | 3 | *0.060* | - | *0* | 0.032 | - | 1 | 0.023 | 0.003-0.044 | 2 |
| Furfural (2-Furaldehyde) | 0.73 | 0.74 | 3 | *0.77* | - | *0* | 0.52 | 0.81 | 7 | 0.61 | - | 1 | 1.10 | 0.12-2.1 | 2 | 1.03 | - | 1 |
| Methyl formate | 0.073 | - | 1 | *0.051* | - | *0* | 0.024 | 0.022-0.027 | 2 | 0.024 | - | 1 | *0.13* | - | *0* | 0.04 | - | *0* |
| Methyl acetate | 0.159 | 0.059-0.26 | 2 | *0.13* | - | *0* | 0.095 | 0.058 | 5 | 0.087 | - | 1 | *0.33* | - | *0* | *0.10* | - | *0* |
| Acetonitrile | 0.17 | 0.07 | 9 | 0.49 | 0.14 | 3 | 0.22 | 0.17 | 14 | 0.31 | 0.10 | 6 | 0.60 | - | 1 | 0.25 | 0.24 | 7 |
| Acrylonitrile | 0.037 | 0.009 | 3 | 0.04 | - | 1 | 0.031 | 0.014 | 6 | *0.068* | - | *0* | - | - | *0* | 0.094 | 0.061 | 3 |
| Propionitrile | 0.027 | 0.012-0.042 | 2 | 0.09 | - | 1 | 0.011 | 0.011-0.012 | 2 | *0.11* | - | *0* | - | - | *0* | 0.17 | - | 1 |
| Pyrrole | 0.013 | - | 1 | 0.12 | - | 1 | 0.062 | 0.085 | 3 | *0.14* | - | *0* | - | - | *0* | 0.22 | - | 1 |
| Trimethylpyazole | - | - | *0* | - | - | *0* | - | - | *0* | - | - | *0* | - | - | *0* | - | - | *0* |
| Methylamine | - | - | *0* | - | - | *0* | - | - | *0* | - | - | *0* | - | - | *0* | - | - | *0* |
| Dimethylamine | - | - | *0* | - | - | *0* | - | - | *0* | - | - | *0* | - | - | *0* | - | - | *0* |
| Ethylamine | - | - | *0* | - | - | *0* | - | - | *0* | - | - | *0* | - | - | *0* | - | - | *0* |
| Trimethylamine | - | - | *0* | - | - | *0* | - | - | *0* | - | - | *0* | - | - | *0* | - | - | *0* |
| n-Pentylamine | - | - | *0* | - | - | *0* | - | - | *0* | - | - | *0* | - | - | *0* | - | - | *0* |
| 2-Methyl-1-butylamine | - | - | *0* | - | - | *0* | - | - | *0* | - | - | *0* | - | - | *0* | - | - | *0* |
| Formic acid | 0.21 | 0.13 | 14 | 0.49 | 0.28 | 4 | 0.91 | 1.18 | 12 | 1.04 | 0.89 | 7 | 0.29 | 0.14 | 3 | 0.56 | 0.45 | 9 |
| Acetic acid | 2.31 | 1.8 | 13 | 3.3 | 0.8 | 3 | 2.74 | 1.60 | 11 | 3.80 | 2.04 | 4 | 4.9 | 0.97 | 3 | 6.1 | 5.9 | 9 |
| $H_2$ | 0.97 | 0.35 | 6 | 3.1 | 0.7 | 5 | 2.1 | 0.4 | 4 | 1.6 | 0.4 | 8 | 1.2 | - | 1 | 2.6 | 2.6-2.7 | 2 |
| $NO_x$ (as NO) | 2.5 | 1.3 | 18 | 2.8 | 1.3 | 7 | 3.0 | 1.8 | 16 | 1.18 | 0.86 | 11 | 1.2 | 0.31-2.2 | 2 | 2.4 | 1.2 | 20 |
| HONO | 0.47 | 0.21 | 6 | 0.85 | - | 1 | 0.33 | 0.17 | 5 | 0.41 | - | 1 | 0.35 | 0.21-0.49 | 2 | 0.37 | 0.04 | 3 |
| $N_2O$ | 0.17 | 0.09 | 12 | *0.20* | - | *0* | 0.25 | 0.12 | 3 | 0.24 | 0.06 | 5 | - | - | *0* | 0.09 | 0.04 | 5 |
| $NH_3$ | 0.89 | 0.49 | 16 | 1.33 | 0.78 | 4 | 0.98 | 0.69 | 22 | 2.5 | 1.75 | 4 | 4.2 | 3.2 | 3 | 0.99 | 0.63 | 14 |
| HCN | 0.44 | 0.26 | 16 | 0.44 | 0.21 | 5 | 0.64 | 0.39 | 12 | 0.53 | 0.30 | 11 | 4.4 | 1.2 | 3 | 0.42 | 0.18 | 7 |
| Cyanogen, $(CN)_2$ | - | - | *0* | - | - | *0* | - | - | *0* | - | - | *0* | - | - | *0* | - | - | *0* |
| $N_2$ | *2.6* | - | *0* | *2.6* | - | *0* | *2.6* | - | *0* | *2.6* | - | *0* | - | - | *0* | *2.6* | - | *0* |
| $SO_2$ | 0.47 | 0.44 | 12 | 0.77 | 0.37 | 3 | 0.70 | 0.48 | 5 | 0.75 | 0.14-0.31 | 2 | 4.3 | - | 1 | 0.80 | 0.71 | 10 |
| Dimethyl sulfide (DMS) | 0.008 | 0.011 | 5 | 0.0022 | - | 1 | 0.014 | 0.015 | 3 | 0.0023 | - | 1 | 0.045 | 0.003-0.088 | 2 | 0.05 | 0.03-0.07 | 2 |
| COS | 0.025 | 0.020 | 4 | 0.050 | 0.047 | 3 | 0.035 | 0.044 | 6 | 0.058 | 0.031 | 3 | 0.110 | - | 1 | 0.059 | 0.070 | 4 |
| HCl | 0.13 | 0.10 | 3 | *0.13* | - | *0* | 0.039 | 0.031 | 3 | *0.13* | - | *0* | 0.008 | - | 1 | 0.18 | 0.255 | 3 |
| $CH_3Cl$ | 0.063 | 0.065 | 16 | 0.029 | 0.02-0.04 | 2 | 0.042 | 0.055 | 8 | 0.060 | 0.033 | 4 | 0.15 | - | 1 | 0.17 | 0.14 | 4 |
| $CH_3Br$ | 0.0027 | 0.0051 | 14 | 0.0078 | 0.005-0.010 | 2 | 0.0015 | 0.0010 | 3 | 0.0029 | 0.0011 | 4 | 0.010 | - | 1 | 0.0011 | - | 1 |
| $CH_3I$ | 0.0007 | 0.0006 | 10 | 0.0068 | - | 1 | 0.0005 | 0.0004-0.001 | 2 | 0.0004 | - | 1 | 0.012 | - | 1 | 0.0002 | - | 1 |
| $Hg^0$ | 4.8E-05 | 4.2E-05 | 4 | 1.0E-04 | 4.7E-5-1.7E-4 | 2 | 2.0E-04 | 1.8E-04 | 6 | 2.3E-04 | 3.0E-04 | 6 | - | - | *0* | 5.1E-05 | 5.0E-05 | 3 |
| $PM_{2.5}$ | 6.7 | 3.3 | 20 | 8.3 | 3.3 | 9 | 18.5 | 14.4 | 29 | 18.7 | 15.9 | 5 | 18.9 | 2.3 | 3 | 8.2 | 4.4 | 18 |
| TPM | 8.7 | 3.1 | 11 | 10.9 | 5.3 | 4 | 18.4 | 8.3 | 11 | 15.3 | 12.3-18.3 | 2 | *27.5* | - | *0* | 12.9 | 7.2 | 7 |
| TC | 3.2 | 1.5 | 10 | 5.5 | 1.6 | 4 | 8.4 | 2.2 | 3 | *9.8* | - | *0* | *14.3* | - | *0* | 5.3 | 3.9 | 22 |

| | | | | | | | | | | | | | | | | | | |
|---|---|---|---|---|---|---|---|---|---|---|---|---|---|---|---|---|---|---|
| OC | 3.0 | 1.5 | 15 | 4.4 | 1.9 | 5 | 10.9 | 7.2 | 13 | 5.9 | 2.5 | 3 | 14.2 | 12.4-16.0 | 2 | 4.9 | 3.6 | 20 |
| BC or EC | 0.53 | 0.35 | 18 | 0.51 | 0.34 | 8 | 0.55 | 0.36 | 14 | 0.43 | 0.21 | 4 | 0.10 | 0.09 | 3 | 0.42 | 0.28 | 24 |
| Levoglucosan | 0.05 | - | 1 | 0.42 | - | 1 | 1.33 | 1.21 | 6 | 1.3 | - | 1 | 0.57 | - | 1 | 0.61 | 0.60 | 8 |
| K | 0.40 | 0.24 | 12 | 0.32 | 0.22 | 4 | 0.17 | 0.16 | 4 | 0.17 | - | 0 | 0.004 | - | 1 | 0.48 | 0.43 | 9 |
| CN | 2.3E+16 | 2.3E+16 | 5 | 3.9E+15 | 1.3E+15 | 3 | 9.2E+15 | - | 0 | 4.2E+15 | - | 1 | - | - | 0 | 4.9E+15 | 2.0E+15 | 4 |
| CCN (0.5% SS) | 7.9E+14 | - | 1 | 1.7E+15 | 1.64E+15-1.68E+15 | 2 | 2.0E+15 | 3.4E+15 | 3 | 1.6E+15 | - | 0 | - | - | 0 | 1.0E+15 | - | 0 |
| N(>~ 0.12 μm diameter) | 1.2E+15 | 8.5E+14 | 4 | 2.7E+15 | - | 1 | 1.0E+15 | - | 0 | 1.0E+15 | - | 0 | - | - | 0 | 1.0E+15 | - | 0 |

[a]) Emission factors are given in gram species per kilogram dry matter burned. See text for the conventions used for reporting uncertainties. Abbreviations
are as follows: PM$_{2.5}$, particulate matter <2.5 mm diameter; TPM, total particulate matter; TC, total carbon; BC, black carbon; CN, condensation nuclei;
CCN(0.5% SS), cloud condensation nuclei at 0.5% supersaturation; and N(>~0.12 mm diam), particles > ~0.12 μm diameter. Values in italics represent estimates for
emission factors that have not been measured directly.

[b]) Estimation method for emission factors for which no measurements are available. See text section 2.4 for deta

[c]) based on field measurements that only include varying (often incomplete) sets of identified species (see text for discussi

[d]) Sum of chemically identified and unidentified species, from online updates to Akagi et al. (201

| Species | Biofuels (without dung) | | | Dung | | | Charcoal making | | | Charcoal burning | | | Garbage burning | | | Lab studies | | | EM[b] |
|---|---|---|---|---|---|---|---|---|---|---|---|---|---|---|---|---|---|---|---|
| | average | std.dev. | N | average | std.dev. | N | average | std.dev. | N | average | std.dev. | N | average | std.dev. | N | average | std.dev. | N | |
| MCE | 0.92 | 0.03 | 39 | 0.88 | 0.04 | 9 | 0.79 | 0.04 | 8 | 0.88 | 0.04 | 15 | 0.93 | 0.02 | 3 | 0.90 | 0.10 | 48 | --- |
| $CO_2$ | 1550 | 170 | 36 | 1050 | 230 | 9 | 490 | 70 | 7 | 2500 | 350 | 14 | 1400 | 180 | 3 | 1590 | 330 | 42 | --- |
| CO | 83 | 29 | 61 | 89 | 42 | 14 | 93 | 39 | 9 | 207 | 63 | 17 | 66 | 20 | 3 | 93 | 61 | 45 | --- |
| $CH_4$ | 6.8 | 6.0 | 28 | 8.9 | 4.9 | 8 | 19.0 | 19.9 | 8 | 6.0 | 2.6 | 9 | 4.2 | 0.6 | 3 | 5.9 | 4.8 | 33 | --- |
| Total VOC | 7.8 | 5.0 | 23 | 14.4 | 10.0-18.8 | 2 | 26.4 | 18.1 | 4 | 6.6 | 4.9 | 6 | 8.2 | - | 1 | 18.0 | 16.4 | 10 | CO |
| Total NMOG, including unidentifie[c] | 58 | - | - | 98 | - | - | 321 | - | - | 11 | - | - | 23 | - | - | --- | --- | --- | --- |
| $C_2H_2$ | 0.68 | 0.37 | 14 | 0.68 | 0.41 | 3 | 0.28 | 0.24 | 3 | 0.27 | 0.18 | 4 | 0.52 | 0.13 | 3 | 0.35 | 0.31 | 26 | --- |
| $C_2H_4$ | 1.33 | 0.90 | 15 | 2.25 | 1.36 | 4 | 1.51 | 0.78 | 4 | 0.51 | 0.34 | 6 | 2.2 | 0.8 | 3 | 1.6 | 0.8 | 29 | --- |
| $C_2H_6$ | 0.63 | 0.61 | 13 | 1.28 | 0.70 | 3 | 2.4 | 1.3-3.4 | 2 | 0.76 | 0.34 | 5 | 1.6 | 1.5-1.7 | 2 | 1.12 | 1.18 | 11 | --- |
| $C_3H_4$ | 0.13 | 0.14 | 3 | 0.11 | 0.06 | 3 | *0.09* | *-* | *0* | *0.21* | *-* | *0* | 0.54 | 0.09-0.99 | 2 | 0.54 | 0.41 | 7 | CO |
| $C_3H_6$ | 0.40 | 0.29 | 13 | 1.45 | 0.46 | 4 | 1.03 | 0.32 | 4 | 0.53 | 0.19 | 3 | 1.6 | 0.3 | 3 | 0.85 | 0.75 | 24 | --- |
| $C_3H_8$ | 0.24 | 0.25 | 7 | 0.50 | 0.31 | 3 | 0.53 | - | 1 | 0.17 | 0.12 | 3 | 0.75 | 0.59-0.90 | 2 | 0.27 | 0.22 | 8 | --- |
| 1-Butene | 0.23 | 0.24 | 9 | 0.31 | 0.131 | 3 | - | - | 0 | 0.12 | 0.040-0.20 | 2 | 1.07 | 1.05-1.09 | 2 | 0.27 | 0.30 | 12 | CO |
| i-Butene | 0.26 | 0.33 | 6 | 0.26 | 0.11 | 3 | - | - | 0 | 0.091 | 0.026-0.16 | 2 | 0.63 | - | 1 | 0.18 | 0.15 | 5 | CO |
| trans-2-Butene | 0.05 | 0.03 | 3 | 0.12 | 0.06 | 3 | - | - | 0 | 0.040 | 0.016-0.063 | 2 | 0.17 | 0.17-0.17 | 2 | 0.123 | 0.152 | 7 | CO |
| cis-2-Butene | 0.04 | 0.02 | 3 | 0.081 | 0.041 | 3 | - | - | 0 | 0.025 | 0.016-0.034 | 2 | 0.13 | 0.12-0.14 | 2 | 0.158 | 0.295 | 7 | CO |
| Butadiene | 0.15 | 0.10 | 9 | 0.30 | 0.10 | 3 | - | - | 0 | 0.11 | 0.09 | 3 | 0.20 | 0.14-0.27 | 2 | 0.17 | 0.10 | 14 | CO |
| n-Butane | 0.19 | 0.41 | 7 | 0.18 | 0.12 | 3 | - | - | 0 | 0.074 | 0.053-0.095 | 2 | 0.40 | 0.28-0.51 | 2 | 0.188 | 0.192 | 7 | CO |
| i-Butane | 0.15 | 0.22 | 3 | 0.10 | 0.10 | 3 | - | - | 0 | 0.012 | 0.010-0.013 | 2 | 0.09 | 0.06-0.12 | 2 | 0.444 | 0.783 | 6 | CO |
| 1-Pentene | 0.03 | 0.03 | 6 | 0.11 | 0.06 | 3 | - | - | 0 | 0.028 | - | 1 | 0.47 | 0.21-0.73 | 2 | 0.136 | 0.315 | 8 | CO |
| 2-Pentenes | 0.02 | 0.02-0.02 | 2 | 0.090 | 0.071 | 3 | - | - | 0 | *0.051* | *-* | *0* | 0.23 | 0.16-0.30 | 2 | 0.090 | 0.091 | 7 | CO |
| n-Pentane | 0.018 | 0.021 | 8 | 0.093 | 0.088 | 3 | - | - | 0 | 0.096 | - | 1 | 0.74 | 0.39-1.08 | 2 | 0.076 | 0.075 | 9 | CO |
| Methyl-butenes | 0.014 | 0.012 | 3 | 0.036 | 0.021 | 3 | - | - | 0 | 0.015 | - | 1 | 0.041 | - | 1 | 0.202 | 0.356 | 7 | CO |
| 2-Methyl-butane | 0.045 | 0.043 | 4 | 0.34 | 0.41 | 3 | - | - | 0 | 0.071 | - | 1 | 0.22 | 0.04-0.39 | 2 | 0.096 | 0.080 | 4 | CO |
| n-Pentadienes | 0.017 | 0.015 | 4 | 0.039 | 0.02-0.06 | 2 | - | - | 0 | *0.084* | *-* | *0* | *0.026* | *-* | *0* | 0.171 | 0.266 | 5 | CO |
| Isoprene | 0.06 | 0.05 | 10 | 0.20 | 0.12 | 3 | - | - | 0 | 0.12 | 0.017-0.22 | 2 | 0.10 | 0.07-0.13 | 2 | 0.34 | 0.397 | 18 | CO |
| Cyclopentene | 0.008 | - | 1 | *0.018* | *-* | *0* | - | - | 0 | 0.035 | - | 1 | *0.014* | *-* | *0* | 0.055 | 0.044 | 7 | CO |
| Cyclopentadiene | 0.061 | 0.047 | 4 | *0.030* | *-* | *0* | - | - | 0 | *0.071* | *-* | *0* | *0.022* | *-* | *0* | 0.038 | 0.038 | 6 | CO |
| 4-Methyl-1-pentene | 0.015 | - | 1 | *0.032* | *-* | *0* | - | - | 0 | *0.074* | *-* | *0* | *0.023* | *-* | *0* | 0.005 | 0.002-0.008 | 2 | CO |
| 2--Methyl-1-pentene | *0.029* | *-* | *0* | *0.031* | *-* | *0* | - | - | 0 | *0.073* | *-* | *0* | *0.023* | *-* | *0* | 0.019 | - | 1 | CO |
| 1-Hexene | 0.018 | 0.007 | 5 | 0.11 | 0.06-0.17 | 2 | - | - | 0 | *0.11* | *-* | *0* | *0.036* | *-* | *0* | 0.045 | 0.040 | 8 | CO |
| Hexadienes | *0.006* | *-* | *0* | *0.006* | *-* | *0* | - | - | 0 | *0.01* | *-* | *0* | *0.005* | *-* | *0* | 0.061 | 0.066 | 5 | CO |
| n-Hexane | 0.009 | 0.006 | 6 | 0.12 | 0.15 | 3 | - | - | 0 | 0.185 | 0.063-0.31 | 2 | 0.23 | 0.17-0.28 | 2 | 0.026 | 0.023 | 8 | CO |
| Isohexanes | 0.065 | 0.084 | 3 | 0.18 | 0.28 | 3 | - | - | 0 | *0.09* | *-* | *0* | 0.17 | 0.05-0.28 | 2 | 0.062 | 0.095 | 6 | CO |
| Heptanes | 0.005 | 0.003 | 6 | 0.11 | - | 1 | - | - | 0 | *0.047* | *-* | *0* | 0.23 | - | 1 | 0.080 | 0.097 | 6 | CO |
| Octenes | 0.007 | 0.014 | 4 | *0.015* | *-* | *0* | - | - | 0 | *0.036* | *-* | *0* | *0.011* | *-* | *0* | 0.044 | 0.038 | 3 | CO |
| Terpenes | 0.10 | 0.14 | 7 | 0.12 | 0.199 | 3 | - | - | 0 | - | - | 0 | 0.092 | - | 1 | 0.46 | 0.51 | 14 | CO |
| Benzene | 0.95 | 0.89 | 17 | 1.25 | 0.63 | 3 | - | - | 0 | 1.23 | 0.72-1.7 | 2 | 1.9 | 0.8 | 3 | 0.60 | 0.79 | 24 | CO |
| Toluene | 0.45 | 0.51 | 14 | 0.87 | 0.39 | 3 | - | - | 0 | 0.41 | 0.20-0.62 | 2 | 0.60 | 0.22 | 3 | 0.40 | 0.54 | 24 | CO |
| Xylenes | 0.13 | 0.15 | 10 | 0.32 | 0.282 | 3 | - | - | 0 | 0.16 | 0.099-0.23 | 2 | 0.30 | 0.27 | 3 | 0.14 | 0.10 | 11 | CO |
| Ethylbenzene | 0.10 | 0.10 | 7 | 0.17 | 0.18 | 3 | - | - | 0 | 0.053 | 0.033-0.074 | 2 | 0.33 | 0.16 | 3 | 0.086 | 0.141 | 10 | CO |
| Styrene | 0.18 | 0.19 | 8 | 0.13 | 0.11 | 3 | - | - | 0 | 0.14 | 0.066-0.22 | 2 | 0.30 | 0.07-0.53 | 2 | 0.064 | 0.062 | 15 | CO |
| PAHs | 0.09 | 0.10 | 13 | 0.023 | - | 1 | - | - | 0 | 0.53 | 0.41-0.66 | 2 | 0.028 | 0.011-0.045 | 2 | 0.061 | 0.119 | 11 | CO |
| Methanol | 2.0 | 1.1 | 9 | 3.2 | 1.1 | 4 | 13.0 | 6.1 | 3 | 1.0 | 0.72-1.24 | 2 | 1.54 | 1.18 | 3 | 2.17 | 2.30 | 30 | --- |
| Ethanol | 0.075 | 0.02-0.13 | 2 | 0.23 | 0.29 | 3 | *0.060* | *-* | *0* | 0.13 | - | 0 | 0.09 | - | 1 | 0.084 | 0.072-0.097 | 2 | CO |
| 1-Propanol | *0.030* | *-* | *0* | *0.032* | *-* | *0* | - | - | 0 | 0.07 | - | 0 | *0.024* | *-* | *0* | 0.25 | - | 1 | CO |
| 2-Propanol | *0.10* | *-* | *0* | 0.10 | - | 0 | - | - | 0 | 0.24 | - | 0 | *0.076* | *-* | *0* | 0.11 | 0.005-0.21 | 2 | LV |
| Butanols | 0.051 | 0.01-0.10 | 2 | *0.052* | *-* | *0* | - | - | 0 | 0.12 | - | 0 | *0.039* | *-* | *0* | 0.015 | 0.006 | 5 | CO |
| Cyclopentanol | *0.026* | *-* | *0* | *0.028* | *-* | *0* | - | - | 0 | 0.06 | - | 0 | *0.020* | *-* | *0* | - | - | 0 | CO |
| Phenol | 0.72 | 1.15 | 7 | 1.58 | 1.0-2.2 | 2 | 4.7 | 2.8-6.6 | 2 | 2.0 | - | 1 | 0.30 | 0.16 | 3 | 1.06 | 1.42 | 18 | Phenol |
| Formaldehyde | 0.87 | 1.00 | 13 | 2.42 | - | 1 | 1.1 | - | 1 | 0.51 | 0.19 | 3 | 0.9 | 1.0 | 4 | 1.37 | 0.73 | 30 | --- |
| Acetaldehyde | 0.41 | 0.32 | 13 | 1.46 | 0.58 | 3 | - | - | 0 | 0.13 | - | 1 | 2.1 | 1.6 | 3 | 1.62 | 1.47 | 21 | CO |
| Hydroxyacetaldehyde (glycolaldeh.) | 0.33 | 0.11 | 3 | 0.50 | - | 1 | - | - | 0 | *0.6* | *-* | *0* | 2.4 | - | 1 | 0.63 | 0.68 | 9 | CO |
| Glyoxal | 0.58 | - | 1 | *0.43* | *-* | *0* | - | - | 0 | *1.0* | *-* | *0* | *0.32* | *-* | *0* | 0.41 | 0.43 | 8 | CO |
| Methylglyoxal | 0.39 | 0.18-0.60 | 2 | *0.42* | *-* | *0* | - | - | 0 | *1.0* | *-* | *0* | *0.31* | *-* | *0* | 0.33 | 0.26 | 11 | CO |
| Acolein (Propenal) | 0.085 | 0.093 | 11 | 0.24 | 0.19-0.30 | 2 | - | - | 0 | *1.0* | *-* | *0* | 0.36 | 0.027-0.70 | 2 | 0.55 | 0.37 | 15 | CO |

| Compound | | | | | | | | | | | | | | | | | | Category |
|---|---|---|---|---|---|---|---|---|---|---|---|---|---|---|---|---|---|---|
| Propanal | 0.072 | 0.069 | 8 | 0.11 | - | 0 | - | - | 0 | 0.27 | - | 0 | 0.11 | - | 1 | 0.23 | 0.28 | 9 | CO |
| Butanals | 0.027 | 0.019 | 11 | 0.07 | 0.035 | 3 | - | - | 0 | 0.26 | - | 0 | 0.26 | - | 1 | 0.09 | 0.08 | 8 | CO |
| Methacrolein | 0.028 | 0.025-0.031 | 2 | 0.14 | - | 0 | - | - | 0 | 0.31 | - | 0 | 0.10 | - | 0 | 0.10 | 0.08 | 6 | CO |
| Crotonaldehyde (2-butenal) | 0.22 | - | 1 | 0.31 | - | 0 | - | - | 0 | 0.72 | - | 0 | 0.034 | - | 1 | 0.09 | 0.10 | 6 | CO |
| Hexanals | 0.006 | 0.003 | 3 | 0.028 | - | 0 | - | - | 0 | 0.06 | - | 0 | 0.020 | - | 0 | 0.009 | 0.008 | 5 | CO |
| Heptanals | 0.003 | - | 0 | 0.004 | - | 0 | - | - | 0 | 0.008 | - | 0 | 0.003 | - | 0 | 0.007 | 0.004-0.011 | 2 | CO |
| Acetone | 0.35 | 0.23 | 11 | 1.5 | 0.7 | 3 | 0.26 | - | 1 | 1.6 | - | 0 | 1.4 | 1.0 | 3 | 0.64 | 0.42 | 18 | CO |
| 2-Butanone | 0.095 | 0.064 | 12 | 0.31 | 0.17 | 3 | 0.29 | - | 0 | 0.65 | - | 0 | 0.13 | 0.04-0.21 | 2 | 0.22 | 0.22 | 15 | CO |
| 2,3-Butanedione | 0.21 | 0.01-0.41 | 2 | 0.60 | - | 0 | 0.63 | - | 0 | 1.4 | - | 0 | 0.85 | - | 1 | 0.51 | 0.44 | 12 | CO |
| 1-Butene-3-one (Methylvinyl ketone) | 0.058 | 0.05-0.07 | 2 | 0.20 | 0.13-0.28 | 2 | 0.253 | - | 0 | 0.56 | - | 0 | 0.18 | - | 0 | 0.16 | 0.17 | 6 | CO |
| Pentanones | 0.029 | - | 1 | 0.055 | - | 0 | - | - | 0 | 0.13 | - | 0 | 0.040 | - | 0 | 0.067 | 0.063 | 7 | CO |
| Hexanones | 0.045 | - | 0 | 0.048 | - | 0 | - | - | 0 | 0.11 | - | 0 | 0.036 | - | 0 | 0.014 | 0.010 | 5 | CO |
| Heptanones | 0.003 | - | 0 | 0.004 | - | 0 | - | - | 0 | 0.008 | - | 0 | 0.003 | - | 0 | 0.063 | 0.009-0.12 | 2 | CO |
| Octanones | 0.017 | - | 0 | 0.018 | - | 0 | - | - | 0 | 0.042 | - | 0 | 0.013 | - | 0 | 0.005 | - | 1 | CO |
| Benzaldehyde | 0.044 | 0.048 | 8 | 0.070 | - | 0 | - | - | 0 | 0.16 | - | 0 | 0.15 | - | 1 | 0.081 | 0.046 | 11 | CO |
| Acetol (hydroxyacetone) | 0.87 | 0.48-1.26 | 2 | 6.40 | 3.2-9.6 | 2 | 9.4 | - | 1 | 3.6 | - | 0 | 1.1 | 0.6-1.7 | 2 | 0.7 | 0.9 | 12 | CO |
| Furan | 0.20 | 0.10 | 8 | 0.49 | 0.35 | 4 | 0.80 | 0.45-1.2 | 2 | 0.39 | - | 1 | 0.21 | - | 1 | 0.53 | 0.62 | 22 | --- |
| 2-Methyl-furan | 0.16 | 0.08 | 6 | 0.30 | 0.12-0.49 | 2 | - | - | 0 | 0.72 | - | 0 | 0.23 | - | 0 | 0.29 | 0.28 | 13 | CO |
| 3-Methyl-furan | 0.013 | - | 1 | 0.038 | - | 0 | - | - | 0 | 0.09 | - | 0 | 0.028 | - | 0 | 0.13 | 0.28 | 7 | CO |
| 2-Ethylfuran | 0.005 | - | 0 | 0.006 | - | 0 | - | - | 0 | 0.013 | - | 0 | 0.004 | - | 0 | 0.011 | 0.012 | 4 | CO |
| 2,4-Dimethyl-furan | 0.003 | 0.001 | 3 | 0.009 | - | 0 | - | - | 0 | 0.021 | - | 0 | 0.007 | - | 0 | 0.013 | 0.015 | 4 | CO |
| 2,5-Dimethyl-furan | 0.035 | 0.009 | 3 | 0.072 | - | 0 | - | - | 0 | 0.17 | - | 0 | 0.053 | - | 0 | 0.11 | 0.16 | 11 | CO |
| Tetrahydrofuran | 0.007 | - | 0 | 0.008 | - | 0 | - | - | 0 | 0.018 | - | 0 | 0.006 | - | 0 | 0.012 | 0.011 | 5 | CO |
| 2,3-Dihydrofuran | 0.008 | - | 0 | 0.009 | - | 0 | - | - | 0 | 0.021 | - | 0 | 0.007 | - | 0 | 0.049 | 0.061 | 4 | CO |
| Benzofuran | 0.046 | - | 1 | 0.044 | - | 0 | - | - | 0 | 0.10 | - | 0 | 0.033 | - | 0 | 0.029 | 0.015 | 12 | CO |
| Furfural (2-Furaldehyde) | 0.28 | 0.37 | 5 | 0.20 | 0.09-0.32 | 2 | 1.6 | - | 1 | 1.5 | - | 0 | 0.49 | - | 0 | 0.84 | 0.74 | 12 | CO |
| Methyl formate | 0.040 | - | 0 | 0.044 | - | 0 | - | - | 0 | 0.10 | - | 0 | 0.03 | - | 0 | 0.043 | - | 1 | CO |
| Methyl acetate | 0.105 | 0.05-0.17 | 2 | 0.11 | - | 0 | - | - | 0 | 0.27 | - | 0 | 0.08 | - | 0 | 0.054 | 0.074 | 6 | CO |
| Acetonitrile | 0.10 | 0.02-0.18 | 2 | - | - | 0 | - | - | 0 | - | - | 0 | 0.58 | - | 1 | 0.21 | 0.18 | 19 | CO |
| Acrylonitrile | 0.030 | - | 1 | - | - | 0 | - | - | 0 | - | - | 0 | - | - | 0 | 0.062 | 0.09 | 9 | CO |
| Propionitrile | - | - | 0 | - | - | 0 | - | - | 0 | - | - | 0 | - | - | 0 | 0.12 | 0.20 | 8 | CO |
| Pyrrole | - | - | 0 | - | - | 0 | - | - | 0 | - | - | 0 | 0.23 | - | 1 | 0.09 | 0.16 | 6 | CO |
| Trimethylpyazole | - | - | 0 | - | - | 0 | - | - | 0 | - | - | 0 | - | - | 0 | 0.124 | - | 1 | LV |
| Methylamine | - | - | 0 | - | - | 0 | - | - | 0 | - | - | 0 | - | - | 0 | 0.057 | - | 1 | LV |
| Dimethylamine | - | - | 0 | - | - | 0 | - | - | 0 | - | - | 0 | - | - | 0 | 0.062 | - | 1 | LV |
| Ethylamine | - | - | 0 | - | - | 0 | - | - | 0 | - | - | 0 | - | - | 0 | 0.005 | 0.0004-0.010 | 2 | LV |
| Trimethylamine | - | - | 0 | - | - | 0 | - | - | 0 | - | - | 0 | - | - | 0 | 0.041 | - | 1 | LV |
| n-Pentylamine | - | - | 0 | - | - | 0 | - | - | 0 | - | - | 0 | - | - | 0 | 0.44 | - | 1 | LV |
| 2-Methyl-1-butylamine | - | - | 0 | - | - | 0 | - | - | 0 | - | - | 0 | - | - | 0 | 0.14 | - | 1 | LV |
| Formic acid | 0.23 | 0.22 | 6 | 0.39 | 0.34-0.43 | 2 | 0.19 | 0.16-0.21 | 2 | 0.09 | 0.02 | 3 | 0.29 | 0.22 | 3 | 0.34 | 0.24 | 30 | --- |
| Acetic acid | 3.9 | 3.7 | 6 | 10.8 | 7.3-14.3 | 2 | 47 | 47 | 3 | 1.85 | 0.49-3.2 | 2 | 1.1 | 1.2 | 3 | 3.6 | 3.9 | 33 | --- |
| $H_2$ | 1.8 | - | 0 | 2.0 | - | 0 | - | - | 0 | 4.6 | - | 0 | 1.5 | - | 0 | - | - | 0 | CO |
| $NO_x$ (as NO) | 1.3 | 0.7 | 18 | 2.7 | 3.9 | 5 | 0.24 | 0.43 | 5 | 2.3 | 1.8 | 7 | 1.5 | 0.7-2.2 | 2 | 2.1 | 0.8 | 32 | --- |
| HONO | 0.37 | 0.29-0.45 | 2 | 0.26 | 0.24-0.28 | 2 | - | - | 0 | 0.30 | 0.17-0.42 | 2 | 0.49 | - | 1 | 0.36 | 0.20 | 17 | --- |
| $N_2O$ | 0.07 | 0.02 | 4 | 0.31 | - | 1 | 0.025 | 0.014 | 4 | 0.34 | 0.37 | 3 | - | - | 0 | 0.071 | 0.006 | 4 | AV |
| $NH_3$ | 0.42 | 0.51 | 5 | 3.1 | 2.2 | 4 | 3.8 | 7.2 | 4 | 0.72 | 0.49 | 4 | 0.90 | 0.7-1.1 | 2 | 1.21 | 1.01 | 27 | --- |
| HCN | 0.39 | 0.22-0.56 | 2 | 1.27 | 0.53-2.0 | 2 | 0.09 | - | 1 | - | - | 0 | 0.43 | - | 1 | 0.83 | 1.08 | 27 | --- |
| Cyanogen, $(CN)_2$ | - | - | 0 | - | - | 0 | - | - | 0 | - | - | 0 | - | - | 0 | 0.007 | - | 1 | --- |
| $N_2$ | 2.6 | - | 0 | - | - | 0 | - | - | 0 | - | - | 0 | - | - | 0 | 4.1 | - | 1 | LV |
| $SO_2$ | 0.56 | 0.70 | 14 | 0.66 | 0.60 | 4 | 0.20 | - | 1 | 0.57 | 0.55 | 3 | 0.50 | - | 1 | 0.88 | 0.92 | 23 | AV |
| Dimethyl sulfide (DMS) | 0.12 | 0.13 | 3 | 0.025 | 0.017 | 3 | - | - | 0 | - | - | 0 | 0.029 | 0.007-0.05 | 2 | 0.23 | 0.31 | 11 | AV |
| COS | 0.017 | 0.01-0.02 | 2 | 0.21 | 0.13 | 3 | - | - | 0 | - | - | 0 | 0.074 | - | 1 | 0.062 | 0.120 | 6 | --- |
| HCl | 0.075 | - | 1 | 0.038 | - | 1 | - | - | 0 | 0.11 | - | 1 | 2.8 | 1.8 | 3 | 0.22 | 0.25 | 7 | AV |
| $CH_3Cl$ | 0.184 | 0.278 | 6 | 2.2 | 2.2 | 3 | - | - | 0 | 0.011 | - | 1 | 0.43 | 0.16-0.70 | 2 | 0.12 | 0.13 | 12 | --- |
| $CH_3Br$ | 0.0007 | 0.0006-0.0008 | 2 | 0.0087 | 0.0049 | 3 | - | - | 0 | - | - | 0 | 0.002 | - | 1 | 0.001 | 0.0005-0.001 | 2 | CO |
| $CH_3I$ | 0.0001 | 0.0001-0.0001 | 2 | 0.0006 | 0.0002 | 3 | - | - | 0 | - | - | 0 | 0.0003 | - | 1 | 0.006 | 0.000-0.011 | 2 | CO |
| $Hg^0$ | 4.7E-05 | 6.8E-6-8.8E-5 | 2 | - | - | 0 | - | - | 0 | - | - | 0 | - | - | 0 | 3.7E-05 | - | 1 | --- |
| $PM_{2.5}$ | 6.8 | 4.4 | 58 | 16.5 | 13.0 | 8 | 20.1 | 2.1-38.2 | 2 | 3.0 | 2.5 | 7 | 9.7 | 2.1 | 3 | 10.5 | 11.0 | 23 | --- |
| TPM | 7.0 | 5.8 | 27 | 6.1 | 6.0 | 5 | 13.8 | 19.3 | 4 | 2.1 | 1.7-2.4 | 2 | - | - | 0 | 8.1 | 6.1 | 12 | CO |
| TC | 3.4 | 1.4 | 28 | 8.1 | 9.9 | 4 | - | - | 0 | 2.0 | 2.0 | 5 | 6.9 | - | 0 | 6.5 | 9.1 | 7 | BE |

| | | | | | | | | | | | | | | | | | | |
|---|---|---|---|---|---|---|---|---|---|---|---|---|---|---|---|---|---|---|
| OC | 3.1 | 2.1 | 66 | 10.2 | 9.4 | 6 | 0.8 | - | 1 | 2.2 | 2.0 | 4 | 5.5 | 5.3-5.7 | 2 | 4.9 | 4.6 | 28 | --- |
| BC or EC | 0.81 | 1.19 | 66 | 0.31 | 0.26 | 7 | 0.02 | - | 1 | 0.27 | 0.15 | 3 | 1.4 | 5.1 | 3 | 0.56 | 0.39 | 29 | --- |
| Levoglucosan | 0.50 | 0.42 | 13 | 0.45 | 0.16-0.74 | 2 | 0.06 | - | 1 | 0.79 | - | 1 | 0.40 | 0.25-0.56 | 2 | 0.45 | 0.44 | 13 | --- |
| K | 0.13 | 0.24 | 22 | 0.09 | 0.10 | 4 | 0.004 | - | 1 | 0.75 | 0.79 | 3 | 0.02 | 0.01-0.02 | 2 | 0.36 | 0.34 | 20 | BE |
| CN | 2.9E+15 | 3.0E+15 | 4 | - | - | 0 | - | - | 0 | 4.9E+15 | 2.5E+15-7.2E+15 | 2 | 5.4E+15 | - | 0 | 4.2E+15 | 6.7E+15 | 5 | CO |
| CCN (0.5% SS) | 1.1E+15 | - | 0 | - | - | 0 | - | - | 0 | - | - | 0 | - | - | 0 | 8.0E+14 | - | 1 | CO |
| N(>~ 0.12 μm diameter) | 1.0E+15 | - | 0 | - | - | 0 | - | - | 0 | - | - | 0 | - | - | 0 | - | - | 0 | CO |

Table 2: Estimates of biomass burned (Tg dry matter) annually in the various fire categories

| Source | Savanna/ grassland | Tropical forest | Temperate forest | Boreal forest | Peat | Agricultural residues | Total open fires | Years |
|---|---|---|---|---|---|---|---|---|
| FINN[a] | 1920 | 3200 | 260 | 137 | --- | 210 | 5730 | 2005-2010 |
| GFED4.1s[b] | 2980 | 690 | 100 | 330 | 161 | 290 | 4550 | 2005-2015 |
| GFAS1.2[c] | 2540 | 910 | 110 | 460 | 183 | 63 | 4260 | 2003-2018 |
| QFED[d] | 3690 | 850 | 280 | 200 | --- | --- | 5560 | 2003-2012 |
| FEER[e] | --- | --- | --- | --- | --- | --- | 9330 | 2000-2012 |
| FLAMBE[f] | 870 | 8750 | 750 | 1120 | --- | 99 | 11580 | 2005-2015 |
| ECLIPSE V6a[g] | --- | --- | --- | --- | --- | 530 | --- | 2005-2010 |
| Average | 2400 | 2880 | 300 | 450 | 172 | 240 | 6440 | |

| | Wood etc. | Charcoal making | Charcoal burning | Agricultural waste | Dung | | Total biofuel | |
|---|---|---|---|---|---|---|---|---|
| Fernandes[h] | 1350 | 156 | 39 | 500 | 75 | | 2120 | 2000 |
| FAO[i] | --- | --- | 53 | --- | --- | | --- | 2014 |
| ECLIPSE V6a[g] | 1780 | --- | 44 | 350 | 89 | | 2270 | 2005-2010 |
| CEDS[j] | 1590 | --- | 46 | 580 | 88 | | 2490 | 2010 |
| Average | 1570 | 180 | 45 | 480 | 84 | | 2360 | |

Grand total from all biomass burning                                       8800

[a]Wiedinmyer et al. (2011)

[b]from http://www.geo.vu.nl/~gwerf/GFED/GFED4/tables/GFED4.1s_C.txt assuming 45% C in biofuel

[c]I. Hüser, personal communication 2019, based on methodology in Kaiser et al. (2012)

[d]A. Darmenov, personal communication 2019, based on methodology in Darmenov and da Silva (2015). Emissions from boreal fires were calculated from extratropical fires north of 50 °N, and temperate emissions were calculated by subtracting boreal from extratropical emissions; emissions from crop residue burning fires are included in the grassland fire category.

[e]Ichoku and Ellison (2014), not included in category averages because breakdown not available

[f]E. Hyer, personal communication 2019, based on methodology in Reid et al. (2009). Temperate and boreal emissions were calculated by splitting extratopical burning 40%/60%.

[g]Z. Klimont, personal communication 2019, based on methodology in Klimont et al. (2017)

[h]Fernandes et al. (2007)

[i]FAO (2015)

[j]S. Smith, personal communication 2019, based on methodology in Hoesly et al. (2019)

Table 3: Global emission of selected species based on the emission factors in Table 1 and the biomass burning estimates in Table 2 (Tg a $^{-1}$).

| | Savanna and grassland | Tropical forest | Temperate forest | Boreal forest | Peat fires | Agricultural residues | Biofuel burning | Charcoal making | Charcoal burning | Total | A&M2001 |
|---|---|---|---|---|---|---|---|---|---|---|---|
| Tg dm burned | 2400 | 2880 | 300 | 450 | 172 | 240 | 2134 | 180 | 45 | 8800 | 8600 |
| | | | | | | | | | | | |
| $CO_2$ | 3980 | 4670 | 470 | 690 | 270 | 340 | 3310 | 90 | 110 | 13900 | 13400 |
| CO | 170 | 300 | 34 | 55 | 45 | 18 | 180 | 17 | 9.4 | 820 | 690 |
| $CH_4$ | 6.5 | 19 | 1.6 | 2.5 | 1.6 | 1.4 | 15 | 3.4 | 0.27 | 50 | 39 |
| Total VOC | 12.2 | 16 | 4.0 | 2.7 | 3.7 | 1.8 | 17 | 4.8 | 0.3 | 62 | 49 |
| Total NMOG[a] | 72 | 149 | 11.7 | 26 | 23 | 12.3 | 123 | 58 | 0.5 | 480 | 49 |
| $C_2H_2$ | 0.75 | 1.0 | 0.09 | 0.13 | 0.02 | 0.07 | 1.4 | 0.05 | 0.012 | 3.6 | 3.7 |
| Methanol | 3.2 | 8.1 | 0.7 | 1.0 | 0.4 | 0.8 | 4.3 | 2.3 | 0.04 | 21 | 12.7 |
| Formaldehyde | 2.9 | 6.9 | 0.6 | 0.8 | 0.2 | 0.4 | 1.9 | --- | 0.02 | 14 | 5.5 |
| Acetaldehyde | 2.0 | 6.5 | 0.36 | 0.37 | 0.20 | 0.43 | 0.87 | --- | 0.01 | 10.8 | 3.5 |
| Acetone | 1.1 | 1.81 | 0.23 | 0.72 | 0.16 | 0.17 | 0.74 | 0.05 | 0.07 | 5.1 | 3.0 |
| Acetonitrile | 0.40 | 1.42 | 0.07 | 0.14 | 0.10 | 0.06 | 0.21 | --- | --- | 2.4 | 1.3 |
| Formic acid | 0.5 | 1.4 | 0.3 | 0.5 | 0.1 | 0.1 | 0.49 | 0.03 | 0.00 | 3.3 | 5.9 |
| Acetic acid | 5.5 | 9.5 | 0.8 | 1.7 | 0.8 | 1.5 | 8.4 | 8.4 | 0.08 | 37 | 12.6 |
| $H_2$ | 2.3 | 8.9 | 0.6 | 0.7 | 0.2 | 0.6 | 3.9 | --- | 0.21 | 18 | 15.3 |
| $NO_x$ | 6.0 | 8.1 | 0.9 | 0.5 | 0.2 | 0.6 | 2.7 | 0.04 | 0.11 | 19 | 21 |
| $N_2O$ | 0.41 | 0.58 | 0.08 | 0.11 | --- | 0.02 | 0.15 | 0.00 | 0.016 | 1.36 | 1.31 |
| $NH_3$ | 2.1 | 3.8 | 0.29 | 1.11 | 0.71 | 0.2 | 0.9 | 0.68 | 0.03 | 10.0 | 10.3 |
| HCN | 1.06 | 1.26 | 0.19 | 0.24 | 0.76 | 0.10 | 0.83 | 0.02 | --- | 4.5 | 0.9 |
| $N_2$ | 6.3 | 7.6 | 0.8 | 1.2 | --- | 0.6 | 5.6 | --- | --- | 22 | 26 |
| $SO_2$ | 1.1 | 2.21 | 0.21 | 0.34 | 0.73 | 0.19 | 1.20 | --- | 0.026 | 6.0 | 3.5 |
| COS | 0.06 | 0.14 | 0.01 | 0.03 | 0.02 | 0.01 | 0.04 | --- | --- | 0.31 | 0.27 |
| $CH_3Cl$ | 0.15 | 0.08 | 0.01 | 0.03 | 0.03 | 0.04 | 0.39 | --- | 0.0005 | 0.73 | 0.65 |
| $CH_3Br$ | 0.007 | 0.022 | 0.000 | 0.001 | 0.002 | 0.000 | 0.001 | --- | --- | 0.034 | 0.029 |
| $CH_3I$ | 0.0017 | 0.0196 | 0.0001 | 0.0002 | 0.0021 | 0.0000 | 0.0002 | --- | --- | 0.024 | 0.014 |
| Hg | 0.0001 | 0.0003 | 0.00006 | 0.00010 | --- | 0.00001 | 0.0001 | --- | --- | 0.0007 | 0.0008 |
| $PM_{2.5}$ | 16 | 24 | 5.5 | 8.4 | 3.2 | 2.0 | 14.5 | 3.6 | 0.14 | 77 | 58 |
| TPM | 21 | 31 | 5.5 | 6.9 | 0.0 | 3.1 | 14.9 | 2.5 | 0.09 | 85 | 82 |
| TC | 7.6 | 16 | 2.5 | 4.4 | 2.5 | 1.3 | 7.3 | --- | 0.09 | 41 | 42 |
| OC | 7.3 | 12.8 | 3.3 | 2.7 | 2.4 | 1.2 | 6.6 | --- | 0.10 | 36 | 36 |
| BC | 1.3 | 1.46 | 0.17 | 0.19 | 0.02 | 0.10 | 1.7 | --- | 0.01 | 4.9 | 4.8 |
| K | 0.95 | 0.93 | 0.05 | 0.08 | 0.001 | 0.12 | 0.28 | --- | 0.03 | 2.4 | 1.9 |
| CN | 5.5E+28 | 1.1E+28 | 2.8E+27 | 1.9E+27 | --- | 1.2E+27 | 6.3E+27 | --- | 2.2E+26 | 7.9E+28 | 2.9E+28 |
| CCN (1% SS) | 1.9E+27 | 4.8E+27 | 6.0E+26 | 7.3E+26 | --- | 2.5E+26 | 2.4E+27 | --- | --- | 1.1E+28 | 1.7E+28 |
| N(acc) | 3.0E+27 | 7.9E+27 | 3.0E+26 | 4.5E+26 | --- | 2.4E+26 | 2.1E+27 | --- | --- | 1.4E+28 | 9.0E+27 |

[a]) using EFs from online updates to Akagi et al. (2011)

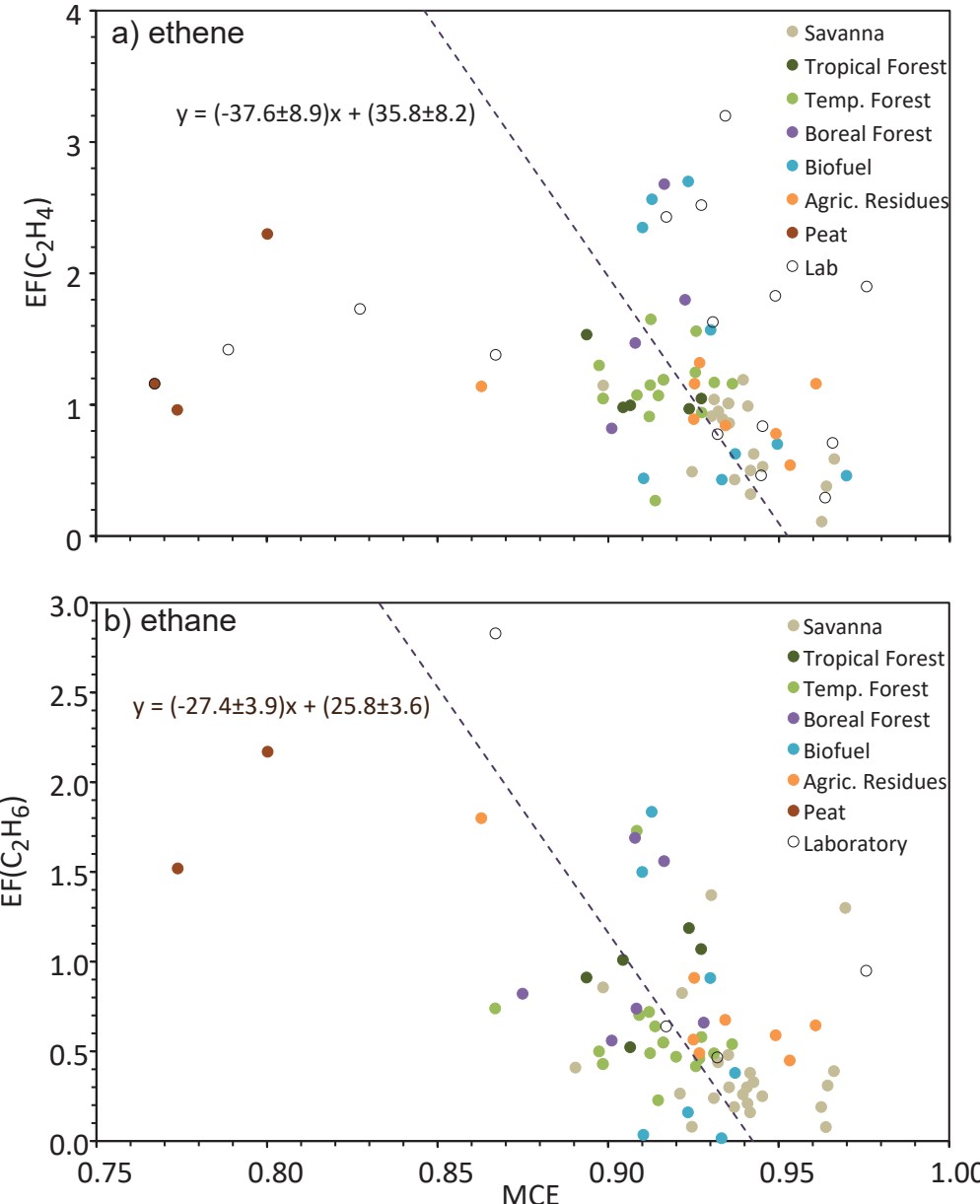

Figure 1: Scatter plots of the emission factors of ethene (a) and ethane (b) against MCE, based on studies in the different combustion categories.

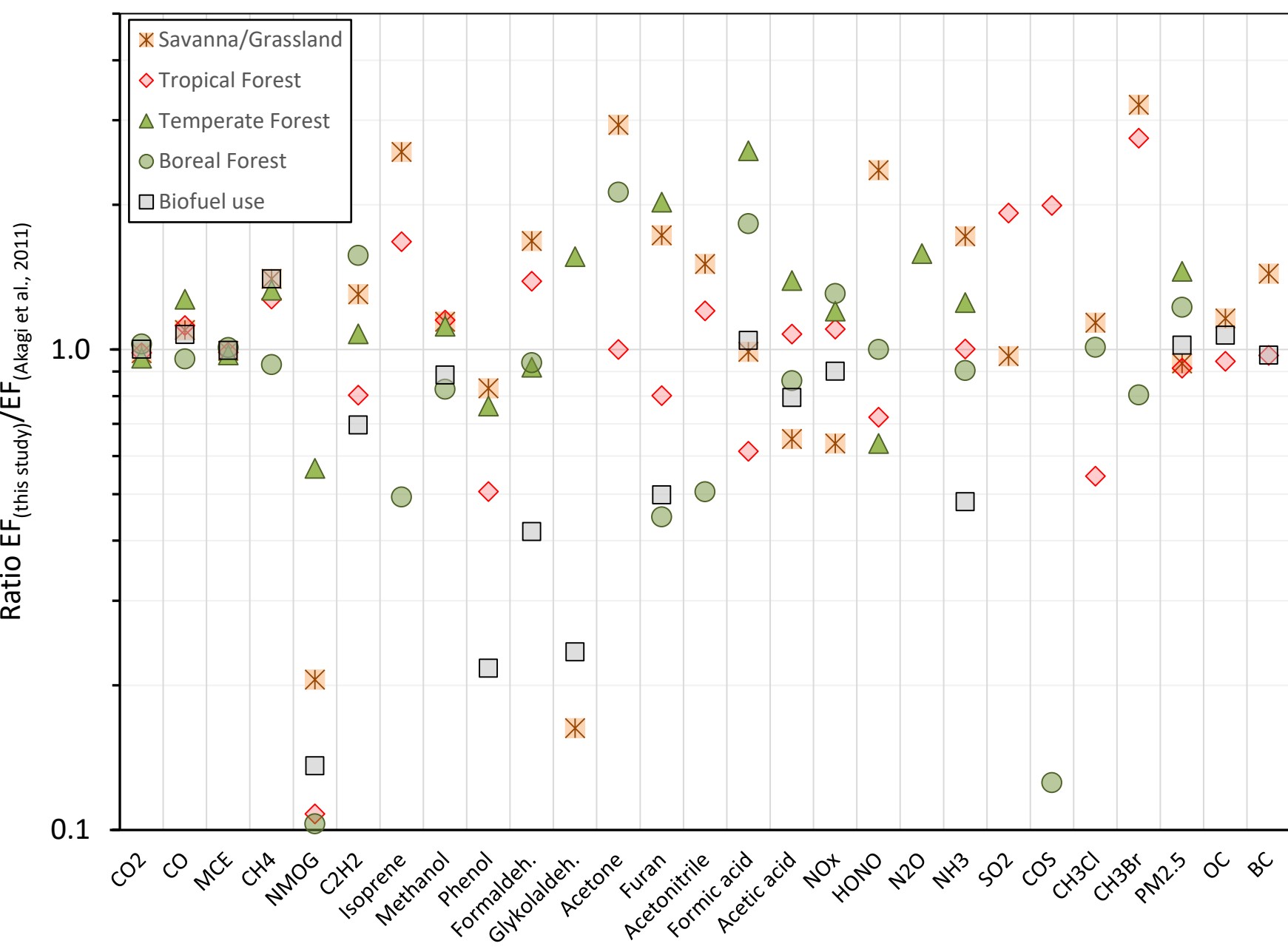

Figure 2: Comparison between the emission factors for selected species between this study and the values in Akagi et el., (2011).