# Peer review of "Emission of trace gases and aerosols from biomass burning – An updated assessment"

_Atmospheric Chemistry and Physics, 2019_

## Referee Comment (RC1) · Anonymous Referee #1 · 24 Apr 2019

This paper summarizes decades of fire emission factor measurements into a table with recommended emission factors for dozens of species. This is a worthy addition to the literature following earlier meta studies by the lead author and by Akagi, Yokelson, and co-authors. A quick look at the reference list also shows that this a testimony to the enormous contribution of dr. Andreae to this research field. I would recommend publication of the paper, but it would benefit from a longer discussion about differences with other meta analyses and recommendations for future research. In addition, I feel the uncertainty in global emission estimates is artificially amplified, please see below

1-21: Fires are obviously a source of $CO_2$, but it would be good to add a statement on whether fires are a net source of $CO_2$ to avoid confusion

2-10: The Johnston et al paper estimated 339,000 annual premature deaths, the num-

ber mentioned here is an interquartile range.

2-22: Please specify the units, C or DM? Also, a range of estimates is not necessarily the same as uncertainty, please see final point below

2-34: To some degree this differentiation was also done by Akagi et al. (2011), would be good to credit them

3-19: I applaud using top-down constraint, but it also makes for blurring the distinction between bottom-up and top-down measurements. For example, it is widely accepted that there is about a factor 3 difference in AOD calculations based on bottom-up and top-down methods (e.g., Kaiser et al., 2012), merging both approaches may hide this issue and thus requires a bit more information on whether and when merging these estimates is appropriate. Also, the author talks about 'appropriate correction methods' but this is not further specified as far as I can see.

One of my main questions is to what degree the approach of this paper ("Ideally, these measurements had been made within minutes after the smoke was released from the fires") differs from that of Akagi ("smoke that has cooled to ambient temperature, but not yet undergone significant photochemical processing"). What does that mean for the number of studies included, what does it mean for the average values, to what degree do ground-based studies (which in general include the smoldering phase) differ from airborne studies which may be biased towards flaming combustion with more pyroconvection, etc? The latter is mentioned in the text (e.g. 6-22) but in the end all measurements are averaged. In general, the modellers which will ultimately use this dataset need to know whether and why these numbers are different from the numbers being used so far. This is a key question but not addressed at all and a table that addresses these differences and potential causes for the most frequently used species would be welcomed

8-26: This is a somewhat surprising statement to me. Differences in bottom-up and top-down results can originate from uncertainty in many parameters, emission factors

being one of them. The standard deviation of CO in boreal and temperate forests is relatively speaking not that much larger than in savannas which to me is not surprising given the large variability in moisture regimes and tree species and density in those forests. I feel it would be more useful to analyze whether there is a difference in ground and airborne studies to say something about RSC.

My other main point of criticism relates to Table 2 and the statement in the conclusions that the uncertainty in biomass burning emissions nowadays is as large as in the 2000s. Table 2 shows various estimates and the large range stems for a substantial part from outliers such as FLAMBE which predict 10 times higher emissions in tropical forests compared to savannas, totally different from for example GFED4 and GFAS1.2 (derived from GFED3). I understand that it is beyond the scope of this paper to assess which one is right but this deserves some explanation, for example using previously mentioned top-down estimates based on CO. Simply combining the 8750 Tg DM in tropical forests from FLAMBE and the CO emission factor (105 g CO per kg DM) indicates CO emissions from this biome alone of 900 Tg CO per year, something not corroborated by inverse estimates and also at odds with the best estimates of deforestation (e.g., Houghton and Nassikas, 2017, https://doi.org/10.1002/2016GB005546).

---

## Referee Comment (RC2) · Charles Ichoku (Referee) · 1 May 2019

The article entitled "Emission of trace gases and aerosols from biomass burning – An updated assessment" by Meinrat O. Andreae, submitted for possible publication in ACP is quite timely and important, as it updates the biomass burning (BB) emission factor (EF) values from the famous Andreae and Merlet (2001) compilation (A&M2001) that has been widely used and referenced by the BB community. This article provides a much-needed and long-awaited update to that compilation, and includes new BB-emitted species that were not previously considered due to lack of measurements. Appropriately, high importance is accorded to data from field measurements, which are more representative of realistic open-air biomass-burning conditions, as opposed to laboratory experiments. Conversion factors have been applied to harmonize certain

similar parameters and their units of measure to render them more compatible and comparable. The Extratropical BB emissions category, which was somewhat confusing in the previous (A&M2001) version has now been separated into Temperate and Boreal forest categories. Open garbage burning is now included, as these are quite significant in certain parts of the world. Original data and references upon which the present compilation is based are being made available in the supplement for the reader to customize the categorization of biomass burning and/or perform further calculations. Referenced literature increased from 130 papers to 350 papers relative to the previous (A&M2001) version. This article is very well written, comprehensive, and succinct, and represents a significant contribution to the biomass burning literature. Therefore, I recommend the publication of this article after the following specific concerns have been addressed.

Specific Concerns: Page 3, Line 15: I believe it is more conventional to refer to the FTIR technique as "spectroscopy" rather than "spectrometry".

Page 6, Line 9: Change "depending of" to "depending on".

Page 9, Lines 5-10: My main concern here is the use of Fire Radiative Power (FRP) as the sole basis to distinguish "top-down" satellite BB emissions methods from "bottom-up". All satellite BB emissions methods described in the article utilize satellite observations (fire-pixel counts, burned area, or FRP) as a way of estimating the biomass burning activity. The use of one or another parameter (FRP or not) does not make a method "top-down" or "bottom-up". Since the driving variable for estimating BB emissions are the factors that convert the activity to emissions (e.g. emission factors, as eloquently discussed in the article), it is the spatio-temporal distribution/configuration of the original input emissions, which went into deriving these EFs that determine whether a method is "top-down" or "bottom-up". If those input emissions were observed at a few locations and limited time periods, and then scaled up globally, the method is "bottom-up". But, if the input emissions were observed globally and regularly, and then scaled down to their sources, it is "top-down", as in the use of satellite-derived aerosol optical depths (AOD) of smoke-dominated aerosols to constrain the "emission coefficients" used to derive the emissions. Bearing this in mind, among the satellite methods described in this article, only QFED and FEER used globally-observed AOD to derive the coefficients that were then used to derive their final BB emission products, and thus may be categorized as "top-down". The others (including GFAS, which is scaled to GFED emissions) used locally-observed BB-emitted constituents to derive emission factors that were then generalized for their global BB emission products, and thus may be categorized as "bottom-up".

Page 9, Line 17: I am concerned about the use of FAO (2015) as the primary reference for a quantitative value, as I am not sure whether FAO (2015) was peer-reviewed. I believe it would be better to find and cite the original (peer-reviewed) source of the 53 Tg/yr estimate reported in FAO (2015).
* * *

---

## Short Comment (SC1) · 5 May 2019

Firstly, I would like to concur with Dr Ichoku that the current contribution represents a timely addition to the biomass burning literature in light of recent contributions by Akagi et. (2011) and Yokelson et al. (2013) both of which were published in Atmospheric Chemistry and Physics. My main criticism with the current contribution involves the activity data that is used to report the final emissions factor in the supplied tables. The authors report the emissions factor per unit of dry fuel consumed (i.e. g/kg dry fuel consumed) whereas I believe that for certain combustion scenarios, especially for charcoal making, it would be more worthwhile to report them as a percentage of total burnt carbon. A fairly recent paper by Surawski et. al. (https://www.nature.com/articles/ncomms11536) demonstrates the biases that are

likely to ensue from neglecting the change in carbon concentration associated with the combustion process. Given the importance of charcoal making in certain parts of the world e.g. Africa, this may be worth revisiting. Apart from referring the authors to my own paper, I find it strange that emissions factors are reported the way they are by Andreae et al. (2019) given that the alternative approach we cite was indeed developed by the Max Planck Institute for Chemistry in the late 1980s/early 1990s (sensu Jürgen Lobert). The benefits of reporting biomass burning emission factor as a percentage of burnt carbon appear clear to my group (and others) so was hoping to see this approach reflected in a revised manuscript.

---

## Referee Comment (RC3) · Robert Yokelson (Referee) · 10 May 2019

Review of Andreae 2019 BB EF

By Bob Yokelson

My colleagues Christine Wiedinmyer and Kelley Barsanti contributed helpful suggestions to this review.

This paper performs a service to the community by copying and sorting large amounts of biomass burning (BB) emission factor (EF) data from other sources into a single document. It also provides a long list of original research papers. The work was done by a top-notch scientist who was among the pioneers of BB research and who, in addition to having a prodigious publication output, has served the fire community as chair of BIBEX and in producing earlier compilations of literature EFs.  It should be published after a few straightforward improvements. First, transcription of a multitude of numbers is actually extremely difficult to do 100% error-free. Second, the manuscript is not comprehensive, nor an educational review, and has just minimal assessment at the global uncertainty level as shown in more detail below. Thus, a more appropriate title would be something like: "Selected literature-average EF for simple biomass burning categories"

Biomass burning is a huge, complex topic and so a full review (or Referee comment) would be equivalent to writing several papers. Due to time constraints, I will only make a quick partial audit of some values that yields some lessons learned and general conclusions, and then point out readily apparent technical issues in the order they appear in the draft.

I will note here that many updated averages calculated now will likely soon be superseded by large-scale recent (WE-CAN, https://www.eol.ucar.edu/field_projects/we-can) or planned work (FIREX-AQ, https://www.esrl.noaa.gov/csd/projects/firex-aq/) of unprecedented scope.

As a reviewer, I thought it incumbent to spot-check the accuracy of at least some numbers from recent important work since data harvesting was the main activity of the paper. I chose to briefly examine selected peat fire entries because the vast majority of the new data are from our papers, papers we contributed to, or a paper I reviewed; so I knew the backstory (and I'm reviewing another peat fire paper also).

The major new work is by Jayarathne et al, Stockwell et al (2 papers), Hatch et al., and Smith et al. All of these papers are used in this review (hereinafter A19) and are also on the Akagi 2011(A11) update website. Thus, one general point is this paper should mention that the A11 assessment has an update website as a community service (http://bai.acom.ucar.edu/Data/fire/). No new global averages for peat fires are computed in A11 primarily because >600 compounds are now identified from peat fires, tropical and temperate peat may burn differently, and a global average is not the only type of desired input. A19 does compute new "snapshot" literature average EF, but based only on tropical peat data, which may or may not be similar to true global averages, but in any case a quick accuracy check was in order.

A number of entries from Stockwell et al 2016 were copied correctly. I was particularly pleased to see that A19 did NOT quote the PM2.5 from Stockwell since it is clearly stated to be a subset of the more extensive PM2.5 data in Jayarathne. This has escaped some readers, so kudos to A19. Next though, I noted that the "BC" entry is actually the "EC" from Jayarathne. EC measurements can be inflated by charring of OC, and the BC by photoacoustic spectroscopy in Stockwell was 0.0055 or ~35 times lower. Also, there was no entry for SSA for peat despite the data in Stockwell et al allowing a reasonable SSA estimate at any wavelength. The EF for SO2 from Tab S3 of Stockwell et al 2015 is probably too high for a global average because it is the only EFSO2 in the study, and SO2 was below detection for most peat fires as revealed by consulting two other tables in the paper. (Factoring in below detection limit data to "averages" is tricky and I will not discuss it in detail here). I checked a handful of NMHCs that were correct, but did note that the sum of 2-methyl-butenes actually included the 3-methyl-butene in "S16", although this is a very minor issue. Is it fair to estimate an error rate from a few spot-checks? I don't know. Overall, this could be a great starting point along with A11, but not using the original material increases the chances of introducing errors!

I also decided to perform a quick check on the formic acid data since the HITRAN parameters for HCOOH were changed by a factor of ~2.2 in 2012, which impacts all orbital and suborbital IR retrievals from before then. In A11 we adjusted all the old data for HCOOH, acetol, and glycolaldehyde based on new IR cross-sections. I randomly chose Yokelson et al., (2003) to see if HCOOH was updated and was surprised to see our formic acid data and nearly all our data from our 2003 paper missing. I found our data in the Sinha et al., (2003) entry where it had also appeared. So I'm glad the data don't appear twice, although it would be easier to trace the source if quoting the original paper. *In any case the old incorrect value is still there.* As an aside, I also noted that Burling et al., 2011 is in the reference list, but the data are not in the spreadsheet, perhaps to avoid duplication?

So again, this is a good resource and a lot of papers were read with some caution per limited spot checks, but users should be encouraged to consult the original work to double-check or trace important values. I think I noted somewhere that A19 has a place to send in corrections so that is a good feature.

This brief dive into the data reveals some general issues that impact the whole paper. A literature average where every study is weighted the same may not be a true global average for many reasons including:

1) No effort is made to weight more modern measurements techniques. For instance A11 preferred thermal optical EC over plain thermal EC. Now we can probably prefer PAS or SP2 BC to any EC or at least be clear about the measurement. E.g. Li, H., Lamb, K. D., Schwarz, J. P., Selimovic, V., Yokelson, R. J., McMeeking, G. R., and May, A.: Inter-comparison of black carbon measurement methods for simulated open biomass burning

emissions, Atmos. Environ., 206, 156-169,
https://doi.org/10.1016/j.atmosenv.2019.03.010, 2019.

2) The values are not weighted by available estimates of relative activity within the category. For instance, some examples:

   a) Clean-burning stoves and dirty open-cooking fires are lumped together and not weighted for the greater prevalence of open-cooking. A11 has separate cooking fire categories for this reason and that should be mentioned in the A19 text. Further, cooking fire studies in labs tend to see different amounts of pollutants than in field studies with some very different results (e.g. Coffey et al., ES&T, 2017, references therein, and references mentioned below).

   b) Wildfires and prescribed fires create a similar amount of emissions in the US annually, but wildfire measurements are much less common in the literature. Wildfire emissions were recently found to differ significantly from prescribed fire emissions (Liu et al., 2017). In practice, A19 included two studies with anomalously large EFPM (up to 4 times the average) that seem to have pulled the temperate forest average to a value in between the most advanced measurements of the wildfire and prescribed fire EFPM. However, separate EFs for prescribed and wildfires has potential to significantly improve air quality modeling.

   c) Crop residue burned in piles is lumped together with crop residue burned loose in the field without the detailed caveats provided in A11 about how drastically the emissions differ between the two burning styles. Some recent papers now estimate how the crop residue is burned (e.g. Lasko et al Environ. Res. Lett. 12 (2017).

   d) Grasses and shrubs are combined as "savannas", but for the dominant moist savanna fires, the fuels are mainly grass and miombo tree leaf litter and then some logs late in the dry season. Some global models assume a pure grassland category. It would be more useful to users of this paper to include separate grassland and shrubland/woody savanna categories.

   e) The studies are not weighted by the amount of sampling: a study sampling 157 whole fires gets the same weight as a study grab sampling one fire (A11 uses weighting).

   f) Some attention is paid to how representative the sampling is, but not a lot.

   g) A global average may be inappropriate for a regional/seasonal application, or regional/seasonal EFs may improve global models. This is alluded to indirectly, but not stressed enough.

Another general issue relates to the most useful reviewer comment we got on A11. What has changed? People are busy and may be curious if changing their model input will matter or which species to double-check in detail. In response we added a figure showing all the large changes for major emissions between A&M2001 and A11. Something similar could be added to this study although the targets are less clear because A11 is updated on web and A&M2001 has been updated by private communication over the years. What has changed between A11 and A19 and the last update widely dispersed by private communication? A11 computed new values for

temperate forest (2014) and savannas (2015), and these are posted on the website. It might be best to compare to the 2015 web update, which includes all the updated averages.

In general A&M2001, A11, and A19 will all be useful resources and highlighting the overlap and complementary strengths will make all these resources more useful to the community. A11 can add A19 to their update page and A19 can do a better job of pointing to A11. Even A&M2001 has some important components (e.g. equations) that are not in A19.

It would be more important to include some assessment of what is new than the global totals in Table 3 if length is an issue. Global totals are/were interesting; especially in the early days of BB research to confirm global importance of BB, but they are less important now. Nearly all fire emissions are too reactive to be well mixed globally and even for relatively inert species such as CO, the location and timing is needed along with amount for inversions. The standard among modelers now is to compare emissions at the regional level.

Another general issue is that the goals and accomplishments of much of the recent BB EF research are not discussed and many new EF results are not included in the tables; even though the papers were used to some extent. At the time of A11 about half the NMOG (by mass) were still unidentified, yet they surely react in real plumes. Tremendous progress has been made in the last 5-7 years with PTR-ToF-MS, 2D-GC-ToF-MS, etc to identify more of the unknowns. In addition, the amount of sampling and especially the sophistication of the instrumentation for sampling of previously undersampled fire types has seen a substantial increase including agricultural fires, wildfires, cooking fires, etc. No amount of measurements can reduce natural variability, but we have nonetheless greatly decreased uncertainty in smoke chemistry, there is an important difference.

Along these lines, no rationale is given for selecting 121 compounds to include out of the 700 plus that have now been measured. No estimate is given of how much additional NMOG is unaccounted for by the A19 tables. These are major issues. The uncertainty in data from carefully-simulated lab fires, especially when scaled to field conditions, is less of a problem than completely ignoring the chemistry of much of the emissions. Other less sweeping issues arise from the apparently ad hoc approach to what data are included. For instance, the sum of all isomers is used for "terpenes" even though some studies speciate the terpenes and they have different reactivity and potential to form SOA. At the same time, lab data for the sum of dimethyl and ethyl amine (same mass) are not quoted and the only data reported provided separate results. It's likely more important to speciate the much more abundant terpenes.

Another critical current issue that is not discussed is measurements of intermediate and semivolatile compounds. These species are important SOA precursors and we need the SOA precursors to get BB-PM and its significant health and climate impacts right. Expert assessment helps because this also gets into the realm where the EF of an SVOC and the EF of organic aerosol can depend strongly on the concentration of the smoke being measured.

Per the other reviews:

I read the comments of Referee #1 and they all seem reasonable.

Ichoku review: I agree with this Referee's important clarification/correction re delineation of top-down and bottom-up, but add a few points. Bottom-up estimates are difficult for many reasons, but top-down is perhaps presented in overly favorable terms and a bit incomplete. Aerosol emissions are not measured globally but estimated on an extensive scale based on column AOD. AOD is reactive and not conserved, and gaps in AOD exist due to clouds, the cloud mask, orbital gaps, extensive time between overpasses, etc., etc. Importantly, attribution of AOD to specific sources is highly uncertain: e.g. plume injection altitudes are not operationally measured, crop waste burning can occur in forest clearings, or cooking fires and crop residue fires occur side by side in Asia where industrial sources, biogenic SOA, and sometimes peat fires also contribute to AOD. Comparing top and bottom is however super helpful. Finally, top-down using CO exists in numerous studies and gets around the "reactive issue" for AOD, but not the other issues although CO sources may be better constrained than AOD sources. The use of CO in inversions is discussed on page 10 when estimating uncertainties in global totals. The potential to use multiple CO sources could be stressed. E.g.

Kopacz, M., Jacob, D. J., Fisher, J. A., Logan, J. A., Zhang, L., Megretskaia, I. A., Yantosca, R. M., Singh, K., Henze, D. K., Burrows, J. P., Buchwitz, M., Khlystova, I., McMillan, W. W., Gille, J. C., Edwards, D. P., Eldering, A., Thouret, V., and Nedelec, P.: Global estimates of CO sources with high resolution by adjoint inversion of multiple satellite datasets (MOPITT, AIRS, SCIAMACHY, TES), Atmos. Chem. Phys., 10, 855-876, 2010.

Further, since the topic of how much biomass burns is included, then another important approach to how much burns is scaling of a-priori bottom-up emissions to match surface and aircraft data and AERONET AOD as in Reddington et al.

Reddington, C. L., Spracklen, D. V., Artaxo, P., Ridley, D. A., Rizzo, L. V., and Arana, A.: Analysis of particulate emissions from tropical biomass burning using a global aerosol model and long-term surface observations, Atmos. Chem. Phys., 16, 11083-11106, https://doi.org/10.5194/acp-16-11083-2016, 2016.

I don't like unspecified "expert judgment." A11 gives several recipes for estimation of unmeasured EF and they recommend trying several. It may be helpful to reference this discussion and clarify which approach(s) were used in A19.

The short comment by Nic Surawski suggests using "burnt carbon" rather than "dry fuel consumed" as the EF basis. The valid underlying issue is that the %C of the fuel may not be the %C of the emissions, which can make the carbon mass balance (CMB) method less rigorous. Neither %C is known in most field studies but in principle char formation causes the %C of the emissions to be lower than the %C of the fuel. On the other hand, Santín et al., (2015) found that

"higher %C" forest fuel components tend to burn with greater completeness, which tends to cause the %C of the emissions to be higher than the %C of fuel. This tends to cancel the impact of char formation on EFs calculated by the CMB.

In charcoal kilns, large pieces of solid charcoal are formed and the large charcoal yield can be measured reasonably accurately. Bertschi et al., (2003b) describe one practical method to adjust the CMB to get EF both per kg wood used and per kg charcoal made.

The situation changes for landscape fires. The charcoal yields are small and the charcoal is manifested mainly as a fine powder mixed in the exported plume or ash layer or a thin black surface layer on otherwise unburned fuel. Further "burnt C" arguably becomes undefined and unmeasurable in practice since some biomass is "affected by the fire" in ways that do not make char. The canopy can be scorched (turned brown by heat from below), creating emissions, but no char. Distillation of stored terpenes in wood occurs at temperatures below those creating char. Deciding what part of a forest was part of the "carbon burned" is not well defined.

On a practical level, there is a large historical database in the literature on fuel consumption, which was estimated as pre-fire minus post-fire biomass. Examples are included in A11 and there is a recent compilation (van Leeuwen et al., 2014). In contrast, there are few to none data for "burnt C" for major fire types. In general though, the impacts on the EF from the CMB is likely less important than the need for more quality measurements of char yields from landscape fires. This needs to be addressed to improve C-cycling estimates since the char is a carbon sink.

References:

Santín, C., S. H. Doerr, C. M. Preston, and G. González-Rodríguez (2015), Pyrogenic organic matter production from wildfires: A missing sink in the global carbon cycle, Global Change Biol., 21(4), 1621–1633, doi:10.1111/gcb.12800.

van Leeuwen, T. T., van der Werf, G. R., Hoffmann, A. A., Detmers, R. G., Rücker, G., French, N. H. F., Archibald, S., Carvalho Jr., J. A., Cook, G. D., de Groot, W. J., Hély, C., Kasischke, E. S., Kloster, S., McCarty, J. L., Pettinari, M. L., Savadogo, P., Alvarado, E. C., Boschetti, L., Manuri, S., Meyer, C. P., Siegert, F., Trollope, L. A., and Trollope, W. S. W.: Biomass burning fuel consumption rates: a field measurement database, Biogeosciences, 11, 7305-7329, https://doi.org/10.5194/bg-11-7305-2014, 2014.

Line by line comments in P, L format

1, 10: "critically evaluated" is probably better as "considered"?

General on abstract include a sentence on how many species changed by e.g. a factor of two since A11?

1, 2: Some carbon cycle people argue that much of the CO2 from fires should not be counted as emissions if the vegetation grows back.

1, 22: A glance at Table 1 seems to show higher EFN2O than I expected. N2O has been found to account for <1% of fuel N while NH3 is a major fate of fuel N. Are the N2O/NH3 ratios in Table 1 high due to including older studies with artifact N2O in canisters? I think not, but worth checking.

1, 22: Insert "BB is the second largest global source of non-methane organic gases (Yokelson et al., 2008, A11)."

2, 3-4: Fire increases locally available P by raising soil pH. See Jordan, C. F. 1985. Nutrient Cycling in Tropical Forest Ecosystems: Principles and Their Application in Management and Conservation. Chichester: Wiley.

2, 5-6: suggest retiring the term "VOCs" and using non-methane organic gases (NMOG) to recognize important gas-phase emissions with intermediate and lower volatility. Cite the following or equivalent:

Robinson, A. L., Donahue, N. M., Shrivastava, M. K., Weitkamp, E. A., Sage, A. M., Grieshop, A. P., Lane, T. E., Pierce, J. R., and Pandis, S. N.: Rethinking organic aerosols: Semivolatile emissions and photochemical aging, Science, 315, 1259–1262, doi:10.1126/science.1133061, 2007.

May, A. A., Levin, E. J. T., Hennigan, C. J., Riipinen, I., Lee, T., Collett, J. L., Jimenez, J. L., Kreidenweis, S. M., and Robinson, A. L.: Gas-particle partitioning of primary organic aerosol emissions: 3. Biomass burning, J. Geophys. Res.-Atmos., 118, 11327–11338, doi:10.1002/jgrd.50828, 2013.

Hatch, L. E., Yokelson, R. J., Stockwell, C. E., Veres, P. R., Simpson, I. J., Blake, D. R., Orlando, J. J., and Barsanti, K. C.: Multi-instrument comparison and compilation of non-methane organic gas emissions from biomass burning and implications for smoke-derived secondary organic aerosol precursors, Atmos. Chem. Phys., 17, 1471-1489, https://doi.org/10.5194/acp-17-1471-2017, 2017.

Hatch, L. E., Rivas-Ubach, A., Jen, C. N., Lipton, M., Goldstein, A. H., and Barsanti, K. C.: Measurements of I/SVOCs in biomass-burning smoke using solid-phase extraction disks and two-dimensional gas chromatography, Atmos. Chem. Phys., 18, 17801-17817, https://doi.org/10.5194/acp-18-17801-2018, 2018.

Jen, C. N., Hatch, L. E., Selimovic, V., Yokelson, R. J., Weber, R., Fernandez, A. E., Kreisberg, N. M., Barsanti, K. C., and Goldstein, A. H.: Speciated and total emission factors of particulate organics from burning western US wildland fuels and their dependence on combustion

efficiency, Atmos. Chem. Phys., 19, 1013-1026, https://doi.org/10.5194/acp-19-1013-2019, 2019.

2, 7: Cite review of O3 formation in BB plumes; Jaffe, D. A., and Wigder, N. L., 2012. Ozone production from wildfires: A critical review. Atmospheric Environment 51, 1–10, doi:10.1016/j.atmosenv.2011.11.063.

2, 7: change "other pollutants" to "secondary PM" or some equivalent term

2, 8: delete "emitted" – these last two changes provide at least minimal recognition that much of the BB-PM impacts are from secondary PM.

2, 9: Akagi et al., (2014) present likely the most comprehensive assessment of toxic gases in fire-line smoke (add to reference string).

2, 16: "disconcerting" perhaps, but given the difficulty of measuring how much BB occurs, not at all surprising.

2, 26: I would add "numerous" before "field" as there are probably too many recent and on-going studies to provide complete references.

2, 27: Most of the EF results can be found in just 2-3 journals. I'd rephrase "The results of these studies are, however, widely dispersed among hundreds of papers in a large number of journals" to "The results of these studies are dispersed among hundreds of papers".

2, 28: add "on a global scale" after "data" since most papers do synthesis/comparison at some scale.

2, 29: define Akagi et al., 2011 as "A11" to facilitate further citation.

2, 30: After "emission factors." insert ~ "I have provided informal updates to A&M2001 and A11 maintains an update website (http://bai.acom.ucar.edu/Data/fire/).

2, 32: Insert "first appeared" after "previous compilations" to make it clear updates have already been readily available.

2, 32: Why 28 out of hundreds of new species?

2, 32: Are any species in A19, but not the original A11? Text should be inserted to clarify that new species were in the updated tables and papers posted on the A11 update site and in informal updates to A&M2001 distributed by the author. Also would be ideal to insert a mention here of recent or planned work that will modify these values, i.e. campaigns I cited above.

2, 33: After "burning types" insert "following A11"

3, 9: since this paragraph paraphrases A11 should add "following A11" before "I only …"

3, 15: Add "solar" before "Fourier" and "spectrometry" should be "spectroscopy"

3, 21: Good place to add that some lab data is adjusted to reflect field conditions using "overlap species", ERs, or MCE as discussed in Yokelson et al., 2013. I think that data appears to have been used.

3, 28: I would change "usually" to sometimes".

3, 31: change "typically" to "may be"

The logic is that some lab studies were carried out in the Missoula Fire Lab using fuels that were locally-collected by forest fuel experts or fuels "Fed-Exed by forest fuel experts. The fires were burned at a scale with flame lengths etc close to real world conditions. Fuel moistures in the FIREX-2016 lab experiment were quite high for example. Canopy fuels sent from the SE US had fuel moistures on the order of 136% on a dry weight basis. Also some lab studies report data adjusted by the MCE, ERs, or field/lab ratio for overlap species (Selimovic et al., 2018; Stockwell et al., 2015; Yokelson et al., 2013; etc). *Most importantly, for a vast number of species, there is only lab data.* To some extent this is clarified on P4, lines 2-4, but these points are important to make consistently in a revised paragraph here.

4, 1: This MCE is of course unrealistic, but not even close to typical of most lab fires.

4, 5-9: This discussion is well done, but some references could be included for the reader interested in more details. The Bond group, for instance, has a number of papers that find lab attempts to replicate field cooking fall short. Stockwell et al 2016a show how MCE dropped off significantly from the lab to field and describe correction factors for the lab data. The risk of lumping all this data together should be clear as noted above.

4, 10-26: This section is good. Should the equation be numbered? Probably apparent that "mixing ratios" can be used interchangeably with "concentrations"?

4, 27: change "easy" to "straightforward"

4, 31: I would change "is readily" to "can sometimes be". The fuel moisture evaporation contributes to mass loss in the lab and fuel moisture is variable within components and between components, which have different combustion factors. We use the carbon mass balance method in the lab, which has the advantage in the lab of minimal distortion of excess CO2 via mixing.

5, 1: We include EC or BC in sum of carbon.

5, 2: More accurately fuel %C tend to be 40-45 for crops and grasses, 50 for wood/foliage, 55+ for peat.

5, 7-12: The equation; number it and check it! What is $EF_{(X/Y)}$?

The equation as presented makes no sense. I think it's trying to say something like:

Say the ER mol/mol of $C_2H_4/CH_4$ was measured as 0.1, but the data needed to compute EF was not collected in the study. If we know from other work that a reasonable guess at $EFCH_4$ is 5, then $EFC_2H_4$ can be estimated as 0.1*(28/16)*5 or 0.875. If this is the intent of the text here, $EF_{(X/Y)}$, which is undefined should be $EF_X$? However, if $EF_Y$ is not known, then it should be made clear this is not the same as a measurement of $EFC_2H_4$, but just an estimation. Thus this discussion, after any needed corrections, from line 7 on, belongs in the discussion of estimates, not under "conversion of units."

5, 17-23: This whole discussion is confusing and may have errors.

If you assume the $EF_X$ is unknown but is proportional to $EF_{CO}$ then that seems to just be suggesting using a corrected version of the equation above with CO as the reference species "Y". If so, then on line 19 $ER_{CO}$ should be $EF_{CO}$, the mass conversion ratio needs to be included, and it makes sense to use the $ER_{(X/CO)}$ from the most similar fuel type available rather than a global average. I.e. crops are grasses so if $ER_{(X/CO)}$ is not known for crops, but is for grasses, use that instead of factoring in the X/CO ratio for e.g. peat and garbage burning.

On lines 19 and 20: what is $ER_X/ER_{CO}$ anyway? Is it just $ER_{(X/CO)}$ used above? If so be consistent – especially since $ER_{CO}$ uses what as a reference species??

Next, for flaming compounds using the consumption weighted average of all categories makes less sense than using the most similar biomass type category as noted just above. Otherwise, the implication is that fire type doesn't matter; inconsistent with the rationale for creating fire type categories in the first place.

Finally, on line 23, what is a subjective best estimate? Some procedure was followed that should be spelled out.

The next four comments are related because smoldering is a combination of distillation, pyrolysis, and glowing combustion; and both glowing combustion and flaming combustion can induce distillation and pyrolysis.

5, 28: change "combustion" to "flaming or glowing".

6, 4-5: I would change "Once most volatile matter is consumed during flaming combustion, the remaining char undergoes gas-solid reactions between oxygen and carbon at the fuel surface, called the smoldering phase" to "In addition to volatile matter being consumed by flaming combustion, char undergoes gas-solid reactions between oxygen and other gases and solid carbon at the fuel surface, called gasification or "glowing" combustion".

Then on 6, 7: change "pyrolysis, flaming, and smoldering combustion" to "flaming and smoldering combustion (vernacular for a changing mix of distillation, pyrolysis, and glowing)"

Also on line 7: since fires can have more than one plume say "the fire plumes at any place and time contain"

6, 15: "peatland" should be "peat" since peatland will have surface fuels that are consumed partially by flaming. Stockwell et al 2016b gives a better overview of how peatland fires play out than Bertschi et al., 2003a and Guillermo Rein's group has published detailed papers on peat combustion dynamics.

6, 18: change "a nocturnal" to "the" and change "serious problems" to "limitations". It is entirely possible for RSC to occur during the daytime and to measure RSC EF using ground-based sampling (e.g. Bertschi et al., 2003a; Christian et al., 2007, Akagi et al., 2013).

6, 19: I would change "will completely miss" to "have trouble measuring"

6, 20: add "or fire blow-ups" after "daytime convection"

6, 20-21: Change "get lost" to "may be distorted by mixing"

6, 21-26: It is not any harder to measure CO/CO2 near the source for RSC than it is for any other source, but it should be done from the ground (see references above). The main problem is the RSC component of fuel consumption is difficult to measure to get a weighted fire average for overall emissions. Thus this paragraph should end with "Ground-based studies of RSC can obtain EFs of trace species, but these are difficult to relate to the corresponding amount of fuel burned." Delete the rest of the paragraph as it is misleading. Refer the reader to Bertschi et al., (2003a) for scenarios of how RSC impacts EF.

6, 32 – 7, 14: This discussion needs to be rewritten from a perspective with more realistic hopes for what MCE can accomplish. Figure 1 throws out almost all available useful data by using one point per study and needs to be deleted or replaced with something useful.

Some general comments followed by specific recommendations: MCE, CO/CO2, BC/CO, and BC/OA are all useful to illustrate how the relative amount of flaming and smoldering can cause BB EFs to vary; especially within a single fuel-/fire-type or study. MCE is most common and as MCE decreases the total products of incomplete combustion increase. The author cites numerous papers with examples of good correlation of EF, even for specific species, vs MCE and this helps make sense of the observed variability and might drive a model at a useful scale (TBD). Low MCE dependence can be "OK" too and can sometimes increase confidence that the average value is close to correct for a range of burning conditions (Table 4, Liu et al., 2016). CO is the indicator of smoldering, but smoldering is a dynamic mix of complex processes and a simple parameter based on two gases should not be expected to predict all the outcomes of thousands of relevant chemical reactions across the planet for all emitted species. On broad scales other factors like fuel type (as noted), fuel N (Burling et al., 2010), geometry (Bertschi et al., 2003a), weather, etc impact emissions and correlations decrease as more conditions are considered.

Every model has a scope and every model has limitations. There might be a user-specific scale/scope where the coverage and correlation of an EF vs MCE model are both adequate to improve emissions estimates. Figure 1 skips over that question, throws out the data, and just demonstrates the obvious conclusion that EF vs MCE is not universal. The proper next step in evaluating EF vs MCE is to compare slopes based on all the data in the original studies aggregated at some intermediate level. To illustrate what I mean I insert a table where that process is started:

| | Fire Type | savanna | | Trop for | | Conifer | | Eucalypt | |
|---|---|---|---|---|---|---|---|---|---|
| | Study | Yok-03 | | Yok-07 | | Burl-11 | | Reis-18 | |
| Species | | slope | r^2 | slope | r^2 | slope | r^2 | slope | r^2 |
| CH4 | | -48 | 0.87 | -47 | 0.52 | -96 | 0.94 | -96 | 0.93 |
| CH3OH | | -21 | 0.8 | -15 | 0.48 | -40 | 0.98 | nm | nm |

A glance at the table suggests some potential for a "fire-type-specific" EF vs MCE model with the level of correlation and aggregation perhaps depending on species also. I have not pursued this due to lack of time and because MCE is not available operationally as fire model input anyway. For now MCE remains most useful as a way to partially deconstruct variability in reported EF data.

With the above discussion as background I suggest the following revisions at a minimum.

7, 3: change "unfortunately" to "however"

7, 4: change "general parameterization of EFs" to "global parameterization of all EFs"

7 4-7: delete "As an illustration, I show in Fig. 1a and 1b plots of the EFs of ethene (C2H4) and ethane (C2H4) vs MCE, based on the studies in the supplemental spreadsheet. In both cases, the results scatter widely, and especially the data from the lab studies, biofuel burning, peat fires, and RSC-dominated fires introduce a large amount of scatter."

Fig. 1 is one point per study rather than comparing slopes using multiple points per study, which might tell a different story and preserves whatever information there is.

7, 7-8: change "The poor correlation between EFs and MCE has been noted previously" to "The limitations of EFs versus MCE have been noted previously"

7, 8-11: delete "In the case of ethene, the correlation using all data points is not significant (R2 = 0.07). However, when only the data from open vegetation fires are included (and after removing three outliers), the correlation improves to an R2 of 0.27. For ethane, the correlation coefficient is R2 = 0.38 for all data, but does not improve substantially by removing the peat fire data."

7, 11-12: change "These results suggest the potential of using MCE as a meaningful, but rough predictor of EFs for at least some species." To "The level of aggregation at which MCE is useful

as a meaningful, but rough predictor of EFs for at least some species has not yet been determined."

7, 13: change "supplement" to "original studies"

7, 14: insert "A new approach to modeling NMOGs from pyrolysis using PMF has potential (Sekimoto et al., 2018); especially if the factors can be related to operationally available input."

Sekimoto, K., Koss, A. R., Gilman, J. B., Selimovic, V., Coggon, M. M., Zarzana, K. J., Yuan, B., Lerner, B. M., Brown, S. S., Warneke, C., Yokelson, R. J., Roberts, J. M., and de Gouw, J.: High- and low-temperature pyrolysis profiles describe volatile organic compound emissions from western US wildfire fuels, Atmos. Chem. Phys., 18, 9263-9281, https://doi.org/10.5194/acp-18-9263-2018, 2018.

7, 15-20: This may be worth trying, but model estimates of fuel consumption by flaming and smoldering would be difficult to validate in the field since access during the fire is problematic. Also the MCE of flaming or smoldering can vary broadening predicted MCEs.

7, 20: The first paper probing the relationship between greenness and MCE was Hoffa et al., 1999. Hoffa, E. A., D. E. Ward, W. M. Hao, R. A. Susott, and R. H. Wakimoto (1999), Seasonality of carbon emissions from biomass burning in a Zambian savanna, J. Geophys. Res., 104, 13,841–13,853.

Korontzi et al., 2003 updated the MCE/Greenness relationship based on new MCE measurements and then combined measured MCE, MCE vs greenness, and EF vs MCE (from other work in the late dry season) to estimate early dry season OVOC EFs.

7, 22: In addition to Korontzi et al., 2005, greenness (PGREEN) was used to predict combustion completeness in Korontzi et al., 2004 and PGREEN was used to predict MCE by Ito and Penner, 2004 (https://agupubs.onlinelibrary.wiley.com/doi/full/10.1029/2003JD004423).

Korontzi et al., Modeling and sensitivity analysis of fire emissions in southern Africa during SAFARI 2000, Remote Sensing of Environment 92, 255–275, 2004.

This approach has potential, but so far has been used for savannas only and works best for species that correlate strongly with MCE. The results have not been tested with field measurements to my knowledge. The discussion might be revised slightly.

7, 22: Maybe wrap up this section with something like "For now we should use the average EFs, but be aware they can vary considerably fire to fire."

7, 27: Not sure what this means "The averages in this column can only be seen as general indications, since all types of fuels and burning methods are included,"

Pages 7-8 in general: A19 has adopted some of improvements of A11, which is good.

8, 4: after "category" it could be useful to cite this resource of garbage burning activity and EF: Wiedinmyer, C., Yokelson, R. J., and Gullett, B. K.: Global emissions of trace gases, particulate matter, and hazardous air pollutants from open burning of domestic waste, Environ. Sci. Technol., 48, 9523-9530, doi:10.1021/es502250z, 2014.

8, 21-22: An EF for particle number concentration is problematic and potentially meaningless or misleading due to rapid coagulation near sources! Warning label needed.

8, 23: EFs for "brown carbon" (BrC) as g/kg are problematic because there are likely hundreds of contributing trace components with different absorption cross-sections that are also evidently reactive. But there is BrC emissions data in the form of Ångström absorption exponents (AAE) and BrC absorption EFs (as $m^2$/kg following the Bond and Moosmüller groups) in the UV for fresh emissions from carefully simulated lab fires and numerous field fires for different BB types (Stockwell et al., 2016a, b; Goetz et al., 2018; etc). Total absorption EFs in the UV are also given for users who may prefer them.

To clarify misleading text: the discussions in Selimovic et al., (2018 and 2019) show AAE near 3.7 (field Forrister et al., 2015) and 3.3 (lab Selimovic et al., 2018) for fresh smoke, but decaying with age as shown in Forrister et al and with BrC accounting for ~50% of absorption at 401 nm in *"moderately aged"* smoke (Selimovic et al., 2018). Most of these papers are in the A19 tables, but BrC data, which is important as the author says, is not tabulated in general.

Forrister, H., Liu, J., Scheuer, E., Dibb, J., Ziemba, L., Thornhill, K. L., Anderson, B., Diskin, G., Perring, A. E., Schwarz, J. P., Campuzano-Jost, P., Day, D. A., Palm, B. B., Jimenez, J. L., Nenes, A., and Weber, R. J.: Evolution of brown carbon in wildfire plumes. Geophys. Res. Lett., 42, 4623–4630, https://doi.org/10.1002/2015GL063897, 2015.

Goetz, J. D., Giordano, M. R., Stockwell, C. E., Christian, T. J., Maharjan, R., Adhikari, S., Bhave, P. V., Praveen, P. S., Panday, A. K., Jayarathne, T., Stone, E. A., Yokelson, R. J., and DeCarlo, P. F.: Speciated online PM1 from South Asian combustion sources – Part 1: Fuel-based emission factors and size distributions, Atmos. Chem. Phys., 18, 14653-14679, https://doi.org/10.5194/acp-18-14653-2018, 2018.

The Goetz paper above and Jayarathne papers cited include data for ions and metals in PM. Major ions and metals are tabulated in A11, but not A19, a point worth making in A19.

8, 26-33: I would delete this paragraph or at least revise it extensively.  In part because the "most" serious problem is subjective depending on the workers area. For instance, top-down estimates of BB are probably most concerned with the issues such as observational constraints I outlined in my general comment on top-down estimates above. Workers looking at SOA may care more about EFs for SVOC, etc. In general this represents the authors troubles measuring RSC from an aircraft and other issues could lead to the underestimates of regional CO emissions mentioned. Also, it's misleading because RSC does not affect only tropical forest fires. RSC

accounts for a significant part of the emissions for all forest fires, pasture fires, and wooded savanna, and virtually all the emissions from peat fires for example. However, the situation is far from hopeless. Bertschi et al., (2003a) outlined a range of impacts when RSC accounts for 10% to 50% of the total fuel consumption in a fire. At the upper end with 50% of fuel consumption by RSC the CO2 and CO EF changed by about -7% and +13% respectively. The larger impacts of RSC are for other gases like NH3 and CH4. Further, in A11 the tropical forest EF were adjusted based on an assumed RSC component of just 5% per available evidence at the time.

9, 5: This discussion doesn't include all fire inventories so change "Three of them use a bottom up approach" to "Four of them (for example) use a bottom up approach"

9, 7: change "The other three products are top-down, based on fire radiative power (FRP):" to "Two other products are top-down:" since GFAS is bottom-up, FRP is still just based on hotspots, and (for example) Ron Cohen's group (Mebust et al) also has a top-down approach.

9, 9: Agree with Charles Ichoku, GFAS is bottom-up. In this section on how much biomass is burned it could help to foreshadow the later discussion of CO inversions, list sources of uncertainty, and the other issues I noted in my general comments above.

9, 25: Are global numbers for reactive gases still important? More important than Table 3 might be to include a summary of what is new in this compilation as discussed in my general comments.

9, 28 "the previous assessment" should be "A&M2001" since there are so many global estimates.

9, 30-32: The fire to fire variability and even real day to day variability for a single fire can be much higher than the standard deviation of the literature mean. This can be important in many modeling applications (Yates et al., 2016). Change to "global emissions uncertainties" on line 32.

10, 1-14: This discussion is useful and adds confidence to global totals. There is a large body of work in this area and I have not attempted a comprehensive critique, but like the idea of using multiple CO products as noted above.

10, 15: A11 also reported these differences so useful to change to "As noted in A11, major …"

10, 20-22: I would rephrase this to say that there has been good progress in OVOC and HCN emissions as just noted and in reducing the percentage of un-identified compounds, sampling under-sampled sources, measuring I/SVOC, and sampling post-emission evolution, but quantifying global activity levels remains difficult. This is to be expected due to clouds, orbital gaps, small fires, unknown injection altitudes and diurnal cycles, etc. More measurements can add info but not reduce natural variability. Measuring EF and quantifying biomass burned present a different set of challenges. Most model inputs cannot be measured operationally. Thus, the author's proposed CO inversions are just one idea.

10, 29: Table 1 doesn't include the major new research front in I/SVOC when it comes to setting future priorities.

11, 6: The conclusions remain focused on the problem of estimating global totals, which is just one part of BB research. It may not be the most important part, but is probably the hardest. Bottom-up or top-down models are super-sensitive to plume injection altitude, terrain flattening, diurnal cycles, complex transport, and chemical/physical evolution; often at subgrid scales. These things cannot be measured operationally. Actual recent/upcoming work such as WE-CAN and FIREX-AQ focus instead on advanced instrumentation and combining an unprecedented scope of airborne and ground-based measurements with new satellite products. This will eventually also be helpful to estimating global totals.

In summary, a quick check identifies some significant improvements needed in the paper. This is all I have time for now, but I may comment again before the discussion closes.

---

## Short Comment (SC2) · 15 May 2019

The Author is to be commended for taking on the task of assembling, digesting, and tabulating reported biomass burning emission factors (EFs). However, the author's large-scale dismissal of laboratory studies that present EFs seems limiting and un-justified. The reasoning of the author seems to be based mostly on one completely unrealistic study, but many other studies have shown how to harmonize the lab studies with field studies including Yokelson et al., (2013), Stockwell et al., (2015), and Se-limovic et al., (2018). While some of these studies have been used, the methodology there-in could have been applied to many other recent state-of-the-art studies. Instead the fuel-specific info for a wealth of important compounds not measured in the field is lumped into a large "lab average category." Over reliance on field data mean that fast

chemistry may not be captured properly, e.g. HONO, and perhaps other fast actors such as 2,3-butanedione, especially for large fires that could not be sampled close-up. In addition, this arbitrary decision has resulted in the Author missing (omitting) several new developments in BB emissions measurements. The work of Sekimoto et al. 2018 that found that most of the variability in VOC emissions was explained by just two factors, related to low and high temperature pyrolysis, and that are valid for a variety of fuels, was not mentioned. No mention was made of the importance of isocyanic acid (HNCO), an N-compound of emerging health interest (Roberts et al., 2011) and which Koss et al., (2018) have shown to often be more abundant than HCN in laboratory fires. These new features/results do not yet have field measurements of EFs to back them up, but soon will. Strangely, 25+ year old laboratory results from the Mainz group were included in several cases in the main table (Table 1), while important new results were overly consolidated and relegated to the SI spreadsheet (e.g. the I/SVOC work of Hatch et al., 2015), or not listed at all (HNCO; Hatch et al., 2017; 2018).

References

Hatch, L. E., Luo, W., Pankow, J. F., Yokelson, R. J., Stockwell, C. E., and Barsanti, K. C.: Identification and quantification of gaseous organic compounds emitted from biomass burning using two-dimensional gas chromatography/time-of-flight mass spectrometry, Atmos. Chem. Phys., 15, 1865-1899, 2015.

Hatch, L. E., Rivas-Ubach, A., Jen, C. N., Lipton, M., Goldstein, A. H., and Barsanti, K. C.: Measurements of I/SVOCs in biomass-burning smoke using solid-phase extraction disks and two-dimensional gas chromatography, Atmos. Chem. Phys., 18, 17801-17817, 2018.

Hatch, L. E., Yokelson, R. J., Stockwell, C. E., Veres, P. R., Simpson, I. J., Blake, D. R., Orlando, J. J., and Barsanti, K. C.: Multi-instrument comparison and compilation of non-methane organic gas emissions from biomass burning and implications for smoke-derived secondary organic aerosol precursors, Atmos. Chem. Phys., 17, 1471-1489,

2017.

Koss, A. R., K., S., Gilman, J. B., Selimovic, V., Coggon, M. M., Zarzana, K. J., Yuan, B., Lerner, B. M., Brown, S. S., Jimenez, J. L., J., K., Roberts, J. M., Warneke, C., Yokelson, R. J., and de Gouw, J.: Non-methane organic gas emissions from biomass burning: identification, quantification, and emission factors from PTR-ToF during the FIREX 2016 laboratory experiment, Atmos. Chem. Phys., 18, 3299-3319, 2018.

Roberts, J. M., Veres, P. R., Cochran, A. K., Warneke, C., Burling, I. R., Yokelson, R. J., Lerner, B. M., Gilman, J. B., Kuster, W. C., Fall, R., and de Gouw, J.: Isocyanic acid in the atmosphere and its possible link to smoke-related health effects, PNAS, 108, 8966-8971, 2011.

Sekimoto, K., Koss, A. R., Gilman, J. B., Selimovic, V., Coggon, M. M., Zarzana, K. J., Yuan, B., Lerner, B. M., Brown, S. S., Warneke, C., Yokelson, R. J., Roberts, J. M., and de Gouw, J.: High- and low-temperature pyrolysis profiles describe primary emissions of volatile organic compounds from western US wildfire fuels, Atmos. Chem. Phys., 18, 9263-9281, 2018.

Selimovic, V., Yokelson, R. J., Warneke, C., Roberts, J. M., deGouw, J. A., Reardon, J., and Griffith, D. W. T.: Aerosol optical properties and trace gas emissions by PAX and OP-FTIR for laboratory-simulated western US wildfires during FIREX, Atmos. Chem. Phys., 18, accepted for discussion, 2018.

Stockwell, C. E., Veres, P. R., Williams, J., and Yokelson, R. J.: Characterization of biomass burning smoke from cooking fires, peat, crop residue and other fuels with high resolution proton-transfer-reaction time-of-flight mass spectrometry, Atmos. Chem. Phys., 15, 845-865, 2015.

Yokelson, R. J., Burling, I. R., Gilman, J. B., Warneke, C., Stockwell, C. E., de Gouw, J. A., Akagi, S. K., Urbanski, S. P., Veres, P., Roberts, J. M., Kuster, W. C., Reardon, J., Griffith, D. W. T., Johnson, T. J., Hosseini, S., Miller, J. W., Cocker III, D. R., Jung,

H., and Weise, D. R.: Coupling field and laboratory measurements to estimate the emission factors of identified and unidentified trace gases for prescribed fires, Atmos. Chem. Phys., 13, 89-116, 2013.

---

## Author Comment (AC1) · 31 May 2019

Response to Comment by Nic Surawski

I thank Nic Surawski for his positive and interesting comments. My responses are detailed below. (Suwawski's comments in italics).

*Firstly, I would like to concur with Dr Ichoku that the current contribution represents a timely addition to the biomass burning literature in light of recent contributions by Akagi et. (2011) and Yokelson et al. (2013) both of which were published in Atmospheric Chemistry and Physics. My main criticism with the current contribution involves the activity data that is used to report the final emissions factor in the supplied tables. The authors report the emissions factor per unit of dry fuel consumed (i.e. g/kg dry fuel consumed)...*

Actually, all emission factors in this paper are in units of g/kg of dry mass burned. This is stated in the footnote to Table 1. It is now also stated explicitly in section 2.3.

*... whereas I believe that for certain combustion scenarios, especially for charcoal making, it would be more worthwhile to report them as a percentage of total burnt carbon. A fairly recent paper by Surawski et. al. (https://www.nature.com/articles/ncomms11536) demonstrates the biases that are likely to ensue from neglecting the change in carbon concentration associated with the combustion process. Given the importance of charcoal making in certain parts of the world e.g. Africa, this may be worth revisiting.*

The problem here is that while it might be desirable to do so, the information needed is generally not available. In most studies, the amount of carbon or fuel burned is inferred from the mass balance of measured atmospheric carbon species, while assuming a fuel carbon content. Converting to carbon burnt would require an additional assumption about the yield of char. Furthermore, most users of the EF data apply them to activity estimates in units of dry fuel burnt or consumed per unit time. Providing the EFs in units of carbon burnt would require me to make additional assumptions, and subsequently the users to reverse these assumptions or make additional ones to calculate emissions. For a more detailed discussion, please refer to the review of this paper by Yokelson (https://doi.org/10.5194/acp-2019-303-RC3).

*Apart from referring the authors to my own paper, I find it strange that emissions factors are reported the way they are by Andreae et al. (2019) given that the alternative approach we cite was indeed developed by the Max Planck Institute for Chemistry in the late 1980s/early 1990s (sensu Jürgen Lobert). The benefits of reporting biomass burning emission factor as a percentage of burnt carbon appear clear to my group (and others) so was hoping to see this approach reflected in a revised manuscript.*

Lobert's work was based on laboratory measurements, where he had the benefit of being able to make a complete budget of fuel carbon, char residue, and emissions to the air. This information is not available in most field studies.

---

## Author Comment (AC2) · 31 May 2019

Response to Comment by James Roberts

I thank James Roberts for his positive and interesting comments. My responses are detailed below. (Roberts's comments in italics).

*The Author is to be commended for taking on the task of assembling, digesting, and tabulating reported biomass burning emission factors (EFs). However, the author's large-scale dismissal of laboratory studies that present EFs seems limiting and unjustified. The reasoning of the author seems to be based mostly on one completely unrealistic study, but many other studies have shown how to harmonize the lab studies with field studies including Yokelson et al., (2013), Stockwell et al., (2015), and Selimovic et al., (2018). While some of these studies have been used, the methodology there-in could have been applied to many other recent state-of-the-art studies. Instead the fuel-specific info for a wealth of important compounds not measured in the field is lumped into a large "lab average category."*

The decision to rely (with few exceptions) on field studies was made after careful consideration and comparison of lots of field and lab data. The "unrealistic" study that Roberts refers to is but one extreme example. There are just so many variables that are important in field fires, which cannot be reproduced in the lab, e.g., fuel moisture, air flow, spread rate, etc., but which influence emissions. In addition to the papers mentioned by Roberts, the studies by Christian et al. (2003) and Burling et al. (2011) analyze the differences between field and lab measurements. Including lab studies would have required me to make many, more or less arbitrary, decisions about which lab studies are "realistic" and which not, and what kind of corrections to apply to other scientists' data. The required correction factors are often large (see Yokelson et al., 2013) and for most studies, the information required to make such corrections would not have been available.

*Over reliance on field data mean that fast chemistry may not be captured properly, e.g. HONO, and perhaps other fast actors such as 2,3-butanedione, especially for large fires that could not be sampled close-up.*

Indeed, this is a shortcoming, which is severe for some fast reacting species. For other species, which are very prone to surface uptake, e.g., HCl, open-path lab studies may be the only way to obtain realistic emission measurements (Burling et al., 2011). One can hope that improved measurement techniques may provide in the future the possibility to measure these substances close to sources in the field. However, in this author's judgement, the alternative of making more or less arbitrary decisions about which lab data to blend in for which species, and how to assign them to the combustion categories used in the paper, would not provide enough benefits to justify the loss of a clear criterion for data selection. To give readers the option to use lab data for specific purposes, I have included all lab studies that I am aware of in the Supplement. This will help them find the original papers and evaluate them before using these values for their specific purposes.

*In addition, this arbitrary decision has resulted in the Author missing (omitting) several new developments in BB emissions measurements.*

I object to the use of "arbitrary" here. The decision to base this study on field studies and to include lab measurements only in the form of a general column in Table 1, but with full documentation in the supplement, was taken after careful study of the literature and deliberation of the alternatives. I am very familiar with the studies mentioned by Roberts, and will discuss them below.

*The work of Sekimoto et al. 2018 that found that most of the variability in VOC emissions was explained by just two factors, related to low and high temperature pyrolysis, and that are valid for a variety of fuels, was not mentioned.*

This study is based on lab measurements and does not report explicit emissions data, and is thus not included in the data base for Table 1. However, I agree with Roberts that it provides a novel and interesting approach to generalizing emissions data and have now included a sentence referencing it in Section 3.1: "An interesting and novel approach to generalizing VOC emissions is provided by Sekimoto et al. (2018), who showed that most of the variability in VOC emissions measured in a lab study using a wide variety of fuels was explained by just two factors, related to low and high temperature pyrolysis."

*No mention was made of the importance of isocyanic acid (HNCO), an N-compound of emerging health interest (Roberts et al., 2011) and which Koss et al., (2018) have shown to often be more abundant than HCN in laboratory fires. These new features/results do not yet have field measurements of EFs to back them up, but soon will.*

Regrettably, future measurement cannot be included. I am aware of the potential importance of HNCO, but have not found enough field-relevant data to justify inclusion at this time. But, as stated in the conclusions, I am continuously updating the data base and Table 1, and would be happy to include this species once enough data are provided. I encourage Roberts and other interested readers to supply me with relevant data on this and other substances in the future.

*Strangely, 25+ year old laboratory results from the Mainz group were included in several cases in the main table (Table 1), while important new results were overly consolidated and relegated to the SI spreadsheet (e.g. the I/SVOC work of Hatch et al., 2015), or not listed at all (HNCO; Hatch et al., 2017; 2018).*

I don't quite understand this comment. For one, Table 1 does not contain primary data from specific studies, but only aggregated data. Then, the fact that data are 25+ years old does not qualify them for exclusion from the data base.

The Hatch et al. (2015) paper is an excellent study, but based on lab data only, and was therefore included in the lab studies section of the supplement. The Hatch et al. (2018) paper is a methods intercomparison and synthesis of data that had already been published in previous studies, therefore it had not been included. I made efforts not to include the same data published twice or more in the database used for the specific combustion types in Table 1. However, I have now included it in the lab studies section. The Hatch et al. (2018) study on I/SVOC emissions, while very interesting, did not contain species that I had considered for inclusion in Table 1. In view of the potential importance of these compounds, I have added a sentence and reference in the Conclusions: "Another example are the emissions of semivolatile and intermediate-volatile compounds (I/SVOCs), which are important in the context of organic aerosol from biomass burning, but for which at this time only laboratory measurements are available (Hatch et al., 2018)."

---

## Author Comment (AC3) · 31 May 2019

Response to Reviewer 1

I thank the reviewer for his/her constructive comments. My responses are detailed below. (Reviewer comments in italics).

*1-21: Fires are obviously a source of CO2, but it would be good to add a statement on whether fires are a net source of CO2 to avoid confusion*

This is an extremely complex subject that requires comprehensive earth system models to address and goes far beyond the scope of this paper, which is focusing on gross emissions. To give the reader some indication of the magnitude of fire contributions to atmospheric $CO_2$, I have included the following sentences: "While a significant fraction of the emitted $CO_2$ is taken up again by vegetation regrowth, much of it remains in the atmosphere for years and potentially even up to centuries, e.g., in the case of tropical deforestation fires or peat soil burning (van der Werf et al., 2017). Model simulations suggest that in the absence of fires, atmospheric $CO_2$ concentrations would be about 40 ppm lower, indicating the importance of fires for the atmospheric carbon budget (Ward et al., 2012)."

*2-10: The Johnston et al paper estimated 339,000 annual premature deaths, the number mentioned here is an interquartile range.*

The original text says "…accounting for as much as 600,000 premature deaths per year globally", indicating that this number is an upper limit. It is not the interquartile range, which would be the difference between upper and lower quartile. In response to the reviewer, and to make things fully transparent, I have changed the text to "…accounting for up to 600,000 premature deaths per year globally (75th percentile of model estimates; Johnston et al., 2012)".

*2-22: Please specify the units, C or DM? Also, a range of estimates is not necessarily the same as uncertainty, please see final point below*

The text here does not discuss biomass or carbon burned, which could be expressed as C or DM. Rather, it discusses carbon released, which could be expressed as C or $CO_2$. However, the sentence here is completely unambiguous: "…the estimates in these studies of the annual amounts of **carbon** released still range over a factor of three from 1.5 to 4.7 Pg a$^{-1}$." The rules of the SI system explicitly forbid including the use of constructs such as "4.7 Pg C a$^{-1}$", and suggest including this information in the text, as I have done here.

I did not state that the estimate range is the same as uncertainty. Rather, in the introductory sentence to this paragraph I state "… large uncertainties persist regarding the amounts emitted…". This is illustrated in the following text by wide range of current estimates, which I feel clearly supports the existence of large uncertainties, without equating the range of estimates with the range of uncertainty.

*2-34: To some degree this differentiation was also done by Akagi et al. (2011), would be good to credit them*

Sentence on 2-34 modified to "…the other by Akagi et al. (2011), which included more recent data and additional species and burning types..."

*3-19: I applaud using top-down constraint, but it also makes for blurring the distinction between bottom-up and top-down measurements. For example, it is widely accepted that there is about a factor 3 difference in AOD calculations based on bottom-up and top-down methods (e.g., Kaiser et al., 2012), merging both approaches may hide this issue and thus requires a bit more information on whether and when merging these estimates is appropriate. Also, the author talks about 'appropriate correction methods' but this is not further specified as far as I can see.*

It was not an easy decision for me whether to include these data from remote sensing or not. In the end, I felt that most users of my compilation would be better served if this information was included, since it provided large-scale coverage and data for some species poorly sampled by other techniques, e.g., COS. Users who do not want these data included for particular reasons can easily eliminate them by using the spreadsheet in the supplement and removing the corresponding lines. I did not include any aerosol data in these estimates, as I do not feel that there is any way to address post-emission changes to reconstruct aerosol emission data from large-scale AOD measurements. The data included are only for species that are either long-lived on the time/space scales in question (e.g., COS and HCN), or for which emissions can be reconstructed with some confidence by the use of chemical modeling, e.g., ethane. The "appropriate methods" are actually already discussed in the preceding sentence "…included a correction for atmospheric transformations, using model calculations involving transport times and reaction rates of the species concerned." I have made this more explicit by changing the sentence to: "…appropriate correction methods (i.e., chemistry-transport model calculations to correct for atmospheric transformations)". Readers interested in the details for each paper would have to go back to the original papers, as going into specifics here would not be of interest to most readers.

*One of my main questions is to what degree the approach of this paper ("Ideally, these measurements had been made within minutes after the smoke was released from the fires") differs from that of Akagi ("smoke that has cooled to ambient temperature, but not yet undergone significant photochemical processing"). What does that mean for the number of studies included, what does it mean for the average values, to what degree do ground-based studies (which in general include the smoldering phase) differ from airborne studies which may be biased towards flaming combustion with more pyroconvection, etc? The latter is mentioned in the text (e.g. 6-22) but in the end all measurements are averaged. In general, the modellers which will ultimately use this dataset need to know whether and why these numbers are different from the numbers being used so far. This is a key question but not addressed at all and a table that addresses these differences and potential causes for the most frequently used species would be welcomed*

In the end, what both Akagi et al. and I want to represent is "fresh smoke", i.e., the material that is emitted by the fire and that can serve as starting point for model calculations at scales typically used by a variety of models. In that sense, there is no difference between my approach and Akagi's. Ideally, one would have an objective criterion, such as Akagi's "cooled to ambient temperature", but in reality I had to look at the data in some 400 papers, and had to decide for each species and each set of measurements whether in my judgement they represented "fresh smoke". This led to the exclusion of numerous studies, of some shorter-lived species from some studies from which somewhat longer-lived species could be included, and the exclusion of some flights or samplings from studies where both fresh and somewhat aged plumes had been investigated. I don't think there is any reasonable way to discuss these issues for each of 370 papers (now included in the revised version) and 121 species. In the end, I can only ask the user of these data to trust my judgement, gained over working on this subject for almost 40 years, about which data to include. Averaging over as many data (that are judged to be valid) as possible should reduce biases that result from the inclusion of any atypical data.

The issue of airborne vs. ground-based studies has been discussed extensively in the literature, including A&M2001, A2011, Burling et al. (2011), and briefly in this paper. I did not feel that there is anything new that I could contribute here other than highlighting the problems resulting from this issue. In response to the reviewer, I added the reference to Burling et al. (2011) on p. 5 and a sentence to the last paragraph of section 3.2: "A representative measurement of fire-average $\Delta CO/\Delta CO_2$ emission ratios from large forest fires is very difficult if not impossible, as ground-based measurements in such violent fires are not possible and aircraft measurements are prone to undersampling the smoldering emissions, especially the contributions from RSC."

*8-26: This is a somewhat surprising statement to me. Differences in bottom-up and top-down results can originate from uncertainty in many parameters, emission factors being one of them. The standard deviation of CO in boreal and temperate forests is relatively speaking not that much larger than in savannas which to me is not surprising given the large variability in moisture regimes and tree species and density in those forests. I feel it would be more useful to analyze whether there is a difference in ground and airborne studies to say something about RSC.*

I'm not quite sure what the reviewer is getting at here. In response, I have changed "…which may be responsible for…" to "… which may significantly contribute to…". The role of RSC is highlighted in the new sentence added: "A representative measurement of fire-average $\Delta CO/\Delta CO_2$ emission ratios from large forest fires is very difficult if not impossible, as ground-based measurements in such violent fires are not possible and aircraft measurements are prone to undersampling the smoldering emissions, especially the contributions from RSC." Regrettably, I am not able to propose a simple solution to this problem.

*My other main point of criticism relates to Table 2 and the statement in the conclusions that the uncertainty in biomass burning emissions nowadays is as large as in the 2000s. Table 2 shows various estimates and the large range stems for a substantial part from outliers such as*

*FLAMBE which predict 10 times higher emissions in tropical forests compared to savannas, totally different from for example GFED4 and GFAS1.2 (derived from GFED3). I understand that it is beyond the scope of this paper to assess which one is right but this deserves some explanation, for example using previously mentioned top-down estimates based on CO. Simply combining the 8750 Tg DM in tropical forests from FLAMBE and the CO emission factor (105 g CO per kg DM) indicates CO emissions from this biome alone of 900 Tg CO per year, something not corroborated by inverse estimates and also at odds with the best estimates of deforestation (e.g., Houghton and Nassikas, 2017, https://doi.org/10.1002/2016GB005546).*

My point here is to highlight a problem, not to analyze the validity of the various studies that are listed in Table 2, for which I am not qualified and which would also go far beyond the scope of this paper. Therefore, I had to accept each of these peer-reviewed (by peers much more qualified than me) and published studies at face value. I did not feel that it was my place to call any of these studies an "outlier". What I am hoping for is that the community sees these large discrepancies by having them juxtaposed in one Table, and makes efforts to address them. Thus also the deliberatively provocative statement that "the uncertainty in biomass burning emissions nowadays is as large as in the 2000s". I would very much welcome a paper by the remote sensing community, maybe in the form of an intercomparison project, that will prove me wrong! But to address the reviewer's specific concern about FLAMBE, I have added a statement paraphrasing his/her comment and a reference to the review itself: "The inverse analyses may also be useful to indicate unlikely estimates based on remote-sensing techniques. For example, the burning of 8750 Tg dm in tropical forests estimated by FLAMBE, combined with the corresponding $EF_{CO}$ (105 g kg$^{-1}$) would produce CO emissions of 900 Tg a$^{-1}$ from this biome alone, well above the range of inverse CO emission estimates for all open burning (see also the comments by Reviewer 1, https://doi.org/10.5194/acp-2019-303-RC1)."

---

## Author Comment (AC4) · 31 May 2019

Response to Reviewer 2 (Charles Ichoku)

I thank the reviewer for his positive and constructive comments. My responses are detailed below. (Reviewer comments in italics).

*Page 3, Line 15: I believe it is more conventional to refer to the FTIR technique as "spectroscopy" rather than "spectrometry".*

There is a lot of discussion about this in the literature. I use the definition of the "IUPAC Gold Book", the authoritative reference for chemists: "Spectroscopy: The study of physical systems by the electromagnetic radiation with which they interact or that they produce. Spectrometry is the measurement of such radiations as a means of obtaining information about the systems and their components." (http://goldbook.iupac.org/html/S/S05848.html).

*Page 6, Line 9: Change "depending of" to "depending on".*

Corrected.

*Page 9, Lines 5-10: My main concern here is the use of Fire Radiative Power (FRP) as the sole basis to distinguish "top-down" satellite BB emissions methods from "bottom- up". All satellite BB emissions methods described in the article utilize satellite observations (fire-pixel counts, burned area, or FRP) as a way of estimating the biomass burning activity. The use of one or another parameter (FRP or not) does not make a method "top-down" or "bottom-up". Since the driving variable for estimating BB emissions are the factors that convert the activity to emissions (e.g. emission factors, as eloquently discussed in the article), it is the spatio-temporal distribution/configuration of the original input emissions, which went into deriving these EFs that determine whether a method is "top-down" or "bottom-up". If those input emissions were observed at a few locations and limited time periods, and then scaled up globally, the method is "bottom- up". But, if the input emissions were observed globally and regularly, and then scaled down to their sources, it is "top-down", as in the use of satellite-derived aerosol optical depths (AOD) of smoke-dominated aerosols to constrain the "emission coefficients" used to derive the emissions. Bearing this in mind, among the satellite methods described in this article, only QFED and FEER used globally-observed AOD to derive the coefficients that were then used to derive their final BB emission products, and thus may be categorized as "top-down". The others (including GFAS, which is scaled to GFED emissions) used locally-observed BB-emitted constituents to derive emission factors that were then generalized for their global BB emission products, and thus may be categorized as "bottom-up".*

Since the distinction between top-down and bottom-up satellite products is not really important to the discussion in this paper, I am limiting the use of "top-down" to inverse studies, where an emission is estimated from a concentration field. "Bottom-up" is now used only as a general term, where emissions are estimated based on a product of activity and emission factor estimates.

The satellite emission products are now referred to based on the quantity sensed (FRP, burned-area, etc.).

*Page 9, Line 17: I am concerned about the use of FAO (2015) as the primary reference for a quantitative value, as I am not sure whether FAO (2015) was peer-reviewed. I believe it would be better to find and cite the original (peer-reviewed) source of the 53 Tg/yr estimate reported in FAO (2015).*

The FAO studies are not based on a particular paper, but on reports from the individual UN member states to FAO regarding the amounts of agricultural and forestry activity in each country. As such, the FAO reports are only as good as the quality of reporting from each country, which of course varies considerably. Nevertheless, the FAO reports are generally considered to be an authoritative source on agricultural activity. FAO states: "Prior to publication, these reports are subject to quality control through a standardized peer-review mechanism that allows relevant stakeholders to provide valuable feedback on initial drafts of the reports."

---

## Author Comment (AC5) · 31 May 2019

Response to Reviewer 3 (Robert Yokelson)

I thank the reviewer (and his colleagues) for his positive and constructive comments. My responses are detailed below. (Reviewer comments in italics).

*I will note here that many updated averages calculated now will likely soon be superseded by large-scale recent (WE-CAN, https://www.eol.ucar.edu/field_projects/we-can) or planned work (FIREX-AQ, https://www.esrl.noaa.gov/csd/projects/firex-aq/) of unprecedented scope.*

I am looking forward to seeing these results. They will be incorporated in the online spreadsheet that I am making available (see last sentence of the conclusions with URL of data archive).

*The major new work is by Jayarathne et al, Stockwell et al (2 papers), Hatch et al., and Smith et al. All of these papers are used in this review (hereinafter A19) and are also on the Akagi 2011(A11) update website. Thus, one general point is this paper should mention that the A11 assessment has an update website as a community service (http://bai.acom.ucar.edu/Data/fire/).*

This information was added in the Introduction.

*No new global averages for peat fires are computed in A11 primarily because >600 compounds are now identified from peat fires, tropical and temperate peat may burn differently, and a global average is not the only type of desired input. A19 does compute new "snapshot" literature average EF, but based only on tropical peat data, which may or may not be similar to true global averages, but in any case a quick accuracy check was in order.*

Since tropical peat fires are likely the largest global source of these emissions, and since there are no field data from extratropical open peat fires, this type of snapshot is the only available option.

*A number of entries from Stockwell et al 2016 were copied correctly. I was particularly pleased to see that A19 did NOT quote the PM2.5 from Stockwell since it is clearly stated to be a subset of the more extensive PM2.5 data in Jayarathne. This has escaped some readers, so kudos to A19. Next though, I noted that the "BC" entry is actually the "EC" from Jayarathne. EC measurements can be inflated by charring of OC, and the BC by photoacoustic spectroscopy in Stockwell was 0.0055 or ~35 times lower.*

In general, the data set contains both BC (optical) and EC (thermochemical) measurements. Admittedly, this is not ideal, but the entire issue of BC and EC measurements is so full of problems (see our Andreae and Gelencser, 2006, paper for an overview) that I did not feel that this assessment was the place to take on this issue. In the case mentioned by Yokelson, I ended up selecting the EC value, since it seemed a more direct measurement that the PAS one. Both are peer-reviewed published results. I am open to suggestions, which value to pick.

*Also, there was no entry for SSA for peat despite the data in Stockwell et al allowing a reasonable SSA estimate at any wavelength.*

SSA is an intensive optical property and as such does not fit into a Table of emission factors,

*The EF for SO2 from Tab S3 of Stockwell et al 2015 is probably too high for a global average because it is the only EFSO2 in the study, and SO2 was below detection for most peat fires as revealed by consulting two other tables in the paper. (Factoring in below detection limit data to "averages" is tricky and I will not discuss it in detail here).*

Again, this is a published value, and I am not in a position to make up my own data by factoring in below-detection data in some arbitrary way. The fact that it is listed here with "N=1" and that the value is higher than most other emission factors should alert both the users to exercise caution and the researcher community to the need for more measurements.

*I checked a handful of NMHCs that were correct, but did note that the sum of 2-methyl-butenes actually included the 3-methyl-butene in "S16", although this is a very minor issue. Is it fair to estimate an error rate from a few spot-checks? I don't know. Overall, this could be a great starting point along with A11, but not using the original material increases the chances of introducing errors!*

Actually, this is a typo. There should not have been a 2 there – this entry is the sum of measured methylbutenes. Corrected.

*I also decided to perform a quick check on the formic acid data since the HITRAN parameters for HCOOH were changed by a factor of ~2.2 in 2012, which impacts all orbital and suborbital IR retrievals from before then. In A11 we adjusted all the old data for HCOOH, acetol, and glycolaldehyde based on new IR cross-sections. I randomly chose Yokelson et al., (2003) to see if HCOOH was updated and was surprised to see our formic acid data and nearly all our data from our 2003 paper missing. I found our data in the Sinha et al., (2003) entry where it had also appeared. So I'm glad the data don't appear twice, although it would be easier to trace the source if quoting the original paper. In any case the old incorrect value is still there. As an aside, I also noted that Burling et al., 2011 is in the reference list, but the data are not in the spreadsheet, perhaps to avoid duplication?*

I did my best to avoid double listing of data that were published in several different publications to avoid bias in the averages. The updated HCOOH, acetol, and glycolaldehyde data has been entered in the database.

*So again, this is a good resource and a lot of papers were read with some caution per limited spot checks, but users should be encouraged to consult the original work to double-check or trace important values. I think I noted somewhere that A19 has a place to send in corrections so that is a good feature.*

*This brief dive into the data reveals some general issues that impact the whole paper. A literature average where every study is weighted the same may not be a true global average for many reasons including:*

I deliberately refrained from applying weighting factors, because I did not feel that I have enough of a basis to derive objective and quantitative values for such factors. I did not want to bias the results based on some kind of factors based on "best guess" or "expert judgement". This paper has a global focus, and there are many regional differences. Appropriate weighting factors are likely different in different regions, and I doubt that anyone has the information required to derive globally representative weighting factors. Specific comments follow:

*1)        No effort is made to weight more modern measurements techniques. For instance A11 preferred thermal optical EC over plain thermal EC. Now we can probably prefer PAS or SP2 BC to any EC or at least be clear about the measurement. E.g. Li, H., Lamb, K. D., Schwarz, J. P., Selimovic, V., Yokelson, R. J., McMeeking, G. R., and May, A.: Inter-comparison of black carbon measurement methods for simulated open biomass burning emissions, Atmos. Environ., 206, 156-169, https://doi.org/10.1016/j.atmosenv.2019.03.010, 2019.*

I am well aware of this issue. However, what are the alternatives? Putting in some weighting factor based on "expert judgement" (which the reviewer dislikes, see below) on each study? I went on the assumption that as more "modern" studies accumulate, their weight will increase in the average.

As mentioned above, BC and EC are the most problematic category. The optical properties sensed by PAS can range easily over a factor of two for the same amount of soot carbon. The SP2 measures well-formed soot, but may miss some of the BC end of the BrC/BC continuum, which should be included with BC. For climate modeling purposes, do we want to know the actual mass of soot carbon, or rather the optically effective equivalent, BCe? And so on… After having been in the BC field for 40 years now, I just think there is no "best" measurement or best general answer, and no way to plug in some weighting factor to "correct" or eliminate bias. I am including a statement in Section 2.1 to make the reader aware of these issues.

*2)        The values are not weighted by available estimates of relative activity within the category. For instance, some examples:*

*a)        Clean-burning stoves and dirty open-cooking fires are lumped together and not weighted for the greater prevalence of open-cooking. A11 has separate cooking fire categories for this reason and that should be mentioned in the A19 text. Further, cooking fire studies in labs tend to*

*see different amounts of pollutants than in field studies with some very different results (e.g. Coffey et al., ES&T, 2017, references therein, and references mentioned below).*

This issue is discussed briefly in section 2.1. A full review of the widely diverse emissions from biofuel use would require a separate paper, if not several papers. I have selected as the basis of Table 1 those studies that reported measurements in actual households or lab studies that tried to recreate household conditions. I have not included modern clean-burning stoves used in first-world countries. In papers that studied both traditional and modern stoves, I have only extracted the data from traditional stoves. Consequently, my average is intended to be representative of traditional biofuel use. At present, there are huge and rapid shifts in the patterns of domestic fuel use, which make any weighting by activity of limited use. I have added a sentence referring to A11 in Section 2.1. Users with specific interests, who need less aggregated data, can easily obtain them by using the Supplement. I have also added a sentence in Section 3.2: "Valuable detail about the various burning types and further breakdown of some categories, e.g., biofuel use, into relevant subcategories can be found in Akagi et al. (2011)."

*b)      Wildfires and prescribed fires create a similar amount of emissions in the US annually, but wildfire measurements are much less common in the literature. Wildfire emissions were recently found to differ significantly from prescribed fire emissions (Liu et al., 2017). In practice, A19 included two studies with anomalously large EFPM (up to 4 times the average) that seem to have pulled the temperate forest average to a value in between the most advanced measurements of the wildfire and prescribed fire EFPM. However, separate EFs for prescribed and wildfires has potential to significantly improve air quality modeling.*

As reflected in the standard deviations, PM emissions vary widely, and results are also highly dependent on measurement approaches (lab vs field, aircraft vs ground, optical vs gravimetric, etc.). I'm not sure whether we can at this point generalize that one type of fire always has larger emissions than another, based on the limited data available. Does the work of Liu et al represent all wildfires and prescribed fires worldwide? And again, the data are easily separated out using the Supplement Table.

*c)      Crop residue burned in piles is lumped together with crop residue burned loose in the field without the detailed caveats provided in A11 about how drastically the emissions differ between the two burning styles. Some recent papers now estimate how the crop residue is burned (e.g. Lasko et al Environ. Res. Lett. 12 (2017).*

Maybe, but I don't know how much residue is burned in piles or loose across five continents. Again, this paper has a global focus, and thus cannot address fine-grained issues such as this.

*d)      Grasses and shrubs are combined as "savannas", but for the dominant moist savanna fires, the fuels are mainly grass and miombo tree leaf litter and then some logs late in the dry*

*season. Some global models assume a pure grassland category. It would be more useful to users of this paper to include separate grassland and shrubland/woody savanna categories.*

There are always costs and benefits in splitting and lumping. Modelers who want to use this data would probably prefer a split between C3 and C4 plants over one between grasslands and savannas. Then, the difference between grassland and savanna is often not very clear. In a miombo woodland, or a southern or West African savanna, a large part of the fuel is grass. How to draw the line? Then, splitting reduces the number of data in each category, increasing the likelihood that "atypical" values bias the mean.

*e)      The studies are not weighted by the amount of sampling: a study sampling 157 whole fires gets the same weight as a study grab sampling one fire (A11 uses weighting).*

Again, what is the appropriate weighting factor? 157 to 1? Or 10 to 1? I worked on the concept that a study represents a particular biome, and that in some cases there is only one measurement in that biome and in other cases there are many. For some studies that refer to several biomes, I have included these biomes separately.

*f)      Some attention is paid to how representative the sampling is, but not a lot.*

OK.

*g)      A global average may be inappropriate for a regional/seasonal application, or regional/seasonal EFs may improve global models. This is alluded to indirectly, but not stressed enough.*

I added a sentence in the conclusions: "While the approach here was focused on global averages, future work should also emphasize regional and seasonal differences in order to better support more highly geographically resolved modeling."

*Another general issue relates to the most useful reviewer comment we got on A11. What has changed? People are busy and may be curious if changing their model input will matter or which species to double-check in detail. In response we added a figure showing all the large changes for major emissions between A&M2001 and A11. Something similar could be added to this study although the targets are less clear because A11 is updated on web and A&M2001 has been updated by private communication over the years. What has changed between A11 and A19 and the last update widely dispersed by private communication? A11 computed new values for temperate forest (2014) and savannas (2015), and these are posted on the website. It might be best to compare to the 2015 web update, which includes all the updated averages.*

*In general A&M2001, A11, and A19 will all be useful resources and highlighting the overlap and complementary strengths will make all these resources more useful to the community. A11 can add A19 to their update page and A19 can do a better job of pointing to A11. Even A&M2001 has some important components (e.g. equations) that are not in A19.*

I have now added a direct reference to the A11 website with an URL.

Some of the equations and other detail from A&M2001 have not been included, since I wanted to avoid duplication. The methods section is focused on what's different in the approach of the current paper from that in our previous one.

Comparing the EFs in this paper to previously published data is a valuable suggestion. However, I don't feel that it would be appropriate to make a comparison between the results in this paper and data provided informally by me or made available by others on a website. I have therefore decided to compare selected values from A19 to the EFs in A11. The results are in Fig. 2, and discussed in the text in Section 3.2.

*It would be more important to include some assessment of what is new than the global totals in Table 3 if length is an issue. Global totals are/were interesting; especially in the early days of BB research to confirm global importance of BB, but they are less important now. Nearly all fire emissions are too reactive to be well mixed globally and even for relatively inert species such as CO, the location and timing is needed along with amount for inversions. The standard among modelers now is to compare emissions at the regional level.*

I was motivated to write this paper by requests for updated emissions data by several modelers, all of which were focused on global studies. In inversion studies, location and timing is provided by the model, whereas this paper is focused on emission factors and uses global emissions only as illustration of the magnitude of emissions and to highlight global scale uncertainties. Some of the motivation to include Table 3 was also that I am seeing this paper as an update of A&M2001 rather than as a completely new approach, which I am encouraging other authors to undertake.

*Another general issue is that the goals and accomplishments of much of the recent BB EF research are not discussed and many new EF results are not included in the tables; even though the papers were used to some extent. At the time of A11 about half the NMOG (by mass) were still unidentified, yet they surely react in real plumes. Tremendous progress has been made in the last 5-7 years with PTR-ToF-MS, 2D-GC-ToF-MS, etc to identify more of the unknowns. In addition, the amount of sampling and especially the sophistication of the instrumentation for sampling of previously undersampled fire types has seen a substantial increase including agricultural fires, wildfires, cooking fires, etc. No amount of measurements can reduce natural variability, but we have nonetheless greatly decreased uncertainty in smoke chemistry, there is an important difference.*

This paper is not meant to be a general review of the progress of BB emission studies. The main objective was to provide an update to the A&M2001 data set in published and referenceable form. All the papers have been included that I am aware of and which provide data from which emission factors can be calculated. Because of the restriction to field measurements, some of the really exciting lab studies are not included in the data on which the EFs for the different fire categories are based. I am always happy to include pertinent information if brought to my attention.

*Along these lines, no rationale is given for selecting 121 compounds to include out of the 700 plus that have now been measured. No estimate is given of how much additional NMOG is unaccounted for by the A19 tables. These are major issues. The uncertainty in data from carefully-simulated lab fires, especially when scaled to field conditions, is less of a problem than completely ignoring the chemistry of much of the emissions. Other less sweeping issues arise from the apparently ad hoc approach to what data are included. For instance, the sum of all isomers is used for "terpenes" even though some studies speciate the terpenes and they have different reactivity and potential to form SOA. At the same time, lab data for the sum of dimethyl and ethyl amine (same mass) are not quoted and the only data reported provided separate results. It's likely more important to speciate the much more abundant terpenes.*

The compounds were selected based on the availability of enough field data to derive meaningful estimates and the importance of the species for climate and/or chemistry or their use as burning tracers. To address the very serious issue of underestimation of total volatile organics emissions, I have added some discussion in Sections 2.1 and 3.2, and included the NMOG emissions estimates from the online updates to A11 in Tables 1 and 3.

Unfortunately, there are not enough consistent field data on specified terpenes to enable species-specific data in Table 1. Regarding dimethyl and ethyl amine, I have avoided including mixtures of isobaric species from PTRMS measurements that could not be resolved to specific compounds.

*Another critical current issue that is not discussed is measurements of intermediate and semivolatile compounds. These species are important SOA precursors and we need the SOA precursors to get BB-PM and its significant health and climate impacts right. Expert assessment helps because this also gets into the realm where the EF of an SVOC and the EF of organic aerosol can depend strongly on the concentration of the smoke being measured.*

This issue is now discussed in the conclusions.

*Per the other reviews:*

*I read the comments of Referee #1 and they all seem reasonable.*

*Ichoku review: I agree with this Referee's important clarification/correction re delineation of top-down and bottom-up, but add a few points. Bottom-up estimates are difficult for many reasons, but top-down is perhaps presented in overly favorable terms and a bit incomplete.*

*Aerosol emissions are not measured globally but estimated on an extensive scale based on column AOD. AOD is reactive and not conserved, and gaps in AOD exist due to clouds, the cloud mask, orbital gaps, extensive time between overpasses, etc., etc. Importantly, attribution of AOD to specific sources is highly uncertain: e.g. plume injection altitudes are not operationally measured, crop waste burning can occur in forest clearings, or cooking fires and crop residue fires occur side by side in Asia where industrial sources, biogenic SOA, and sometimes peat fires also contribute to AOD. Comparing top and bottom is however super helpful. Finally, top-down using CO exists in numerous studies and gets around the "reactive issue" for AOD, but not the other issues although CO sources may be better constrained than AOD sources. The use of CO in inversions is discussed on page 10 when estimating uncertainties in global totals. The potential to use multiple CO sources could be stressed. E.g.*

*Kopacz, M., Jacob, D. J., Fisher, J. A., Logan, J. A., Zhang, L., Megretskaia, I. A., Yantosca, R. M., Singh, K., Henze, D. K., Burrows, J. P., Buchwitz, M., Khlystova, I., McMillan, W. W., Gille, J. C., Edwards, D. P., Eldering, A., Thouret, V., and Nedelec, P.: Global estimates of CO sources with high resolution by adjoint inversion of multiple satellite datasets (MOPITT, AIRS, SCIAMACHY, TES), Atmos. Chem. Phys., 10, 855-876, 2010.*

*Further, since the topic of how much biomass burns is included, then another important approach to how much burns is scaling of a-priori bottom-up emissions to match surface and aircraft data and AERONET AOD as in Reddington et al.*

*Reddington, C. L., Spracklen, D. V., Artaxo, P., Ridley, D. A., Rizzo, L. V., and Arana, A.: Analysis of particulate emissions from tropical biomass burning using a global aerosol model and long-term surface observations, Atmos. Chem. Phys., 16, 11083-11106, https://doi.org/10.5194/acp-16-11083-2016, 2016.*

*I don't like unspecified "expert judgment." A11 gives several recipes for estimation of unmeasured EF and they recommend trying several. It may be helpful to reference this discussion and clarify which approach(s) were used in A19.*

The methods used for estimating unmeasured EFs are discussed in Section 2.4 and the specific method used for each compound was given in the last column of Table 1. I now see that somehow this column was lost when the pdf was transferred to the published version.

*The short comment by Nic Surawski suggests using "burnt carbon" rather than "dry fuel consumed" as the EF basis. The valid underlying issue is that the %C of the fuel may not be the %C of the emissions, which can make the carbon mass balance (CMB) method less rigorous.*

*Neither %C is known in most field studies but in principle char formation causes the %C of the emissions to be lower than the %C of the fuel. On the other hand, Santín et al., (2015) found that "higher %C" forest fuel components tend to burn with greater completeness, which tends to cause the %C of the emissions to be higher than the %C of fuel. This tends to cancel the impact of char formation on EFs calculated by the CMB.*

*In charcoal kilns, large pieces of solid charcoal are formed and the large charcoal yield can be measured reasonably accurately. Bertschi et al., (2003b) describe one practical method to adjust the CMB to get EF both per kg wood used and per kg charcoal made.*

*The situation changes for landscape fires. The charcoal yields are small and the charcoal is manifested mainly as a fine powder mixed in the exported plume or ash layer or a thin black surface layer on otherwise unburned fuel. Further "burnt C" arguably becomes undefined and unmeasurable in practice since some biomass is "affected by the fire" in ways that do not make char. The canopy can be scorched (turned brown by heat from below), creating emissions, but no char. Distillation of stored terpenes in wood occurs at temperatures below those creating char.*

*Deciding what part of a forest was part of the "carbon burned" is not well defined.*

*On a practical level, there is a large historical database in the literature on fuel consumption, which was estimated as pre-fire minus post-fire biomass. Examples are included in A11 and there is a recent compilation (van Leeuwen et al., 2014). In contrast, there are few to none data for "burnt C" for major fire types. In general though, the impacts on the EF from the CMB is likely less important than the need for more quality measurements of char yields from landscape fires. This needs to be addressed to improve C-cycling estimates since the char is a carbon sink.*

*References:*

*Santín, C., S. H. Doerr, C. M. Preston, and G. González-Rodríguez (2015), Pyrogenic organic matter production from wildfires: A missing sink in the global carbon cycle, Global Change Biol., 21(4), 1621–1633, doi:10.1111/gcb.12800.*

*van Leeuwen, T. T., van der Werf, G. R., Hoffmann, A. A., Detmers, R. G., Rücker, G., French, N. H. F., Archibald, S., Carvalho Jr., J. A., Cook, G. D., de Groot, W. J., Hély, C., Kasischke, E. S., Kloster, S., McCarty, J. L., Pettinari, M. L., Savadogo, P., Alvarado, E. C., Boschetti, L., Manuri, S., Meyer, C. P., Siegert, F., Trollope, L. A., and Trollope, W. S. W.: Biomass burning fuel consumption rates: a field measurement database, Biogeosciences, 11, 7305-7329, https://doi.org/10.5194/bg-11-7305-2014, 2014.*

See my response to the comment by Surawski.

*Line by line comments in P, L format*

*1, 10: "critically evaluated" is probably better as "considered"?*

I did evaluate them critically for validity and appropriateness for inclusion.

*General on abstract include a sentence on how many species changed by e.g. a factor of two since A11?*

I added a sentence: "For key species, the updated emission factors are compared with previously published values."

*1, 2: Some carbon cycle people argue that much of the CO2 from fires should not be counted as emissions if the vegetation grows back.*

This issue has now been addressed in the Introduction: "While a significant fraction of the emitted $CO_2$ is taken up again by vegetation regrowth, much of it remains in the atmosphere for years and potentially even up to centuries, e.g., in the case of tropical deforestation fires or peat soil burning (van der Werf et al., 2017). Model simulations suggest that in the absence of fires, atmospheric $CO_2$ concentrations would be about 40 ppm lower, indicating the importance of fires for the atmospheric carbon budget (Ward et al., 2012)."

*1, 22: A glance at Table 1 seems to show higher EFN2O than I expected. N2O has been found to account for <1% of fuel N while NH3 is a major fate of fuel N. Are the N2O/NH3 ratios in Table 1 high due to including older studies with artifact N2O in canisters? I think not, but worth checking.*

The old artefactual data were not included.

*1, 22: Insert "BB is the second largest global source of non-methane organic gases (Yokelson et al., 2008, A11)."*

Done.

*2, 3-4: Fire increases locally available P by raising soil pH. See Jordan, C. F. 1985. Nutrient Cycling in Tropical Forest Ecosystems: Principles and Their Application in Management and Conservation. Chichester: Wiley.*

This is correct and important, but is not directly related to atmospheric emissions, the topic of this paper.

*2, 5-6: suggest retiring the term "VOCs" and using non-methane organic gases (NMOG) to recognize important gas-phase emissions with intermediate and lower volatility. Cite the following or equivalent:*

Since VOCs is used much more widely, I am retaining it here when referring to the species set used in A&M2001. I am introducing NMOG for the more comprehensive species set measured in more recent work.

*Robinson, A. L., Donahue, N. M., Shrivastava, M. K., Weitkamp, E. A., Sage, A. M., Grieshop, A. P., Lane, T. E., Pierce, J. R., and Pandis, S. N.: Rethinking organic aerosols: Semivolatile emissions and photochemical aging, Science, 315, 1259–1262, doi:10.1126/science.1133061, 2007.*

*May, A. A., Levin, E. J. T., Hennigan, C. J., Riipinen, I., Lee, T., Collett, J. L., Jimenez, J. L., Kreidenweis, S. M., and Robinson, A. L.: Gas-particle partitioning of primary organic aerosol emissions: 3. Biomass burning, J. Geophys. Res.-Atmos., 118, 11327–11338, doi:10.1002/jgrd.50828, 2013.*

*Hatch, L. E., Yokelson, R. J., Stockwell, C. E., Veres, P. R., Simpson, I. J., Blake, D. R., Orlando, J. J., and Barsanti, K. C.: Multi-instrument comparison and compilation of non- methane organic gas emissions from biomass burning and implications for smoke-derived secondary organic aerosol precursors, Atmos. Chem. Phys., 17, 1471-1489, https://doi.org/10.5194/acp-17-1471-2017, 2017.*

*Hatch, L. E., Rivas-Ubach, A., Jen, C. N., Lipton, M., Goldstein, A. H., and Barsanti, K. C.: Measurements of I/SVOCs in biomass-burning smoke using solid-phase extraction disks and two-dimensional gas chromatography, Atmos. Chem. Phys., 18, 17801-17817, https://doi.org/10.5194/acp-18-17801-2018, 2018.*

*Jen, C. N., Hatch, L. E., Selimovic, V., Yokelson, R. J., Weber, R., Fernandez, A. E., Kreisberg, N. M., Barsanti, K. C., and Goldstein, A. H.: Speciated and total emission factors of particulate organics from burning western US wildland fuels and their dependence on combustion efficiency, Atmos. Chem. Phys., 19, 1013-1026, https://doi.org/10.5194/acp-19-1013-2019, 2019.*

Added here and/or further down in the text.

*2, 7: Cite review of O3 formation in BB plumes; Jaffe, D. A., and Wigder, N. L., 2012. Ozone production from wildfires: A critical review. Atmospheric Environment 51, 1–10, doi:10.1016/j.atmosenv.2011.11.063.*

Done.

*2, 7: change "other pollutants" to "secondary PM" or some equivalent term*

Done

*2, 8: delete "emitted" – these last two changes provide at least minimal recognition that much of the BB-PM impacts are from secondary PM.*

Done.

*2, 9: Akagi et al., (2014) present likely the most comprehensive assessment of toxic gases in fire-line smoke (add to reference string).*

Done.

*2, 16: "disconcerting" perhaps, but given the difficulty of measuring how much BB occurs, not at all surprising.*

OK.

*2, 26: I would add "numerous" before "field" as there are probably too many recent and on-going studies to provide complete references.*

Done.

*2, 27: Most of the EF results can be found in just 2-3 journals. I'd rephrase "The results of these studies are, however, widely dispersed among hundreds of papers in a large number of journals" to "The results of these studies are dispersed among hundreds of papers".*

I looked at the reference list in the supplement and reached ten different sources before I had gone past the letter B in the authors list.

*2, 28: add "on a global scale" after "data" since most papers do synthesis/comparison at some scale.*

Done.

*2, 29: define Akagi et al., 2011 as "A11" to facilitate further citation.*

I prefer retaining the full reference.

*2, 30: After "emission factors." insert ~ "I have provided informal updates to A&M2001 and A11 maintains an update website (http://bai.acom.ucar.edu/Data/fire/).*

Done.

*2, 32: Insert "first appeared" after "previous compilations" to make it clear updates have already been readily available.*

Done

*2, 32: Why 28 out of hundreds of new species?*

As mentioned above, criteria were availability of sufficient field data and perceived importance for climate, chemistry, and/or tracers.

*2, 32: Are any species in A19, but not the original A11? Text should be inserted to clarify that new species were in the updated tables and papers posted on the A11 update site and in informal updates to A&M2001 distributed by the author. Also would be ideal to insert a mention here of recent or planned work that will modify these values, i.e. campaigns I cited above.*

The fact that A11 included additional fire categories and species is mentioned already in my addition on p3, l6 (new), and does not need to be reiterated here. I prefer not to specifically refer to future activities.

*2, 33: After "burning types" insert "following A11"*

The fact that A11 included additional fire categories is mentioned already in my addition on p3, l6 (new), and does not need to be reiterated here.

*3, 9: since this paragraph paraphrases A11 should add "following A11" before "I only …"*

Not quite sure what the reviewer means here. Actually, the approach here is different from A11, since they generally used a blend of field and lab data and I used lab data only as a "last resort", as stated here.

*3, 15: Add "solar" before "Fourier" and "spectrometry" should be "spectroscopy"*

Solar was added. With regard to "spectrometry" I am following the IUPAC Gold Book definitions. See my comments to the Ichoku review.

*3, 21: Good place to add that some lab data is adjusted to reflect field conditions using "overlap species", ERs, or MCE as discussed in Yokelson et al., 2013. I think that data appears to have been used.*

Done.

*3, 28: I would change "usually" to sometimes". 3, 31: change "typically" to "may be"*

*The logic is that some lab studies were carried out in the Missoula Fire Lab using fuels that were locally-collected by forest fuel experts or fuels "Fed-Exed by forest fuel experts. The fires were burned at a scale with flame lengths etc close to real world conditions. Fuel moistures in the FIREX-2016 lab experiment were quite high for example. Canopy fuels sent from the SE US had fuel moistures on the order of 136% on a dry weight basis. Also some lab studies report data adjusted by the MCE, ERs, or field/lab ratio for overlap species (Selimovic et al., 2018; Stockwell et al., 2015; Yokelson et al., 2013; etc). Most importantly, for a vast number of species, there is only lab data. To some extent this is clarified on P4, lines 2-4, but these points are important to make consistently in a revised paragraph here.*

I replaced "usually" with "often" and changed "typically" to "may be". The use of field-adjusted lab data has also been added. I don't think this is the place to go into an extended discussion about the merit of lab vs field data. This discussion can already be found in the literature, especially in papers from the Yokelson team.

*4, 1: This MCE is of course unrealistic, but not even close to typical of most lab fires.*

Extreme, but not altogether untypical of lots of the lab studies on biofuel burning, unfortunately. I pointed out in the text that this is an extreme example.

*4, 5-9: This discussion is well done, but some references could be included for the reader interested in more details. The Bond group, for instance, has a number of papers that find lab attempts to replicate field cooking fall short. Stockwell et al 2016a show how MCE dropped off significantly from the lab to field and describe correction factors for the lab data. The risk of lumping all this data together should be clear as noted above.*

Again, I really did not want to go into an extended discussion on this. This topic would require a separate review paper. I am just talking about data selection here.

*4, 10-26: This section is good. Should the equation be numbered? Probably apparent that "mixing ratios" can be used interchangeably with "concentrations"?*

I put mixing ratio in parenthesis behind concentration. In principle, this should not make a difference, since the properties are ratioed.

*4, 27: change "easy" to "straightforward"*

Done.

*4, 31: I would change "is readily" to "can sometimes be". The fuel moisture evaporation contributes to mass loss in the lab and fuel moisture is variable within components and between components, which have different combustion factors. We use the carbon mass balance method in the lab, which has the advantage in the lab of minimal distortion of excess CO2 via mixing.*

Changed to "can be". The main point here was to point out that it is very difficult in the field.

*5, 1: We include EC or BC in sum of carbon.*

Added: ", and elemental carbon [EC] or black carbon [BC]"

*5, 2: More accurately fuel %C tend to be 40-45 for crops and grasses, 50 for wood/foliage, 55+ for peat.*

Ok, but many authors just use 45%. That's why the sentence starts with "Often". No way to go into the individual assumptions used in each and every study.

*5, 7-12: The equation; number it and check it! What is EF(X/Y)? The equation as presented makes no sense. I think it's trying to say something like: Say the ER mol/mol of C2H4/CH4 was measured as 0.1, but the data needed to compute EF was not collected in the study. If we know from other work that a reasonable guess at EFCH4 is 5, then EFC2H4 can be estimated as 0.1\*(28/16)\*5 or 0.875. If this is the intent of the text here, EF(X/Y), which is undefined should be EFX? However, if EFY is not known, then it should be made clear this is not the same as a measurement of EFC2H4, but just an estimation. Thus this discussion, after any needed corrections, from line 7 on, belongs in the discussion of estimates, not under "conversion of units."*

Equation numbers should not be necessary, since the equations are not being referred to further. If the Journal style requires it, they will be added. EF(X/Y) was a typo, it should simply be EF$_X$. The text was amended to point out that this is an estimate. The heading "Conversion of Units" was removed.

*5, 17-23: This whole discussion is confusing and may have errors.*

*If you assume the EFX is unknown but is proportional to EFCO then that seems to just be suggesting using a corrected version of the equation above with CO as the reference species "Y". If so, then on line 19 ERCO should be EFCO, the mass conversion ratio needs to be included, and it makes sense to use the ER(X/CO) from the most similar fuel type available rather than a global average. I.e. crops are grasses so if ER(X/CO) is not known for crops, but is for grasses, use that instead of factoring in the X/CO ratio for e.g. peat and garbage burning.*

*On lines 19 and 20: what is ERX/ERCO anyway? Is it just ER(X/CO) used above? If so be consistent – especially since ERCO uses what as a reference species?*

Sorry, another typo. It should have been $EF_X/EF_{CO}$ and $EF_{CO}$ etc. I hope that makes sense now. I use simply the proportionality of the emission of species X to that of CO to scale smoldering-dominated emissions.

*Next, for flaming compounds using the consumption weighted average of all categories makes less sense than using the most similar biomass type category as noted just above. Otherwise, the implication is that fire type doesn't matter; inconsistent with the rationale for creating fire type categories in the first place.*

Unfortunately, the column in Table 1 that specified the estimation technique was lost in the process of creating the ACPD version. AV was actually only used for $N_2O$, $SO_2$, DMS, and HCl. I started out using this category for flaming-dominant species, but in the end applied it only to some hetero-element-containing species, where the N or S content of the fuel are likely more important than fire type. I amended the text.

*Finally, on line 23, what is a subjective best estimate? Some procedure was followed that should be spelled out.*

I added "Specifically, for missing values of total particulate carbon emissions, the sum of OC and EC emissions was used, and for aerosol potassium emissions in boreal forest fires I used the temperate forest value."

*The next four comments are related because smoldering is a combination of distillation, pyrolysis, and glowing combustion; and both glowing combustion and flaming combustion can induce distillation and pyrolysis.*

*5, 28: change "combustion" to "flaming or glowing".*

Done.

*6, 4-5: I would change "Once most volatile matter is consumed during flaming combustion, the remaining char undergoes gas-solid reactions between oxygen and carbon at the fuel surface,*

*called the smoldering phase" to "In addition to volatile matter being consumed by flaming combustion, char undergoes gas-solid reactions between oxygen and other gases and solid carbon at the fuel surface, called gasification or "glowing" combustion".*

Done

*Then on 6, 7: change "pyrolysis, flaming, and smoldering combustion" to "flaming and smoldering combustion (vernacular for a changing mix of distillation, pyrolysis, and glowing)"*

 Done.

*Also on line 7: since fires can have more than one plume say "the fire plumes at any place and time contain"*

Done.

*6, 15: "peatland" should be "peat" since peatland will have surface fuels that are consumed partially by flaming. Stockwell et al 2016b gives a better overview of how peatland fires play out than Bertschi et al., 2003a and Guillermo Rein's group has published detailed papers on peat combustion dynamics.*

Done.

*6, 18: change "a nocturnal" to "the" and change "serious problems" to "limitations". It is entirely possible for RSC to occur during the daytime and to measure RSC EF using ground-based sampling (e.g. Bertschi et al., 2003a; Christian et al., 2007, Akagi et al., 2013).*

Of course. But what I am leading up to here is the specific problem of measuring the contribution of RSC to fire-integrated EFs, given the RSC emissions into a shallow boundary layer where both pyrogenic and biogenic $CO_2$ are present at high levels. Ground-based sampling gives EFs from specific RSC point sources, but the problem is integrating that into fire-integrated averages. I tried to rewrite this passage to make the point clearer.

*6, 19: I would change "will completely miss" to "have trouble measuring" 6, 20: add "or fire blow-ups" after "daytime convection"*

Done.

*6, 20-21: Change "get lost" to "may be distorted by mixing"*

Done.

*6, 21-26: It is not any harder to measure CO/CO2 near the source for RSC than it is for any other source, but it should be done from the ground (see references above). The main problem is the RSC component of fuel consumption is difficult to measure to get a weighted fire average for overall emissions. Thus this paragraph should end with "Ground-based studies of RSC can obtain EFs of trace species, but these are difficult to relate to the corresponding amount of fuel burned." Delete the rest of the paragraph as it is misleading. Refer the reader to Bertschi et al., (2003a) for scenarios of how RSC impacts EF.*

Done.

*6, 32 – 7, 14: This discussion needs to be rewritten from a perspective with more realistic hopes for what MCE can accomplish. Figure 1 throws out almost all available useful data by using one point per study and needs to be deleted or replaced with something useful.*

*Some general comments followed by specific recommendations: MCE, CO/CO2, BC/CO, and BC/OA are all useful to illustrate how the relative amount of flaming and smoldering can cause BB EFs to vary; especially within a single fuel-/fire-type or study. MCE is most common and as MCE decreases the total products of incomplete combustion increase. The author cites numerous papers with examples of good correlation of EF, even for specific species, vs MCE and this helps make sense of the observed variability and might drive a model at a useful scale (TBD). Low MCE dependence can be "OK" too and can sometimes increase confidence that the average value is close to correct for a range of burning conditions (Table 4, Liu et al., 2016). CO is the indicator of smoldering, but smoldering is a dynamic mix of complex processes and a simple parameter based on two gases should not be expected to predict all the outcomes of thousands of relevant chemical reactions across the planet for all emitted species. On broad scales other factors like fuel type (as noted), fuel N (Burling et al., 2010), geometry (Bertschi et al., 2003a), weather, etc impact emissions and correlations decrease as more conditions are considered.*

*Every model has a scope and every model has limitations. There might be a user-specific scale/scope where the coverage and correlation of an EF vs MCE model are both adequate to improve emissions estimates. Figure 1 skips over that question, throws out the data, and just demonstrates the obvious conclusion that EF vs MCE is not universal. The proper next step in evaluating EF vs MCE is to compare slopes based on all the data in the original studies aggregated at some intermediate level. To illustrate what I mean I insert a table where that process is started:*

*A glance at the table suggests some potential for a "fire-type-specific" EF vs MCE model with the level of correlation and aggregation perhaps depending on species also. I have not pursued this due to lack of time and because MCE is not available operationally as fire model input anyway. For now MCE remains most useful as a way to partially deconstruct variability in reported EF data.*

I guess I had not make my point clear enough for the reviewer. What I wanted to say is that, while MCE is a wonderful tool for specific parametrizations within a confined group of fires, it is not a generally useful "one size fits all" way of deriving unknown EFs or parametrizing EFs in global models. Yes, this is obvious to the reviewer, but not necessarily to all potential users. I also thought it would be worthwhile to test this possibility by using this fairly comprehensive data set. In contrast to what the reviewer states, I am not throwing out data, but using all data in the form of their means. If there were a significant general relationship between EFs and MCEs, this relationship should show up even clearer in the means than the individual values. I am making some changes as suggested by the reviewer to make this more clear, but I leave any deeper analysis of EF/MCE relationships to future authors.

*With the above discussion as background I suggest the following revisions at a minimum. 7, 3: change "unfortunately" to "however"*

Done.

*7, 4: change "general parameterization of EFs" to "global parameterization of all EFs"*

Done.

*7 4-7: delete "As an illustration, I show in Fig. 1a and 1b plots of the EFs of ethene (C2H4) and ethane (C2H4) vs MCE, based on the studies in the supplemental spreadsheet. In both cases, the results scatter widely, and especially the data from the lab studies, biofuel burning, peat fires, and RSC-dominated fires introduce a large amount of scatter."*

*Fig. 1 is one point per study rather than comparing slopes using multiple points per study, which might tell a different story and preserves whatever information there is.*

I changed the sentence to: "As an illustration, I show in Fig. 1a and 1b plots of the EFs of ethene ($C_2H_4$) and ethane ($C_2H_6$) vs MCE, based on the average values from the studies in the supplemental spreadsheet…". I don't see a reason to change the rest, since it is simply a description of what is in the figures.

*7, 7-8: change "The poor correlation between EFs and MCE has been noted previously" to "The limitations of EFs versus MCE have been noted previously"*

Done.

*7, 8-11: delete "In the case of ethene, the correlation using all data points is not significant ($R2 = 0.07$). However, when only the data from open vegetation fires are included (and after*

*removing three outliers), the correlation improves to an R2 of 0.27. For ethane, the correlation coefficient is R2 = 0.38 for all data, but does not improve substantially by removing the peat fire data."*

Again, this is simply a description of the data and figure. I don't see what's wrong with that.

*7, 11-12: change "These results suggest the potential of using MCE as a meaningful, but rough predictor of EFs for at least some species." To "The level of aggregation at which MCE is useful as a meaningful, but rough predictor of EFs for at least some species has not yet been determined."*

Done.

*7, 13: change "supplement" to "original studies"*

Done.

*7, 14: insert "A new approach to modeling NMOGs from pyrolysis using PMF has potential (Sekimoto et al., 2018); especially if the factors can be related to operationally available input."*

*Sekimoto, K., Koss, A. R., Gilman, J. B., Selimovic, V., Coggon, M. M., Zarzana, K. J., Yuan, B., Lerner, B. M., Brown, S. S., Warneke, C., Yokelson, R. J., Roberts, J. M., and de Gouw, J.: High- and low-temperature pyrolysis profiles describe volatile organic compound emissions from western US wildfire fuels, Atmos. Chem. Phys., 18, 9263-9281, https://doi.org/10.5194/acp-18-9263-2018, 2018.*

A sentence regarding the Sekimoto study was added.

*7, 15-20: This may be worth trying, but model estimates of fuel consumption by flaming and smoldering would be difficult to validate in the field since access during the fire is problematic. Also the MCE of flaming or smoldering can vary broadening predicted MCEs.*

OK.

*7, 20: The first paper probing the relationship between greenness and MCE was Hoffa et al., 1999. Hoffa, E. A., D. E. Ward, W. M. Hao, R. A. Susott, and R. H. Wakimoto (1999), Seasonality of carbon emissions from biomass burning in a Zambian savanna, J. Geophys. Res., 104, 13,841–13,853. Korontzi et al., 2003 updated the MCE/Greenness relationship based on new MCE measurements and then combined measured MCE, MCE vs greenness, and EF vs MCE (from other work in the late dry season) to estimate early dry season OVOC EFs.*

Reference added.

*7, 22: In addition to Korontzi et al., 2005, greenness (PGREEN) was used to predict combustion completeness in Korontzi et al., 2004 and PGREEN was used to predict MCE by Ito and Penner, 2004 (https://agupubs.onlinelibrary.wiley.com/doi/full/10.1029/2003JD004423).*

*Korontzi et al., Modeling and sensitivity analysis of fire emissions in southern Africa during SAFARI 2000, Remote Sensing of Environment 92, 255–275, 2004.*

*This approach has potential, but so far has been used for savannas only and works best for species that correlate strongly with MCE. The results have not been tested with field measurements to my knowledge. The discussion might be revised slightly.*

References added. I prefer not to go deeper into this specific issue.

*7, 22: Maybe wrap up this section with something like "For now we should use the average EFs, but be aware they can vary considerably fire to fire."'*

Done.

*7, 27: Not sure what this means "The averages in this column can only be seen as general indications, since all types of fuels and burning methods are included,"*

Changed to: "The averages in this column can only be seen as general indication of the magnitude of emission factors found in the lab studies, since all types of fuels and burning methods are included in the averages. However, the original data and references are provided in the supplement for readers interested in the details."

*Pages 7-8 in general: A19 has adopted some of improvements of A11, which is good.*

Thanks!

*8, 4: after "category" it could be useful to cite this resource of garbage burning activity and EF: Wiedinmyer, C., Yokelson, R. J., and Gullett, B. K.: Global emissions of trace gases, particulate matter, and hazardous air pollutants from open burning of domestic waste, Environ. Sci. Technol., 48, 9523-9530, doi:10.1021/es502250z, 2014.*

Reference added.

*8, 21-22: An EF for particle number concentration is problematic and potentially meaningless or misleading due to rapid coagulation near sources! Warning label needed.*

Text added: "The rapid coagulation of particles very near the source makes it difficult to choose the most appropriate plume age for such a measurement (Hobbs et al., 2003; Sakamoto et al., 2016; Hodshire et al., 2019). However, a survey of available meas-urements suggests that the ratio of excess particle number concentration to ΔCO stabilizes at the scale of typical aircraft measurements in plumes as a consequence of the sharp decrease of the coagulation rate with increasing dilution (Janhäll et al., 2010). More field studies on the development of aerosol number concentrations and size distributions as a function of plume age under different conditions (fire size, wind speed, flux density, etc.) are warranted."

*8, 23: EFs for "brown carbon" (BrC) as g/kg are problematic because there are likely hundreds of contributing trace components with different absorption cross-sections that are also evidently reactive. But there is BrC emissions data in the form of Ångström absorption exponents (AAE) and BrC absorption EFs (as m2/kg following the Bond and Moosmüller groups) in the UV for fresh emissions from carefully simulated lab fires and numerous field fires for different BB types (Stockwell et al., 2016a, b; Goetz et al., 2018; etc). Total absorption EFs in the UV are also given for users who may prefer them.*

*To clarify misleading text: the discussions in Selimovic et al., (2018 and 2019) show AAE near 3.7 (field Forrister et al., 2015) and 3.3 (lab Selimovic et al., 2018) for fresh smoke, but decaying with age as shown in Forrister et al and with BrC accounting for ~50% of absorption at 401 nm in "moderately aged" smoke (Selimovic et al., 2018). Most of these papers are in the A19 tables, but BrC data, which is important as the author says, is not tabulated in general.*

*Forrister, H., Liu, J., Scheuer, E., Dibb, J., Ziemba, L., Thornhill, K. L., Anderson, B., Diskin, G., Perring, A. E., Schwarz, J. P., Campuzano-Jost, P., Day, D. A., Palm, B. B., Jimenez, J. L., Nenes, A., and Weber, R. J.: Evolution of brown carbon in wildfire plumes. Geophys. Res. Lett., 42, 4623–4630, https://doi.org/10.1002/2015GL063897, 2015.*

*Goetz, J. D., Giordano, M. R., Stockwell, C. E., Christian, T. J., Maharjan, R., Adhikari, S., Bhave, P. V., Praveen, P. S., Panday, A. K., Jayarathne, T., Stone, E. A., Yokelson, R. J., and DeCarlo, P. F.: Speciated online PM1 from South Asian combustion sources – Part 1: Fuel-based emission factors and size distributions, Atmos. Chem. Phys., 18, 14653-14679, https://doi.org/10.5194/acp-18-14653-2018, 2018.*

*The Goetz paper above and Jayarathne papers cited include data for ions and metals in PM. Major ions and metals are tabulated in A11, but not A19, a point worth making in A19.*

Text and references added: "Providing EFs for this species is problematic because of the very complex and variable mixture of compounds that make up BrC as well as its potential for rapid change in abundance and optical properties during plume evolution (Forrister et al., 2015). To some extent, data on the optical properties of BB aerosols can substitute for direct measurements

of BrC (Stockwell et al., 2016a; Stockwell et al., 2016b; Goetz et al., 2018; Selimovic et al., 2018)."

Information about ions and metals was added in Section 2.1: "Emission data for ionic species and trace metals are not included in this data set. They are tabulated in Akagi et al. (2011), and additional information can be found in a number of papers (e.g., Goetz et al., 2018; Jayarathne et al., 2018a; Jayarathne et al., 2018b)."

*8, 26-33: I would delete this paragraph or at least revise it extensively. In part because the "most" serious problem is subjective depending on the workers area. For instance, top-down estimates of BB are probably most concerned with the issues such as observational constraints I outlined in my general comment on top-down estimates above. Workers looking at SOA may care more about EFs for SVOC, etc. In general this represents the authors troubles measuring RSC from an aircraft and other issues could lead to the underestimates of regional CO emissions mentioned. Also, it's misleading because RSC does not affect only tropical forest fires. RSC accounts for a significant part of the emissions for all forest fires, pasture fires, and wooded savanna, and virtually all the emissions from peat fires for example. However, the situation is far from hopeless. Bertschi et al., (2003a) outlined a range of impacts when RSC accounts for 10% to 50% of the total fuel consumption in a fire. At the upper end with 50% of fuel consumption by RSC the CO2 and CO EF changed by about -7% and +13% respectively. The larger impacts of RSC are for other gases like NH3 and CH4. Further, in A11 the tropical forest EF were adjusted based on an assumed RSC component of just 5% per available evidence at the time.*

I changed the beginning of this paragraph to: "Regarding the role of vegetation fires in the global carbon cycle, the most problematic uncertainty pertains…" Some of the text in this paragraph has also been changed in response to other comments. At no point does the text state that the problem is limited to tropical forests.

*9, 5: This discussion doesn't include all fire inventories so change "Three of them use a bottom up approach" to "Four of them (for example) use a bottom up approach"*

*9, 7: change "The other three products are top-down, based on fire radiative power (FRP):" to "Two other products are top-down:" since GFAS is bottom-up, FRP is still just based on hotspots, and (for example) Ron Cohen's group (Mebust et al) also has a top-down approach.*

*9, 9: Agree with Charles Ichoku, GFAS is bottom-up. In this section on how much biomass is burned it could help to foreshadow the later discussion of CO inversions, list sources of uncertainty, and the other issues I noted in my general comments above.*

I eliminated the whole top-down/bottom-up terminology. See my response to Ichoku.

*9, 25: Are global numbers for reactive gases still important? More important than Table 3 might be to include a summary of what is new in this compilation as discussed in my general comments.*

Such a summary has been added (see above)

*9, 28 "the previous assessment" should be "A&M2001" since there are so many global estimates.*

Changed to "our previous assessment".

*9, 30-32: The fire to fire variability and even real day to day variability for a single fire can be much higher than the standard deviation of the literature mean. This can be important in many modeling applications (Yates et al., 2016). Change to "global emissions uncertainties" on line 32.*

Done.

*10, 1-14: This discussion is useful and adds confidence to global totals. There is a large body of work in this area and I have not attempted a comprehensive critique, but like the idea of using multiple CO products as noted above.*

Thanks!

*10, 15: A11 also reported these differences so useful to change to "As noted in A11, major ..."*

Done.

*10, 20-22: I would rephrase this to say that there has been good progress in OVOC and HCN emissions as just noted and in reducing the percentage of un-identified compounds, sampling under-sampled sources, measuring I/SVOC, and sampling post-emission evolution, but quantifying global activity levels remains difficult. This is to be expected due to clouds, orbital gaps, small fires, unknown injection altitudes and diurnal cycles, etc. More measurements can add info but not reduce natural variability. Measuring EF and quantifying biomass burned present a different set of challenges. Most model inputs cannot be measured operationally. Thus, the author's proposed CO inversions are just one idea.*

 I prefer my text. The conclusions are the place where the author should present his take-away from the previous discussion. I admit to having a bias towards global perspectives and carbon cycle issues.

*10, 29: Table 1 doesn't include the major new research front in I/SVOC when it comes to setting future priorities.*

This has now been added in the conclusions.

*11, 6: The conclusions remain focused on the problem of estimating global totals, which is just one part of BB research. It may not be the most important part, but is probably the hardest.*

*Bottom-up or top-down models are super-sensitive to plume injection altitude, terrain flattening, diurnal cycles, complex transport, and chemical/physical evolution; often at subgrid scales.*

*These things cannot be measured operationally. Actual recent/upcoming work such as WE-CAN and FIREX-AQ focus instead on advanced instrumentation and combining an unprecedented scope of airborne and ground-based measurements with new satellite products. This will eventually also be helpful to estimating global totals.*

I admit to having a bias towards global perspectives and carbon cycle issues. I am also looking forward to the results of the campaigns mentioned by the reviewer.